# On the Convergence of No-Regret Learning Dynamics in Time-Varying Games

**Ioannis Anagnostides**
Carnegie Mellon University
`ianagnos@cs.cmu.edu`

**Ioannis Panageas**
University of California Irvine
`ipanagea@ics.uci.edu`

**Gabriele Farina**
MIT
`gfarina@mit.edu`

**Tuomas Sandholm**
Carnegie Mellon University
Strategic Machine, Inc.
Strategy Robot, Inc.
Optimized Markets, Inc.
`sandholm@cs.cmu.edu`

## Abstract

Most of the literature on learning in games has focused on the restrictive setting where the underlying repeated game does not change over time. Much less is known about the convergence of no-regret learning algorithms in dynamic multiagent settings. In this paper, we characterize the convergence of *optimistic gradient descent (OGD)* in time-varying games. Our framework yields sharp convergence bounds for the equilibrium gap of OGD in zero-sum games parameterized on natural variation measures of the sequence of games, subsuming known results for static games. Furthermore, we establish improved *second-order* variation bounds under strong convexity-concavity, as long as each game is repeated multiple times. Our results also extend to time-varying *general-sum* multi-player games via a bilinear formulation of correlated equilibria, which has novel implications for meta-learning and for obtaining refined variation-dependent regret bounds, addressing questions left open in prior papers. Finally, we leverage our framework to also provide new insights on dynamic regret guarantees in static games.

## 1 Introduction

Most of the classical results in the literature on learning in games—exemplified by, among others, the work of Hart and Mas-Colell [49], Foster and Vohra [42], Freund and Schapire [43]—rest on the assumption that the underlying repeated game remains invariant. Yet, in many paradigmatic learning environments that is unrealistic [35, 93, 17, 59]. One such class is settings where the underlying game is actually changing, such as routing problems on the internet [58], online advertising auctions [55], and dynamic mechanism design [67, 31]. Another such class consists of settings in which many similar games need to be solved [48]. For example, one may want to solve variations of a game for the purpose of sensitivity analysis with respect to the modeling assumptions used to construct the game model.

Despite the considerable interest in such dynamic multiagent environments, much less is known about the convergence of *no-regret learning* algorithms in time-varying games. No-regret dynamics are natural learning algorithms that have desirable convergence properties in static settings. Also, the state-of-the-art algorithms for finding minimax equilibria in two-player zero-sum games are based on advanced forms of no-regret dynamics [37, 11]. Indeed, all the superhuman milestones in poker have used them in the equilibrium-finding module of their architectures [8, 10, 12].

37th Conference on Neural Information Processing Systems (NeurIPS 2023).

In this paper, we seek to fill this knowledge gap by understanding properties of no-regret dynamics in time-varying games. In particular, we primarily investigate the convergence of *optimistic gradient descent (OGD)* [20, 72] in time-varying games. Unlike traditional no-regret learning algorithms, such as (online) gradient descent, OGD has been recently shown to exhibit *last-iterate* convergence in *static* (two-player) zero-sum games [27, 46, 16, 47]. For the more challenging scenario where the underlying game can vary in every round, a fundamental question arises: *Under what conditions on the sequence of games does OGD approximate (with high probability) the sequence of Nash equilibria?*

## 1.1 Our results

In this paper, we develop a framework that enables us to characterize the convergence of OGD, and generalizations thereof, in a number of fundamental time-varying multiagent settings.

**Bilinear saddle-point problems**  First, building on the work of Zhang et al. [93], we identify natural variation measures on the sequence of games whose sublinear growth guarantees that *almost all* iterates of OGD under a constant learning rate are approximate Nash equilibria in time-varying (two-player) zero-sum games (Corollary 3.4). More precisely, in Theorem 3.3 we derive a sharp non-asymptotic characterization of the equilibrium gap of OGD as a function of the *minimal first-order* variation of *approximate* Nash equilibria and the *second-order* variation of the payoff matrices. We stress that our variation measure that depends on the deviation of approximate Nash equilibria of the games can be arbitrarily smaller than the one based on (even the least varying) sequence of exact Nash equilibria (Proposition 3.2), thereby significantly sharpening the measure considered by Zhang et al. [93]. It is also a compelling property, in light of the multiplicity of Nash equilibria, that the variation of the Nash equilibria is measured in terms of the most favorable—*i.e.*, one that minimizes the variation—such sequence.

From a technical standpoint, our analysis revolves around a connection between the convergence of OGD in time-varying games and *dynamic regret*. In particular, the first key observation is that *dynamic regret cannot be too negative under any sequence of approximate Nash equilibria* (Property 3.1); this was first observed by Zhang et al. [93] under a sequence of *exact* Nash equilibria. We also generalize this property by connecting it to the admission of a *minimax theorem* (Property A.4), as well as the so-called *MVI property* in the more general context of time-varying variational inequalities (VIs); this will enable us to extend our scope to certain multi-player games such as *polymatrix zero-sum* [15] (Theorem A.16). By combining Property 3.1 with a dynamic *RVU bound* [93], we obtain a variation-dependent bound for the second-order path length of OGD under a constant learning rate in time-varying zero-sum games. In turn, this leads to our first main result, Theorem 3.3, discussed above. In the special case of static games, our theorem reduces to a tight $T^{-1/2}$ rate. We also instantiate our results to the special setting of *meta-learning*, where each game is repeated multiple times.

**Strongly convex-concave games**  Moreover, for strongly convex-concave time-varying games, we obtain a refined *second-order* variation bound on the sequence of Nash equilibria, as long as each game is repeated multiple times (Theorem 3.5); this is inspired by an improved second-order bound for dynamic regret under analogous conditions due to Zhang et al. [92]. As a byproduct of our techniques, we point out that *any* no-regret learners are approaching a Nash equilibrium under strong convexity-concavity (Proposition 3.6; *cf.* [85]). Those results apply even in non-strongly convex-concave settings by suitably trading off the magnitude of a regularizer that makes the game strongly convex-concave. This offers significant gains in the meta-learning setting as well.

**General-sum multi-player games**  Next, we extend our results to time-varying general-sum multi-player games by expressing *correlated equilibria* via a bilinear saddle-point problem (BSPP); while this formulation goes back to the early work of Hart and Schmeidler [50], it is rarely employed in the literature on learning in games. By leveraging a structural property of the underlying BSPP, we manage to obtain convergence bounds parameterized on the variation of the correlated equilibria (Theorem 3.9). To illustrate the power of our framework, we immediately recover natural and algorithm-independent similarity measures for the meta-learning setting (Proposition A.14) even in general games (Corollary A.25), thereby addressing an open question of Harris et al. [48]. Our techniques also imply new per-player regret bounds in zero-sum and general-sum games (Corollaries A.12 and A.26), the latter addressing a question left open by Zhang et al. [93], albeit under a different learning paradigm (Section 3.3 contains a discussion on this point). We further

parameterize the convergence of (vanilla) gradient descent in time-varying *potential* games in terms of the deviation of the potential functions (Theorem 3.7).

Finally, building on our techniques in time-varying games, we investigate the best dynamic-regret guarantees possible in *static* games (see also the work of Cai and Zheng [14]). We first note that instances of optimistic mirror descent guarantee $O(\sqrt{T})$ dynamic per-player regret (Proposition 3.10), matching the known rate of (online) gradient descent but for the significantly weaker notion of *external* regret. We further point out that $O(\log T)$ dynamic regret is attainable, but in a stronger two-point feedback model. In stark contrast, even obtaining sublinear dynamic regret for each player is precluded in general-sum games (Proposition 3.12). This motivates studying a relaxation of dynamic regret that constrains the number of switches in the comparator, for which we derive accelerates rates in general games (Theorem 3.13) by leveraging the techniques of Syrgkanis et al. [78] in conjunction with the dynamic RVU bound. Interesting, this relaxation of dynamic regret gives rise to a natural refinement of coarse correlated equilibria, recently investigated by Crippa et al. [22].

## 1.2 Further related work

Even in static (two-player) zero-sum games, the pointwise convergence of no-regret learning algorithms is a tenuous affair. Indeed, traditional learning dynamics within the no-regret framework, such as (online) mirror descent, may even diverge away from the equilibrium; *e.g.*, see [74, 60, 83, 45]. Notwithstanding the lack of pointwise convergence, the *empirical frequency* of no-regret learners is well-known to approach the set of Nash equilibria in zero-sum games [43], and the set of coarse correlated equilibria in general-sum games [49, 42]—a standard relaxation of the Nash equilibrium [63, 3]. Unfortunately, those classical results are of little use beyond static games, thereby offering a crucial impetus for investigating iterate-convergence in games with a time-varying component—a ubiquitous theme in many practical scenarios of interest [31, 67, 82, 44, 81, 71, 69, 89, 70, 6]. One concrete and quite broad example worth mentioning here revolves around the *steering* problem [91], where a mediator dynamically modifies the utilities of the game via nonnegative and vanishing payments so as to guide no-regret learners to desirable equilibria. In particular, when the payment function varies over time, the steering problem can be naturally phrased as a time-varying zero-sum game [91].

In this context, there has been a considerable effort endeavoring to extend the scope of traditional game-theoretic results to the time-varying setting, approached from a variety of different standpoints in prior work [55, 93, 17, 58, 59, 35, 41, 79, 40, 86]. In particular, our techniques in Section 3.1 are strongly related to the ones developed by Zhang et al. [93], but our primary focus is different: Zhang et al. [93] were mainly interested in obtaining variation-dependent regret bounds, while our results revolve around iterate-convergence of OGD (under a constant learning rate) to Nash equilibria. Minimizing regret and approaching Nash equilibria are two inherently distinct problems, although connections have emerged [2, 93], and are further cultivated in this paper. Moreover, in an independent work Feng et al. [40] established a surprising separation between the behavior of optimistic GDA and the extra-gradient method in time-varying unconstrained bilinear games; it is open whether such a discrepancy persists in the constrained setting as well. We also point out the concurrent work of Yan et al. [86] that focuses on improved guarantees for strongly monotone games; the results we obtain in Section 3.2 are complementary to theirs.

Another closely related direction is on *meta-learning* in games [48, 77, 34], where each game can be repeated for multiple iterations. Such considerations are motivated in part by a number of use-cases in which many "similar" games—or multiple game variations—ought to be solved [9], such as Poker with different stack-sizes. While the meta-learning problem is a special case of our general setting, our results are strong enough to have new implications for meta-learning in games, even though the algorithms considered herein are not tailored to operate in that setting.

## 2 Preliminaries

**Notation**  We let $\mathbb{N} := \{1, 2, \dots, \}$ be the set of natural numbers. For a number $p \in \mathbb{N}$, we let $[\![p]\!] := \{1, \dots, p\}$. For a vector $\boldsymbol{w} \in \mathbb{R}^d$, we use $\|\boldsymbol{w}\|_2$ to represent its Euclidean norm; we also overload that notation so that $\|\cdot\|_2$ denotes the spectral norm when the argument is a matrix.

For a two-player zero-sum game, we denote by $\mathcal{X} \subseteq \mathbb{R}^{d_x}$ and $\mathcal{Y} \subseteq \mathbb{R}^{d_y}$ the strategy sets of the two players—namely, Player $x$ and Player $y$, respectively—where $d_x, d_y \in \mathbb{N}$ represent the corresponding

dimensions. It is assumed that $\mathcal{X}$ and $\mathcal{Y}$ are nonempty convex and compact sets. For example, in the special case where $\mathcal{X} := \Delta^{d_x}$ and $\mathcal{Y} := \Delta^{d_y}$—each set corresponds to a probability simplex—the game is said to be in *normal form*. Further, we denote by $D_{\mathcal{X}}$ the $\ell_2$-diameter of $\mathcal{X}$, and by $\|\mathcal{X}\|_2$ the maximum $\ell_2$-norm attained by a point in $\mathcal{X}$. We will always assume that the strategy sets remain invariant, while the payoff matrix can change in each round. For notational convenience, we will denote by $z := (x, y)$ the concatenation of $x$ and $y$, and by $\mathcal{Z} := \mathcal{X} \times \mathcal{Y}$ the Cartesian product of $\mathcal{X}$ and $\mathcal{Y}$. In general $n-$player games, we instead use subscripts indexed by $i \in [\![n]\!]$ to specify quantities related to a player. Superscripts are typically reserved to identify the time index. Finally, to simplify the exposition, we often use the $O(\cdot)$ notation to suppress time-independent parameters of the problem.

**Dynamic regret**   We operate in the usual online learning setting under full feedback. Namely, at every time $t \in \mathbb{N}$ the learner decides on a strategy $x^{(t)} \in \mathcal{X}$, and then subsequently observes a utility function $x \mapsto \langle x, u_x^{(t)} \rangle$, for $u_x^{(t)} \in \mathbb{R}^{d_x}$. A strong performance benchmark in this online setting is *dynamic regret*, defined for a time horizon $T \in \mathbb{N}$ as follows:

$$\mathrm{DReg}_x^{(T)}(s_x^{(T)}) := \sum_{t=1}^{T} \langle x^{(t,\star)} - x^{(t)}, u_x^{(t)} \rangle, \tag{1}$$

where $s_x^{(T)} := (x^{(1,\star)}, \ldots, x^{(T,\star)}) \in \mathcal{X}^T$ above is the sequence of *comparators*; by setting $x^{(1,\star)} = x^{(2,\star)} = \cdots = x^{(T,\star)}$ in (1) we recover the standard notion of *(external) regret* (denoted simply by $\mathrm{Reg}_x^{(T)}$), which is commonly used to establish convergence of the time-average strategies in static two-player zero-sum games [43]. On the other hand, the more general notion of dynamic regret, introduced in (1), has been extensively used in more dynamic environments; *e.g.*, [94, 92, 53, 18, 51, 54, 62]. We also define $\mathrm{DReg}_x^{(T)} := \max_{s_x^{(T)} \in \mathcal{X}^T} \mathrm{DReg}_x^{(T)}(s_x^{(T)})$. While ensuring $o(T)$ dynamic regret is clearly hopeless in a truly adversarial environment, Section 3.4 reveals that non-trivial guarantees are possible when learning in zero-sum games (see also [14]).

**Optimistic gradient descent**   *Optimistic gradient descent (OGD)* [20, 72] is a no-regret algorithm defined with the following update rule:

$$\begin{aligned} x^{(t)} &:= \Pi_{\mathcal{X}} \left( \hat{x}^{(t)} + \eta m_x^{(t)} \right), \\ \hat{x}^{(t+1)} &:= \Pi_{\mathcal{X}} \left( \hat{x}^{(t)} + \eta u_x^{(t)} \right). \end{aligned} \tag{OGD}$$

Here, $\eta > 0$ is the *learning rate*; $\hat{x}^{(1)} := \arg\min_{\hat{x} \in \mathcal{X}} \|\hat{x}\|_2^2$ represents the initialization of OGD; $m_x^{(t)} \in \mathbb{R}^{d_x}$ is the *prediction vector* at time $t$, and it is set as $m_x^{(t)} := u_x^{(t-1)}$ when $t \geq 2$, and $m_x^{(1)} := \mathbf{0}_{d_x}$; and finally, $\Pi_{\mathcal{X}}(\cdot)$ represents the Euclidean projection to the set $\mathcal{X}$, which is well-defined, and can be further computed efficiently for structured sets, such as the probability simplex. For our purposes, we will posit access to a projection oracle for the set $\mathcal{X}$, in which case the update rule (OGD) is indeed efficiently implementable.

In a multi-player $n$-player game, each player $i \in [\![n]\!]$ is associated with a utility function $u_i : \bigtimes_{i=1}^{n} \mathcal{X}_i \to \mathbb{R}$. We recall the following central definition [64].

**Definition 2.1** (Approximate Nash equilibrium). A joint strategy profile $(x_1^\star, \ldots, x_n^\star) \in \bigtimes_{i=1}^{n} \mathcal{X}_i$ is an $\epsilon$-approximate Nash equilibrium (NE), for an $\epsilon \geq 0$, if for any player $i \in [\![n]\!]$ and any possible deviation $x_i' \in \mathcal{X}_i$ it holds that $u_i(x_1^\star, \ldots, x_i^\star, \ldots, x_n^\star) \geq u_i(x_1^\star, \ldots, x_i', \ldots, x_n^\star) - \epsilon$.

## 3   Convergence in time-varying games

In this section, we formalize our results regarding convergence in time-varying games. We organize this section as follows: First, in Section 3.1, we study the convergence of OGD in time-varying bilinear saddle-point problems (BSPPs) and beyond, culminating in the non-asymptotic characterization of Theorem 3.3; Section 3.2 formalizes our improvements under strong convexity-concavity; we then extend our scope to time-varying multi-player general-sum and potential games in Section 3.3; and finally, Section 3.4 is concerned with dynamic regret guarantees in static games.

### 3.1 Bilinear saddle-point problems

We first study an online learning setting wherein two players interact in a sequence of time-varying BSPPs [93]. We assume that in every repetition $t \in \llbracket T \rrbracket$ the players select a pair of strategies $(\boldsymbol{x}^{(t)}, \boldsymbol{y}^{(t)}) \in \mathcal{X} \times \mathcal{Y}$. Then, Player $x$ receives the utility $\boldsymbol{u}_x^{(t)} := -\mathbf{A}^{(t)}\boldsymbol{y}^{(t)} \in \mathbb{R}^{d_x}$, where $\mathbf{A}^{(t)} \in \mathbb{R}^{d_x \times d_y}$ represents the payoff matrix at the $t$-th repetition; similarly, Player $y$ receives the utility $\boldsymbol{u}_y^{(t)} := (\mathbf{A}^{(t)})^\top \boldsymbol{x}^{(t)} \in \mathbb{R}^{d_y}$. We also extend our scope to a more general class of time-varying problems, as we formalize in Section 3.1.1. The proofs of this subsection are in Appendix A.1.

We commence by pointing out a crucial property: by selecting a sequence of approximate Nash equilibria as the comparators, the *sum* of the players' dynamic regrets *cannot be too negative*:

**Property 3.1.** *Suppose that* $\mathcal{Z} \ni \boldsymbol{z}^{(t,\star)} = (\boldsymbol{x}^{(t,\star)}, \boldsymbol{y}^{(t,\star)})$ *is an* $\epsilon^{(t)}$*-approximate Nash equilibrium of the $t$-th game. Then, for* $s_x^{(T)} = (\boldsymbol{x}^{(t,\star)})_{1 \le t \le T}$ *and* $s_y^{(T)} = (\boldsymbol{y}^{(t,\star)})_{1 \le t \le T}$,

$$\mathrm{DReg}_x^{(T)}(s_x^{(T)}) + \mathrm{DReg}_y^{(T)}(s_y^{(T)}) \ge -2\sum_{t=1}^{T} \epsilon^{(t)}.$$

A special case of this property was previously noted by Zhang et al. [93] by considering a sequence of *exact* Nash equilibria; the approximate version stated above leads to a significant improvement in the convergence bounds, as we shall see in the sequel. In fact, as we note in Property A.4, Property 3.1 applies even in certain (time-varying) nonconvex-nonconcave min-max optimization problems, and it is a consequence of the *minimax theorem*. Property 3.1 also holds for time-varying variational inequalities (VIs) that satisfy the so-called *MVI property*, as we elaborate further in Section 3.1.1.

Now, building on the work of Zhang et al. [93], let us introduce some natural measures of the games' variation. The first-order variation of the Nash equilibria was defined by Zhang et al. [93] for $T \ge 2$ as $\mathcal{V}_{\mathrm{NE}}^{(T)} := \inf_{\boldsymbol{z}^{(t,\star)} \in \mathcal{Z}^{(t,\star)}, \forall t \in \llbracket T \rrbracket} \sum_{t=1}^{T-1} \|\boldsymbol{z}^{(t+1,\star)} - \boldsymbol{z}^{(t,\star)}\|_2$, where $\mathcal{Z}^{(t,\star)}$ is the (nonempty) set of Nash equilibria of the $t$-th game. We recall that there can be a multiplicity of Nash equilibria [80]; as such, a compelling feature of the above variation measure is that it depends on the most favorable sequence of Nash equilibria—one that minimizes the first-order variation.

It is important, however, to come to terms with the well-known fact that Nash equilibria can change abruptly even under a "small" perturbation in the payoff matrix (see Example A.5), which is an important caveat of the variation $\mathcal{V}_{\mathrm{NE}}^{(T)}$. To address this, and in accordance with our more general Property 3.1, we consider a more favorable variation measure, defined as

$$\mathcal{V}_{\epsilon-\mathrm{NE}}^{(T)} := \inf \left\{ \sum_{t=1}^{T-1} \|\boldsymbol{z}^{(t+1,\star)} - \boldsymbol{z}^{(t,\star)}\|_2 + C \sum_{t=1}^{T} \epsilon^{(t)} \right\}, \tag{2}$$

for a sufficiently large parameter $C > 0$ (see (28)); the infimum above is subject to $\epsilon^{(t)} \in \mathbb{R}_{\ge 0}$ and $\boldsymbol{z}^{(t,\star)} \in \mathcal{Z}_{\epsilon^{(t)}}^{(t,\star)}$ for all $t \in \llbracket T \rrbracket$, where $\mathcal{Z}_{\epsilon^{(t)}}^{(t,\star)}$ is the set of $\epsilon^{(t)}$-approximate NE. It is evident that $\mathcal{V}_{\epsilon-\mathrm{NE}}^{(T)} \le \mathcal{V}_{\mathrm{NE}}^{(T)}$ since one can take $\epsilon^{(1)} = \cdots = \epsilon^{(T)} = 0$; in fact, $\mathcal{V}_{\epsilon-\mathrm{NE}}^{(T)}$ can be arbitrarily smaller:

**Proposition 3.2.** *For any* $T \ge 4$*, there is a sequence of $T$ games such that* $\mathcal{V}_{NE}^{(T)} \ge \frac{T}{2}$ *while* $\mathcal{V}_{\epsilon-NE}^{(T)} \le \delta$*, for any* $\delta > 0$.

This shows that $\mathcal{V}_{\epsilon-\mathrm{NE}}^{(T)}$ is a more compelling variation measure compared to $\mathcal{V}_{\mathrm{NE}}^{(T)}$. It is worth noting here that Zhang et al. [93] also considered the variation measure $\mathcal{W}_{\mathbf{A}}^{(T)} := \sum_{t=1}^{T} \|\mathbf{A}^{(t)} - \bar{\mathbf{A}}\|_2$, where $\bar{\mathbf{A}}$ is the average payoff matrix and $\|\cdot\|_2$ denotes the spectral norm.[1] It is in fact not hard to construct instances such that $\mathcal{V}_{\epsilon-\mathrm{NE}}^{(T)} \ll \min\{\mathcal{V}_{\mathrm{NE}}^{(T)}, \mathcal{W}_{\mathbf{A}}^{(T)}\}$ (Proposition A.6).

Moreover, we also recall another quantity that captures the variation of the payoff matrices: $\mathcal{V}_{\mathbf{A}}^{(T)} := \sum_{t=1}^{T-1} \|\mathbf{A}^{(t+1)} - \mathbf{A}^{(t)}\|_2^2$. Unlike $\mathcal{V}_{\mathrm{NE}}^{(T)}$, the variation measure $\mathcal{V}_{\mathbf{A}}^{(T)}$ depends on the *second-order* variation (of the payoff matrices), which could translate to a lower-order impact compared to $\mathcal{V}_{\mathrm{NE}}^{(T)}$

---

[1]Any equivalent norm in the definition of $\mathcal{W}_{\mathbf{A}}^{(T)}$ suffices for the purpose of Proposition A.6.

(see, *e.g.*, Corollary A.9). We stress that while our convergence bounds will be parameterized based on $\mathcal{V}_{\epsilon-NE}^{(T)}$ and $\mathcal{V}_{\mathbf{A}}^{(T)}$, the underlying algorithm—namely `OGD`—will remain oblivious to those variation measures. We only make the mild assumption that players know in advance the value of $L := \max_{1 \leq t \leq T} \|\mathbf{A}^{(t)}\|_2$, so that they can tune the learning rate $\eta$ appropriately.

We are now ready to state the main result of this subsection. Below, we use the notation $\text{EQGAP}^{(t)}(\boldsymbol{z}^{(t)})$ to represent the Nash equilibrium gap of $(\boldsymbol{x}^{(t)}, \boldsymbol{y}^{(t)}) = \boldsymbol{z}^{(t)} \in \mathcal{Z}$ at the $t$-th game. More precisely, $\text{EQGAP}^{(t)}(\boldsymbol{z}^{(t)}) := \max\{\text{BR}_x^{(t)}(\boldsymbol{x}^{(t)}), \text{BR}_y^{(t)}(\boldsymbol{y}^{(t)})\}$, where $\text{BR}_x^{(t)}(\boldsymbol{x}^{(t)}) := \max_{\boldsymbol{x} \in \mathcal{X}}\{\langle \boldsymbol{x}, \boldsymbol{u}_x^{(t)}\rangle\} - \langle \boldsymbol{x}^{(t)}, \boldsymbol{u}_x^{(t)}\rangle$ is the best response gap of Player $x$ (and analogously for Player $y$).

**Theorem 3.3** (Detailed version in Theorem A.11). *Suppose that both players employ* `OGD` *with learning rate* $\eta = \frac{1}{4L}$ *in a sequence of time-varying BSPPs, where* $L := \max_{1 \leq t \leq T} \|\mathbf{A}^{(t)}\|_2$. *Then,*
$\sum_{t=1}^{T} \left(\text{EQGAP}^{(t)}(\boldsymbol{z}^{(t)})\right)^2 = O\left(1 + \mathcal{V}_{\epsilon-NE}^{(T)} + \mathcal{V}_{\mathbf{A}}^{(T)}\right)$, *where* $(\boldsymbol{z}^{(t)})_{1 \leq t \leq T}$ *is the sequence of joint strategy profiles produced by* `OGD`.

The proof of this theorem is based on upper bounding the *second-order path length* of `OGD`, $\sum_{t=1}^{T} \left(\|\boldsymbol{z}^{(t)} - \hat{\boldsymbol{z}}^{(t)}\|_2^2 + \|\boldsymbol{z}^{(t)} - \hat{\boldsymbol{z}}^{(t+1)}\|_2^2\right)$, as a function of the variation measures $\mathcal{V}_{\epsilon-NE}^{(T)}$ and $\mathcal{V}_{\mathbf{A}}^{(T)}$. This is shown via a *dynamic RVU* bound [93] (Lemmas A.1 and A.2) in conjunction with Property 3.1.

It is worth noting that when the deviation of the payoff matrices is controlled by the deviation of the players' strategies, in the sense that $\sum_{t=1}^{T-1} \|\mathbf{A}^{(t+1)} - \mathbf{A}^{(t)}\|_2^2 \leq W^2 \sum_{t=1}^{T-1} \|\boldsymbol{z}^{(t+1)} - \boldsymbol{z}^{(t)}\|_2^2$ for some parameter $W \in \mathbb{R}_{>0}$, the variation measure $\mathcal{V}_{\mathbf{A}}^{(T)}$ in Theorem 3.3 can be entirely eliminated; see Corollary A.9. The same in fact applies under an improved prediction mechanism (see Remark A.13); while that prediction is not implementable in the online learning setting, it can be used, for example, when the sequence of games is known in advance (as is the case in certain applications).

We next state some immediate consequences of Theorem 3.3. (Item 2 below follows from Theorem 3.3 by Jensen's inequality.)

**Corollary 3.4.** *In the setting of Theorem 3.3,*

1. *If at least a $\delta$-fraction of the iterates have $\epsilon > 0$ NE gap, $\epsilon^2 \delta \leq O\left(\frac{1}{T}\left(1 + \mathcal{V}_{\epsilon-NE}^{(T)} + \mathcal{V}_{\mathbf{A}}^{(T)}\right)\right)$.*

2. *The average NE gap is bounded as $O\left(\sqrt{\frac{1}{T}\left(1 + \mathcal{V}_{\epsilon-NE}^{(T)} + \mathcal{V}_{\mathbf{A}}^{(T)}\right)}\right)$.*

In particular, in terms of asymptotic implications, if $\lim_{T \to +\infty} \frac{\mathcal{V}_{\epsilon-NE}^{(T)}}{T}, \lim_{T \to +\infty} \frac{\mathcal{V}_{\mathbf{A}}^{(T)}}{T} = 0$, then (i) for any $\epsilon > 0$ the fraction of iterates of `OGD` with at least an $\epsilon$ Nash equilibrium gap converges to 0; and (ii) the average Nash equilibrium gap of the iterates of `OGD` converges to 0.

In the special case where $\mathcal{V}_{\epsilon-NE}^{(T)}, \mathcal{V}_{\mathbf{A}}^{(T)} = O(1)$, Theorem 3.3 recovers the $T^{-1/2}$ iterate-convergence rate of `OGD` in static bilinear saddle-point problems.

**Meta-learning** Our results also have immediate applications in the *meta-learning* setting. More precisely, meta-learning in games is a special case of time-varying games which consists of a sequence of $H \in \mathbb{N}$ separate games, each of which is repeated for $m \in \mathbb{N}$ consecutive rounds, so that $T := m \times H$. The central goal in meta-learning is to obtain convergence bounds parameterized by the *similarity* of the games; identifying suitable similarity metrics is a central question in that line of work.

In this context, we highlight that Theorem 3.3 shown above readily provides a meta-learning guarantee parameterized by the following notion of similarity between the Nash equilibria: $\inf_{\boldsymbol{z}^{(h,\star)} \in \mathcal{Z}^{(h,\star)}, \forall h \in \llbracket H \rrbracket} \sum_{h=1}^{H-1} \|\boldsymbol{z}^{(h+1,\star)} - \boldsymbol{z}^{(h,\star)}\|_2$, where $\mathcal{Z}^{(h,\star)}$ is the set of Nash equilibria of the $h$-th game in the meta-learning sequence,[2] as well as the similarity of the payoff matrices—corresponding to the term $\mathcal{V}_{\mathbf{A}}^{(T)}$. In fact, under a suitable prediction—the one used by Harris et al. [48]—the dependence on $\mathcal{V}_{\mathbf{A}}^{(T)}$ can be entirely removed; see Proposition A.14 for our formal result. A compelling aspect of our meta-learning guarantee is that the considered algorithm is oblivious to the boundaries of the meta-learning. It is also worth noting that our similarity metric can be arbitrarily

---

[2]In accordance with Theorem 3.3, this similarity metric can be refined using a sequence of approximate NE.

smaller than the one considered by Harris et al. [48] (Proposition A.15). We further provide some novel results on meta-learning in general-sum games in Section 3.3.

### 3.1.1 Beyond bilinear saddle-point problems

As we alluded to earlier, our approach can be generalized beyond time-varying bilinear saddle-point problems to a more general class of time-varying *variational inequality (VIs)* as follows. Let $F^{(t)} : \mathcal{Z} \to \mathcal{Z}$ be the (single-valued) operator of the VI problem at time $t \in \mathbb{N}$. $F^{(t)}$ is said to satisfy the *MVI property* [61, 36] if there exists a point $\mathbf{z}^{(t,\star)} \in \mathcal{Z}$ such that $\langle \mathbf{z} - \mathbf{z}^{(t,\star)}, F^{(t)}(\mathbf{z}) \rangle \geq 0$ for any $\mathbf{z} \in \mathcal{Z}$. For example, in the special case of a bilinear saddle-point problem we have that $F : \mathbf{z} := (\mathbf{x}, \mathbf{y}) \mapsto (\mathbf{A}\mathbf{y}, -\mathbf{A}^\top \mathbf{x})$, and the MVI property is satisfied by virtue of Von Neumann's minimax theorem. It is direct to see that Property 3.1 applies to any time-varying sequence of VIs with respect to $(\mathbf{z}^{(t,\star)})_{1 \leq t \leq T}$ as long as every operator in the sequence $(F^{(1)}, \dots, F^{(T)})$ satisfies the MVI property. (Even more broadly, it would suffice if almost all operators in the sequence—in that their fraction converges to 1 as $T \to +\infty$—satisfy the MVI property.) This observation enables extending Theorem 3.3 to a more general class of problems. As a concrete example, we provide a generalization of Theorem 3.3 (Theorem A.16) to *polymatrix zero-sum* games [15] in Appendix A.1.7. By contrast, in problems where the MVI property fails it appears that a much different approach is called for.

We finally point out that our framework could have certain implications for solving (static) general VIs, as we discuss in Appendix A.1.8.

## 3.2 Strongly convex-concave games

In this subsection, we show that under additional structure we can significantly improve the variation measures established in Theorem 3.3. More precisely, we first assume that each objective function $f(\mathbf{x}, \mathbf{y})$ is $\mu$-strongly convex with respect to $\mathbf{x}$ and $\mu$-strongly concave with respect to $\mathbf{y}$. Our second assumption is that each game is played for *multiple* rounds $m \in \mathbb{N}$, instead of only a single round; this is akin to the meta-learning setting. The key insight is that as long as $m$ is large enough, $m = \Omega(1/\mu)$, those two assumptions suffice to obtain a *second-order* variation bound in terms of the sequence of Nash equilibria, $\mathcal{S}_{\mathrm{NE}}^{(H)} := \sum_{h=1}^{H-1} \|\mathbf{z}^{(h+1,\star)} - \mathbf{z}^{(h,\star)}\|_2^2$, where $\mathbf{z}^{(h,\star)}$ is a Nash equilibrium of the $h$-th game. This significantly refines the result of Theorem 3.3, and is inspired by the improved dynamic regret bounds obtained by Zhang et al. [92]. Below we sketch the key ideas of the improvement; full proofs are deferred to Appendix A.2.

In this setting, it is assumed that Player $x$ obtains the utility $\mathbf{u}_x^{(t)} := -\nabla_{\mathbf{x}} f^{(t)}(\mathbf{x}^{(t)}, \mathbf{y}^{(t)})$ at every time $t \in [\![T]\!]$, while its regret will be denoted by $\mathrm{Reg}_{\mathcal{L},y}^{(T)}$; similar notation applies for Player $y$. The first observation is that, focusing on a single (static) game, under strong convexity-concavity the sum of the players' regrets are *strongly nonnegative* (Lemma A.18):

$$\mathrm{Reg}_{\mathcal{L},x}^{(m)}(\mathbf{x}^\star) + \mathrm{Reg}_{\mathcal{L},y}^{(m)}(\mathbf{y}^\star) \geq \frac{\mu}{2} \sum_{t=1}^{m} \|\mathbf{z}^{(t)} - \mathbf{z}^\star\|_2^2, \tag{3}$$

for any Nash equilibrium $\mathbf{z}^\star \in \mathcal{Z}$ of the game. In turn, this can be cast in terms of dynamic regret over the sequence of the $h$ games (Lemma A.19). Next, combining those dynamic-regret lower bounds with a suitable RVU-type property leads to a refined second-order path length bound as long as $m = \Omega(1/\mu)$, which in turn leads to our main result below. Before we present its statement, let us introduce the following measure of variation of the gradients: $\mathcal{V}_{\nabla f}^{(H)} := \sum_{h=1}^{H-1} \max_{\mathbf{z} \in \mathcal{Z}} \|F^{(h+1)}(\mathbf{z}) - F^{(h)}(\mathbf{z})\|_2^2$, where let $F : \mathbf{z} := (\mathbf{x}, \mathbf{y}) \mapsto (\nabla_{\mathbf{x}} f(\mathbf{x}, \mathbf{y}), -\nabla_{\mathbf{y}} f(\mathbf{x}, \mathbf{y}))$. This variation measure is analogous to $\mathcal{V}_{\mathbf{A}}^{(T)}$ we introduced earlier for time-varying BSPPs.

**Theorem 3.5** (Detailed version in Theorem A.21). *Let $f^{(h)} : \mathcal{X} \times \mathcal{Y}$ be a $\mu$-strongly convex-concave and $L$-smooth function, for all $h \in [\![H]\!]$. Suppose further that both players employ* OGD *with learning rate $\eta = \min\left\{\frac{1}{8L}, \frac{1}{2\mu}\right\}$ for $T \in \mathbb{N}$ repetitions, where $T = m \times H$ and $m \geq \frac{2}{\eta\mu}$. Then,*

$$\sum_{t=1}^{T} \left(\mathrm{EqGap}^{(t)}(\mathbf{z}^{(t)})\right)^2 = O(1 + \mathcal{S}_{NE}^{(H)} + \mathcal{V}_{\nabla f}^{(H)}).$$

Our techniques also imply the improved regret bounds $\text{Reg}_{\mathcal{L},x}^{(T)}, \text{Reg}_{\mathcal{L},y}^{(T)} = O\left(\sqrt{1 + \mathcal{S}_{\text{NE}}^{(H)} + \mathcal{V}_{\nabla f}^{(H)}}\right)$, under suitable tuning of the learning rate (see Corollary A.22).

There is another immediate but important implication of (3): *any* no-regret algorithm in a (static) strongly convex-concave setting ought to be approaching the Nash equilibrium;[3] in contrast, this property is spectacularly false in (general) monotone settings [60].

**Proposition 3.6.** *Let $f : \mathcal{X} \times \mathcal{Y} \to \mathbb{R}$ be a $\mu$-strongly convex-concave function. If players incur regrets such that $\text{Reg}_{\mathcal{L},x}^{(T)} + \text{Reg}_{\mathcal{L},y}^{(T)} \leq CT^{1-\omega}$, for some parameters $C > 0$ and $\omega \in (0, 1]$, then for any $\epsilon > 0$ and $T > \left(\frac{2C}{\mu\epsilon^2}\right)^{1/\omega}$ there is a pair of strategies $\boldsymbol{z}^{(t)} \in \mathcal{Z}$ such that $\|\boldsymbol{z}^{(t)} - \boldsymbol{z}^\star\|_2 \leq \epsilon$, where $\boldsymbol{z}^\star \in \mathcal{Z}$ is a Nash equilibrium.*

The insights of this subsection are also of interest in general monotone settings by incorporating a strongly convex regularizer; tuning its magnitude allows us to trade off between a better approximation and the benefits of strong convexity-concavity revealed in this subsection.

### 3.3 General-sum multi-player games

Next, we turn our attention to general-sum multi-player games. For simplicity, in this subsection we posit that the game is represented in normal form, so that each player $i \in [\![n]\!]$ has a finite set of available actions $\mathcal{A}_i$, and $\mathcal{X}_i := \Delta(\mathcal{A}_i)$. The proofs of this subsection are included in Appendix A.3.

**Potential games**  First, we study the convergence of (online) gradient descent (GD) in time-varying *potential games* (see Definition A.23 for the formal description); we recall that unlike two-player zero-sum games, gradient descent is known to approach Nash equilibria in potential games. In our time-varying setup, it is assumed that each round $t \in [\![T]\!]$ corresponds to a different potential game described with a potential function $\Phi^{(t)}$. We further let $d : (\Phi, \Phi') \mapsto \max_{\boldsymbol{z} \in \times_{i=1}^n \mathcal{X}_i} (\Phi(\boldsymbol{z}) - \Phi'(\boldsymbol{z}))$, so that $\mathcal{V}_\Phi^{(T)} := \sum_{t=1}^{T-1} d(\Phi^{(t)}, \Phi^{(t+1)})$; we call attention to the fact that $d(\cdot, \cdot)$ is not symmetric. Analogously to Theorem 3.3, we use $\text{EQGAP}^{(t)}(\boldsymbol{z}^{(t)}) \in \mathbb{R}_{\geq 0}$ to represent the NE gap of the joint strategy profile $\boldsymbol{z}^{(t)} := (\boldsymbol{x}_1^{(t)}, \ldots, \boldsymbol{x}_n^{(t)})$ at the $t$-th game.

**Theorem 3.7.** *Suppose that each player employs (online) GD with a sufficiently small learning rate in a sequence of time-varying potential games. Then, $\sum_{t=1}^{T} \left(\text{EQGAP}^{(t)}(\boldsymbol{z}^{(t)})\right)^2 = O(\Phi_{max} + \mathcal{V}_\Phi^{(T)})$, where $\Phi_{max}$ is such that $|\Phi^{(t)}(\cdot)| \leq \Phi_{max}$.*

We refer to Appendix B for some illustrative experiments related to Theorem 3.7.

**General games**  Unfortunately, unlike the settings considered thus far, computing Nash equilibria in general games is computationally hard even under a crude approximation $\epsilon = \Theta(1)$ [25, 19, 30]. Instead, learning algorithms are known to converge—in a time-average sense—to relaxations of the Nash equilibrium, known as *(coarse) correlated equilibria*. For our purposes, we will employ a bilinear formulation of correlated equilibria, which dates back to the seminal work of Hart and Schmeidler [50] (see also [84, Chapter 12] for an excellent exposition). This will allow us to translate the results of Section 3.1 to general multi-player games.

Specifically, correlated equilibria[4] can be phrased via a game between the $n$ players and a *mediator*, an additional agent. At a high level, the mediator is endeavoring to identify a correlated strategy $\boldsymbol{\mu} \in \Xi := \Delta\left(\times_{i=1}^n \mathcal{A}_i\right)$ for which no player has an incentive to deviate from the recommendation. In contrast, the players are trying to optimally deviate so as to maximize their own utility. More precisely, there exist matrices $\mathbf{A}_1, \ldots, \mathbf{A}_n$, with each matrix $\mathbf{A}_i$ depending solely on the utility of Player $i$, for which the bilinear saddle-point problem can be expressed as

$$\min_{\boldsymbol{\mu} \in \Xi} \max_{(\bar{\boldsymbol{x}}_1, \ldots, \bar{\boldsymbol{x}}_n) \in \times_{i=1}^n \bar{\mathcal{X}}_i} \sum_{i=1}^{n} \boldsymbol{\mu}^\top \mathbf{A}_i \bar{\boldsymbol{x}}_i, \tag{4}$$

---

[3]Such observations were independently documented by Wang et al. [85].

[4]There is also a bilinear formulation tailored to coarse correlated equilibria, but we will focus solely on the stronger variant (CE) for concreteness.

where $\bar{\mathcal{X}}_i$ above is a suitable set of strategies; we elaborate more on this formulation in Appendix A.3. This zero-sum game has the property that there exists a strategy $\boldsymbol{\mu}^\star \in \Xi$ such that $\max_{\bar{\boldsymbol{x}}_i \in \bar{\mathcal{X}}_i} (\boldsymbol{\mu}^\star)^\top \mathbf{A}_i \bar{\boldsymbol{x}}_i \leq 0$, for any player $i \in [\![n]\!]$, which is precisely a correlated equilibrium.

Before we proceed, it is important to note that the learning paradigm considered here deviates from the traditional one in that orchestrating the protocol requires an additional learning agent, resulting in a less decentralized protocol. Yet, the dynamics induced by solving (4) via online algorithms remain *uncoupled* [49], in the sense that each player obtains feedback—corresponding to the deviation benefit—that depends solely on its own utility.

Now in the time-varying setting, the matrices $\mathbf{A}_1, \ldots, \mathbf{A}_n$ that capture the players' utilities can change in each repetition. Crucially, we show that the structure of the induced bilinear problem (4) is such that there is a sequence of correlated equilibria that guarantee nonnegative dynamic regret; this refines Property 3.1 in that only one player's strategies suffice to guarantee nonnegativity, even if the strategy of the other player remains invariant. Below, we denote by $\mathrm{DReg}_\mu^{(T)}$ the dynamic regret of Player min in (4), and by $\mathrm{Reg}_i^{(T)}$ the regret of each player $i \in [\![n]\!]$ up to time $T \in \mathbb{N}$, so that the regret of Player max in (4) can be expressed as $\sum_{i=1}^n \mathrm{Reg}_i^{(T)}$.

**Property 3.8.** *Suppose that $\Xi \ni \boldsymbol{\mu}^{(t,\star)}$ is a correlated equilibrium of the game at any time $t \in [\![T]\!]$. Then, $\mathrm{DReg}_\mu^{(T)}(\boldsymbol{\mu}^{(1,\star)}, \ldots, \boldsymbol{\mu}^{(T,\star)}) + \sum_{i=1}^n \mathrm{Reg}_i^{(T)} \geq 0$.*

As a result, this enables us to apply Theorem 3.3 parameterized on (i) the variation of the CE $\mathcal{V}_{\mathrm{CE}}^{(T)} := \inf_{\boldsymbol{\mu}^{(t,\star)} \in \Xi^{(t,\star)}, \forall t \in [\![T]\!]} \sum_{t=1}^{T-1} \|\boldsymbol{\mu}^{(t+1,\star)} - \boldsymbol{\mu}^{(t,\star)}\|_2$, where $\Xi^{(t,\star)}$ denotes the set of CE of the $t$-th game, and (ii) the variation in the players' utilities $\mathcal{V}_{\mathbf{A}}^{(T)} := \sum_{i=1}^n \sum_{t=1}^{T-1} \|\mathbf{A}_i^{(t+1)} - \mathbf{A}_i^{(t)}\|_2^2$; below, we denote by $\mathrm{CEGAP}^{(t)}(\boldsymbol{\mu}^{(t)})$ the CE gap of $\boldsymbol{\mu}^{(t)} \in \Xi$ at the $t$-th game.

**Theorem 3.9.** *Suppose that each player employs $\mathtt{OGD}$ in a sequence of time-varying BSPPs (4) with a sufficiently small learning rate. Then, $\sum_{t=1}^T \left( \mathrm{CEGAP}^{(t)}(\boldsymbol{\mu}^{(t)}) \right)^2 = O(1 + \mathcal{V}_{\mathrm{CE}}^{(T)} + \mathcal{V}_{\mathbf{A}}^{(T)})$.*

There are further interesting implications of our framework that are worth highlighting. First, we obtain meta-learning guarantees for general games that depend on the (algorithm-independent) similarity of the correlated equilibria (Corollary A.25); that was left as an open question by Harris et al. [48], where instead algorithm-dependent similarity metrics were derived. Further, by applying Corollary A.12, we derive natural variation-dependent per-player regret bounds in general games (Corollary A.26); this addresses a question left by Zhang et al. [93], albeit under a different learning paradigm. We suspect that obtaining such results—parameterized on the variation of the CE—are not possible without the presence of the additional agent, as in (4).

### 3.4 Dynamic regret bounds in static games

Finally, in this subsection we switch gears by investigating dynamic regret guarantees when learning in static games. The proofs of this subsection are included in Appendix A.4.

First, we point out that while traditional no-regret learning algorithms guarantee $O(\sqrt{T})$ *external* regret, instances of $\mathtt{OMD}$—a generalization of $\mathtt{OGD}$; see (5) in Appendix A—in fact guarantee $O(\sqrt{T})$ *dynamic* regret in two-player zero-sum games, which is a much stronger performance measure:

**Proposition 3.10.** *Suppose that both players in a (static) two-player zero-sum game employ $\mathtt{OMD}$ with a smooth regularizer. Then, $\mathrm{DReg}_x^{(T)}, \mathrm{DReg}_y^{(T)} = O(\sqrt{T})$.*

In proof, the dynamic regret of each player under $\mathtt{OMD}$ with a smooth regularizer can be bounded by the *first-order* path length of that player's strategies, which in turn can be bounded by $O(\sqrt{T})$ given that the second-order path length is $O(1)$ (Theorem A.7). In fact, Theorem A.7 readily extends Proposition 3.10 to time-varying zero-sum games as well, implying that

$$\mathrm{DReg}_x^{(T)}, \mathrm{DReg}_y^{(T)} = O\left( \sqrt{T(1 + \mathcal{V}_{\epsilon-\mathrm{NE}}^{(T)} + \mathcal{V}_{\mathbf{A}}^{(T)})} \right).$$

A question that arises from Proposition 3.10 is whether the $O(\sqrt{T})$ guarantee for dynamic regret of $\mathtt{OMD}$ can be improved in the online learning setting. Below, we point out a significant improvement to

$O(\log T)$, but under a stronger two-point feedback model;[5] namely, we posit that in every round each player can select an additional auxiliary strategy, and each player then gets to additionally observe the utility corresponding to the auxiliary strategies. Notably, this is akin to how the *extra-gradient method* works [52] (also *cf.* [72, Section 4.2] for multi-point feedback models in the bandit setting).

**Observation 3.11.** *Under two-point feedback, there exist learning algorithms that guarantee* $\mathrm{DReg}_x^{(T)}, \mathrm{DReg}_y^{(T)} = O(\log T)$ *in two-player zero-sum games.*

In particular, it suffices for each player to employ OMD, but with the twist that the first strategy in each round is the *time-average* of OMD; the auxiliary strategy is the standard output of OMD. Then, the dynamic regret of each player will grow as $O\left(\sum_{t=1}^{T} \frac{1}{t}\right) = O(\log T)$ since the duality gap of the average strategies is decreasing with a rate of $T^{-1}$ [72]. It is an interesting question whether the bound of Observation 3.11 can be improved to $O(1)$ [14]; we conjecture that there is a lower bound of $\Omega(\log T)$.

**General-sum games** In stark contrast, no (computationally efficient) sublinear dynamic regret guarantees are possible in general-sum games:

**Proposition 3.12.** *Any polynomial-time algorithm incurs* $\sum_{i=1}^{n} \mathrm{DReg}_i^{(T)} \geq CT$ *for any polynomial* $T \in \mathbb{N}$, *even if* $n = 2$ *and* $C > 0$ *is an absolute constant, unless ETH for* PPAD *is false [73].*

Indeed, this follows immediately since computing a Nash equilibrium to $O(1)$ precision in two-player games requires superpoylnomial time [73]. As such, Proposition 3.12 applies beyond the online learning setting. This motivates considering a relaxation of dynamic regret, wherein the sequence of comparators is subject to the constraint $\sum_{t=1}^{T-1} \mathbb{1}\{x_i^{(t+1,\star)} \neq x_i^{(t,\star)}\} \leq K - 1$, for some parameter $K \in \mathbb{N}$ [22]; this will be referred to as $K\text{-}\mathrm{DReg}_i^{(T)}$. Naturally, external regret coincides with $K\text{-}\mathrm{DReg}_i^{(T)}$ under $K = 1$. In this context, we employ Lemma A.1 to bound $K\text{-}\mathrm{DReg}^{(T)}$ under OGD:

**Theorem 3.13** (Precise version in Theorem A.28). *Suppose that all $n$ players employ OGD with a suitable learning rate in an L-smooth game. Then, for any $K \in \mathbb{N}$,*

1. $\sum_{i=1}^{n} K\text{-}\mathrm{DReg}_i^{(T)} = O(K\sqrt{n}LD_{\mathcal{Z}}^2)$;

2. $K\text{-}\mathrm{DReg}_i^{(T)} = O(K^{3/4}T^{1/4}n^{1/4}L^{1/2}D_{\mathcal{X}_i}^{3/2})$, *for any player $i \in [\![n]\!]$.*

One question that arises here is whether the per-player bound of $O(K^{3/4}T^{1/4})$ (Item 2) can be improved to $\tilde{O}(K)$, where $\tilde{O}(\cdot)$ hides logarithmic factors. The main challenge is that, even for $K = 1$, all known methods that obtain $\tilde{O}(1)$ [29, 68, 1, 38] rely on non-smooth regularizers that violate the preconditions of Lemma A.2—our dynamic RVU bound beyond (squared) Euclidean regularization; yet, we point out that it is possible under a slightly stronger feedback model (Remark A.29). We finally highlight that the game-theoretic significance of the solution concept that arises as the limit point of no-regret learners when $K\text{-}\mathrm{DReg} = o(T)$ was recently investigated by Crippa et al. [22].

## 4 Conclusions and future work

In this paper, we developed a framework for characterizing iterate-convergence of no-regret learning algorithms—primarily optimistic gradient descent (OGD)—in time-varying games. There are many promising avenues for future research. Besides closing the obvious gaps we highlighted in Section 3.4, it is important to characterize the behavior of no-regret learning algorithms in other fundamental time-varying multiagent settings, such as Stackelberg (security) games [5]. Moreover, our results operate in the full-feedback model where each player receives feedback on all possible actions. Extending the scope of our framework to capture partial-feedback models as well is another interesting direction for future work.

---

[5]The same asymptotic bound was independently established by Cai and Zheng [14] but in the standard feedback model.

## Acknowledgments and Disclosure of Funding

We are grateful to anonymous referees for providing valuable and constructive feedback. We also thank Vince Conitzer and Caspar Oesterheld for many helpful comments. This material is based on work supported by the National Science Foundation under grants IIS-1901403 and CCF-1733556 and by the ARO under award W911NF2210266.

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

# A  Omitted proofs

In this section, we provide the proofs from the main body (Section 3).

## A.1  Proofs from Section 3.1

First, we start with the proofs of the dynamic RVU bounds (Lemmas A.1 and A.2 below). Before we proceed, it will be useful to express the update rule (OGD) in the following equivalent form:

$$\boldsymbol{x}^{(t)} := \arg\max_{\boldsymbol{x} \in \mathcal{X}} \left\{ \Psi_x^{(t)}(\boldsymbol{x}) := \langle \boldsymbol{x}, \boldsymbol{m}_x^{(t)} \rangle - \frac{1}{\eta} \mathcal{B}_{\phi_x}(\boldsymbol{x} \parallel \hat{\boldsymbol{x}}^{(t)}) \right\},$$

$$\hat{\boldsymbol{x}}^{(t+1)} := \arg\max_{\hat{\boldsymbol{x}} \in \mathcal{X}} \left\{ \hat{\Psi}_x^{(t)}(\hat{\boldsymbol{x}}) := \langle \hat{\boldsymbol{x}}, \boldsymbol{u}_x^{(t)} \rangle - \frac{1}{\eta} \mathcal{B}_{\phi_x}(\hat{\boldsymbol{x}} \parallel \hat{\boldsymbol{x}}^{(t)}) \right\}. \tag{5}$$

Here, $\mathcal{B}_{\phi_x}(\cdot \parallel \cdot)$ denotes the *Bregman divergence* induced by the (squared) Euclidean regularizer $\phi_x : \boldsymbol{x} \mapsto \frac{1}{2}\|\boldsymbol{x}\|_2^2$; namely, $\mathcal{B}_{\phi_x}(\boldsymbol{x} \parallel \boldsymbol{x}') := \phi(\boldsymbol{x}) - \phi(\boldsymbol{x}') - \langle \nabla\phi(\boldsymbol{x}'), \boldsymbol{x} - \boldsymbol{x}' \rangle = \frac{1}{2}\|\boldsymbol{x} - \boldsymbol{x}'\|_2^2$, for $\boldsymbol{x}, \boldsymbol{x}' \in \mathcal{X}$. The update rule (5) for general Bregman divergences will be referred to as *optimistic mirror descent (OMD)*.

### A.1.1  Dynamic RVU bounds

The first key ingredient that we need for the proof of Theorem 3.3 is the property of *regret bounded by variation in utilities (RVU)*, in the sense of Syrgkanis et al. [78], but with respect to dynamic regret; such a bound is established below, and is analogous to that obtained by Zhang et al. [93].

**Lemma A.1** (RVU bound for dynamic regret). *Consider any sequence of utilities $(\boldsymbol{u}_x^{(1)}, \dots, \boldsymbol{u}_x^{(T)})$ up to time $T \in \mathbb{N}$. The dynamic regret (1) of OGD with respect to any sequence of comparators $(\boldsymbol{x}^{(1,\star)}, \dots, \boldsymbol{x}^{(T,\star)}) \in \mathcal{X}^T$ can be bounded by*

$$\frac{D_{\mathcal{X}}^2}{2\eta} + \frac{D_{\mathcal{X}}}{\eta} \sum_{t=1}^{T-1} \|\boldsymbol{x}^{(t+1,\star)} - \boldsymbol{x}^{(t,\star)}\|_2 + \eta \sum_{t=1}^{T} \|\boldsymbol{u}_x^{(t)} - \boldsymbol{m}_x^{(t)}\|_2^2$$

$$- \frac{1}{2\eta} \sum_{t=1}^{T} \left( \|\boldsymbol{x}^{(t)} - \hat{\boldsymbol{x}}^{(t)}\|_2^2 + \|\boldsymbol{x}^{(t)} - \hat{\boldsymbol{x}}^{(t+1)}\|_2^2 \right). \tag{6}$$

In the special case of external regret—$\boldsymbol{x}^{(1,\star)} = \boldsymbol{x}^{(2,\star)} = \cdots = \boldsymbol{x}^{(T,\star)}$—(6) recovers the bound for OGD of Syrgkanis et al. [78]. The key takeaway from Lemma A.1 is that the overhead of dynamic regret in (6) grows with the *first-order* variation of the sequence of comparators.

*Proof of Lemma A.1.* First, by $(1/\eta)$-strong convexity of the function $\Psi_x^{(t)}$ (defined in (5)) for any time $t \in [\![T]\!]$, we have that

$$\langle \boldsymbol{x}^{(t)}, \boldsymbol{m}_x^{(t)} \rangle - \frac{1}{2\eta}\|\boldsymbol{x}^{(t)} - \hat{\boldsymbol{x}}^{(t)}\|_2^2 - \langle \hat{\boldsymbol{x}}^{(t+1)}, \boldsymbol{m}_x^{(t)} \rangle + \frac{1}{2\eta}\|\hat{\boldsymbol{x}}^{(t+1)} - \hat{\boldsymbol{x}}^{(t)}\|_2^2 \geq \frac{1}{2\eta}\|\boldsymbol{x}^{(t)} - \hat{\boldsymbol{x}}^{(t+1)}\|_2^2, \tag{7}$$

where we used [75, Lemma 2.8; p. 135]. Similarly, by $(1/\eta)$-strong convexity of the function $\hat{\Psi}_x^{(t)}$ (defined in (5)) for any time $t \in [\![T]\!]$, we have that for any comparator $\boldsymbol{x}^{(t,\star)} \in \mathcal{X}$,

$$\langle \hat{\boldsymbol{x}}^{(t+1)}, \boldsymbol{u}_x^{(t)} \rangle - \frac{1}{2\eta}\|\hat{\boldsymbol{x}}^{(t+1)} - \hat{\boldsymbol{x}}^{(t)}\|_2^2 - \langle \boldsymbol{x}^{(t,\star)}, \boldsymbol{u}_x^{(t)} \rangle + \frac{1}{2\eta}\|\boldsymbol{x}^{(t,\star)} - \hat{\boldsymbol{x}}^{(t)}\|_2^2$$

$$\geq \frac{1}{2\eta}\|\hat{\boldsymbol{x}}^{(t+1)} - \boldsymbol{x}^{(t,\star)}\|_2^2. \tag{8}$$

Thus, adding (7) and (8),

$$\langle \boldsymbol{x}^{(t,\star)} - \hat{\boldsymbol{x}}^{(t+1)}, \boldsymbol{u}_x^{(t)} \rangle + \langle \hat{\boldsymbol{x}}^{(t+1)} - \boldsymbol{x}^{(t)}, \boldsymbol{m}_x^{(t)} \rangle \leq \frac{1}{2\eta} \left( \|\hat{\boldsymbol{x}}^{(t)} - \boldsymbol{x}^{(t,\star)}\|_2^2 - \|\hat{\boldsymbol{x}}^{(t+1)} - \boldsymbol{x}^{(t,\star)}\|_2^2 \right)$$

$$- \frac{1}{2\eta} \left( \|\boldsymbol{x}^{(t)} - \hat{\boldsymbol{x}}^{(t)}\|_2^2 + \|\boldsymbol{x}^{(t)} - \hat{\boldsymbol{x}}^{(t+1)}\|_2^2 \right). \tag{9}$$

We further see that

$$\langle \boldsymbol{x}^{(t,\star)} - \boldsymbol{x}^{(t)}, \boldsymbol{u}_x^{(t)} \rangle = \langle \boldsymbol{x}^{(t)} - \hat{\boldsymbol{x}}^{(t+1)}, \boldsymbol{m}_x^{(t)} - \boldsymbol{u}_x^{(t)} \rangle + \langle \boldsymbol{x}^{(t,\star)} - \hat{\boldsymbol{x}}^{(t+1)}, \boldsymbol{u}_x^{(t)} \rangle$$
$$+ \langle \hat{\boldsymbol{x}}^{(t+1)} - \boldsymbol{x}^{(t)}, \boldsymbol{m}_x^{(t)} \rangle. \qquad (10)$$

Now the first term on the right-hand side can be upper bounded using the fact that, by (7) and (8),

$$\langle \boldsymbol{x}^{(t)} - \hat{\boldsymbol{x}}^{(t+1)}, \boldsymbol{m}_x^{(t)} - \boldsymbol{u}_x^{(t)} \rangle \geq \frac{1}{\eta} \|\hat{\boldsymbol{x}}^{(t+1)} - \boldsymbol{x}^{(t)}\|_2^2 \implies \|\hat{\boldsymbol{x}}^{(t+1)} - \boldsymbol{x}^{(t)}\|_2 \leq \eta \|\boldsymbol{m}_x^{(t)} - \boldsymbol{u}_x^{(t)}\|_2,$$

by Cauchy-Schwarz, in turn implying that $\langle \boldsymbol{x}^{(t)} - \hat{\boldsymbol{x}}^{(t+1)}, \boldsymbol{m}_x^{(t)} - \boldsymbol{u}_x^{(t)} \rangle \leq \eta \|\boldsymbol{m}_x^{(t)} - \boldsymbol{u}_x^{(t)}\|_2^2$. Thus, the proof follows by combining this bound with (9) and (10), along with the fact that

$$\sum_{t=1}^{T} \left( \|\hat{\boldsymbol{x}}^{(t)} - \boldsymbol{x}^{(t,\star)}\|_2^2 - \|\hat{\boldsymbol{x}}^{(t+1)} - \boldsymbol{x}^{(t,\star)}\|_2^2 \right) \leq \|\hat{\boldsymbol{x}}^{(1)} - \boldsymbol{x}^{(1,\star)}\|_2^2$$
$$+ \sum_{t=1}^{T-1} \left( \|\hat{\boldsymbol{x}}^{(t+1)} - \boldsymbol{x}^{(t+1,\star)}\|_2^2 - \|\hat{\boldsymbol{x}}^{(t+1)} - \boldsymbol{x}^{(t,\star)}\|_2^2 \right)$$
$$\leq D_{\mathcal{X}}^2 + 2D_{\mathcal{X}} \sum_{t=1}^{T-1} \|\boldsymbol{x}^{(t+1,\star)} - \boldsymbol{x}^{(t,\star)}\|_2,$$

where the last bound follows since

$$\|\hat{\boldsymbol{x}}^{(t+1)} - \boldsymbol{x}^{(t+1,\star)}\|_2^2 - \|\hat{\boldsymbol{x}}^{(t+1)} - \boldsymbol{x}^{(t,\star)}\|_2^2 \leq 2D_{\mathcal{X}} \left| \|\hat{\boldsymbol{x}}^{(t+1)} - \boldsymbol{x}^{(t+1,\star)}\|_2 - \|\hat{\boldsymbol{x}}^{(t+1)} - \boldsymbol{x}^{(t,\star)}\|_2 \right|$$
$$\leq 2D_{\mathcal{X}} \|\boldsymbol{x}^{(t+1,\star)} - \boldsymbol{x}^{(t,\star)}\|_2,$$

where we recall that $D_{\mathcal{X}}$ denotes the $\ell_2$-diameter of $\mathcal{X}$. $\qquad \square$

As an aside, we remark that assuming that $\boldsymbol{m}_x^{(t)} := \boldsymbol{0}$ and $\|\boldsymbol{u}_x^{(t)}\|_2 \leq 1$ for any $t \in [\![T]\!]$, Lemma A.1 implies that dynamic regret can be upper bounded by $O\left( \sqrt{(1 + \sum_{t=1}^{T-1} \|\boldsymbol{x}^{(t+1,\star)} - \boldsymbol{x}^{(t,\star)}\|_2)T} \right)$, for *any* (bounded)—potentially adversarially selected—sequence of utilities $(\boldsymbol{u}_x^{(1)}, \ldots, \boldsymbol{u}_x^{(T)})$, for $\eta := \sqrt{\frac{D_{\mathcal{X}}^2}{2T} + \frac{D_{\mathcal{X}} \sum_{t=1}^{T-1} \|\boldsymbol{x}^{(t+1,\star)} - \boldsymbol{x}^{(t,\star)}\|_2}{T}}$, which is a well-known result in online optimization [95]; while that requires setting the learning rate based on the first-order variation of the (optimal) comparators, there are standard techniques that would allow bypassing that assumption.

Next, we provide an extension of Lemma A.1 to the more general OMD algorithm under a broad class of regularizers.

**Lemma A.2** (Extension of Lemma A.1 beyond Euclidean regularization). *Consider a 1-strongly convex continuously differentiable regularizer $\phi$ with respect to a norm $\|\cdot\|$ such that (i) $\|\nabla\phi(\boldsymbol{x})\|_* \leq G$ for any $\boldsymbol{x}$, and (ii) $\mathcal{B}_{\phi_x}(\boldsymbol{x} \parallel \boldsymbol{x}') \leq L\|\boldsymbol{x} - \boldsymbol{x}'\|$ for any $\boldsymbol{x}, \boldsymbol{x}'$. Then, for any sequence of utilities $(\boldsymbol{u}_x^{(1)}, \ldots, \boldsymbol{u}_x^{(T)})$ up to time $T \in \mathbb{N}$ the dynamic regret (1) of OMD with respect to any sequence of comparators $(\boldsymbol{x}^{(1,\star)}, \ldots, \boldsymbol{x}^{(T,\star)}) \in \mathcal{X}^T$ can be bounded as*

$$\frac{\mathcal{B}_{\phi_x}(\boldsymbol{x}^{(1,\star)} \parallel \hat{\boldsymbol{x}}^{(1)})}{\eta} + \frac{L + 2G}{\eta} \sum_{t=1}^{T-1} \|\boldsymbol{x}^{(t+1,\star)} - \boldsymbol{x}^{(t,\star)}\| + \eta \sum_{t=1}^{T} \|\boldsymbol{u}_x^{(t)} - \boldsymbol{m}_x^{(t)}\|_*^2$$
$$- \frac{1}{2\eta} \sum_{t=1}^{T} \left( \|\boldsymbol{x}^{(t)} - \hat{\boldsymbol{x}}^{(t)}\|^2 + \|\boldsymbol{x}^{(t)} - \hat{\boldsymbol{x}}^{(t+1)}\|^2 \right).$$

The proof is analogous to that of Lemma A.1, and relies on the well-known three-point identity for the Bregman divergence:

$$\mathcal{B}_{\phi_x}(\boldsymbol{x} \parallel \boldsymbol{x}') = \mathcal{B}_{\phi_x}(\boldsymbol{x} \parallel \boldsymbol{x}'') + \mathcal{B}_{\phi_x}(\boldsymbol{x}'' \parallel \boldsymbol{x}') - \langle \boldsymbol{x} - \boldsymbol{x}'', \nabla\phi(\boldsymbol{x}') - \nabla\phi(\boldsymbol{x}'') \rangle. \qquad (11)$$

In particular, along with the assumptions of Lemma A.2 imposed on the regularizer $\phi_x$, (11) implies that the term $\sum_{t=1}^{T-1} \left( \mathcal{B}_{\phi_x}(\boldsymbol{x}^{(t+1,\star)} \| \hat{\boldsymbol{x}}^{(t+1)}) - \mathcal{B}_{\phi_x}(\boldsymbol{x}^{(t,\star)} \| \hat{\boldsymbol{x}}^{(t+1)}) \right)$ is equal to

$$\sum_{t=1}^{T-1} \left( \mathcal{B}_{\phi_x}(\boldsymbol{x}^{(t+1,\star)} \| \boldsymbol{x}^{(t,\star)}) - \langle \boldsymbol{x}^{(t+1,\star)} - \boldsymbol{x}^{(t,\star)}, \nabla\phi(\hat{\boldsymbol{x}}^{(t+1)}) - \nabla\phi(\boldsymbol{x}^{(t,\star)}) \rangle \right)$$

$$\leq (L + 2G) \sum_{t=1}^{T-1} \| \boldsymbol{x}^{(t+1,\star)} - \boldsymbol{x}^{(t,\star)} \|,$$

since $\mathcal{B}_{\phi_x}(\boldsymbol{x}^{(t+1,\star)} \| \boldsymbol{x}^{(t,\star)}) \leq L\|\boldsymbol{x}^{(t+1,\star)} - \boldsymbol{x}^{(t,\star)}\|$ (by assumption) and

$$\langle \boldsymbol{x}^{(t+1,\star)} - \boldsymbol{x}^{(t,\star)}, \nabla\phi(\hat{\boldsymbol{x}}^{(t+1)}) - \nabla\phi(\boldsymbol{x}^{(t,\star)}) \rangle$$

$$\leq \| \boldsymbol{x}^{(t+1,\star)} - \boldsymbol{x}^{(t,\star)} \| \| \nabla\phi(\hat{\boldsymbol{x}}^{(t+1)}) - \nabla\phi(\boldsymbol{x}^{(t,\star)}) \|_* \quad (12)$$

$$\leq \left( \| \nabla\phi(\hat{\boldsymbol{x}}^{(t+1)}) \|_* + \| \nabla\phi(\boldsymbol{x}^{(t,\star)}) \|_* \right) \| \boldsymbol{x}^{(t+1,\star)} - \boldsymbol{x}^{(t,\star)} \| \quad (13)$$

$$\leq 2G \| \boldsymbol{x}^{(t+1,\star)} - \boldsymbol{x}^{(t,\star)} \|, \quad (14)$$

where (12) follows from the Cauchy-Schwarz inequality; (13) uses the triangle inequality for the dual norm $\| \cdot \|_*$; and (14) follows from the assumption of Lemma A.2 that $\|\nabla\phi(\cdot)\|_* \leq G$. The rest of the proof of Lemma A.2 is analogous to Lemma A.1, and it is therefore omitted. One important question here is whether Lemma A.2 can be extended under a broader class of regularizers; as we explain in Section 3.4, this is related to improving Theorem 3.13.

### A.1.2 Nonnegativity of dynamic regret

We next proceed with the proof of Property 3.1. For completeness, we first prove the following special case [93]; the proof of Property 3.1 is then analogous.

**Property A.3** (Special case of Property 3.1). *Suppose that* $\mathcal{Z} \ni \boldsymbol{z}^{(t,\star)} = (\boldsymbol{x}^{(t,\star)}, \boldsymbol{y}^{(t,\star)})$ *is a Nash equilibrium of the $t$-th game, for any time* $t \in [\![T]\!]$. *Then, for* $s_x^{(T)} = (\boldsymbol{x}^{(t,\star)})_{1 \leq t \leq T}$ *and* $s_y^{(T)} = (\boldsymbol{y}^{(t,\star)})_{1 \leq t \leq T}$,

$$\mathrm{DReg}_x^{(T)}(s_x^{(T)}) + \mathrm{DReg}_y^{(T)}(s_y^{(T)}) \geq 0.$$

*Proof.* Let $v^{(t)} := \langle \boldsymbol{x}^{(t,\star)}, \mathbf{A}^{(t)}\boldsymbol{y}^{(t,\star)} \rangle$ be the value of the $t$-th game, for some $t \in [\![T]\!]$. Then, we have that $v^{(t)} = \langle \boldsymbol{x}^{(t,\star)}, \mathbf{A}^{(t)}\boldsymbol{y}^{(t,\star)} \rangle \leq \langle \boldsymbol{x}, \mathbf{A}^{(t)}\boldsymbol{y}^{(t,\star)} \rangle$ for any $\boldsymbol{x} \in \mathcal{X}$, since $\boldsymbol{x}^{(t,\star)}$ is a best response to $\boldsymbol{y}^{(t,\star)}$; similarly, $v^{(t)} = \langle \boldsymbol{x}^{(t,\star)}, \mathbf{A}^{(t)}\boldsymbol{y}^{(t,\star)} \rangle \geq \langle \boldsymbol{x}^{(t,\star)}, \mathbf{A}^{(t)}\boldsymbol{y} \rangle$ for any $\boldsymbol{y} \in \mathcal{Y}$. Hence, $\langle \boldsymbol{x}^{(t)}, \mathbf{A}^{(t)}\boldsymbol{y}^{(t,\star)} \rangle - \langle \boldsymbol{x}^{(t,\star)}, \mathbf{A}^{(t)}\boldsymbol{y}^{(t)} \rangle \geq 0$, or equivalently, $\langle \boldsymbol{x}^{(t,\star)}, \boldsymbol{u}_x^{(t)} \rangle + \langle \boldsymbol{y}^{(t,\star)}, \boldsymbol{u}_y^{(t)} \rangle \geq 0$. But given that the game is zero-sum, it holds that $\langle \boldsymbol{x}^{(t)}, \boldsymbol{u}_x^{(t)} \rangle + \langle \boldsymbol{y}^{(t)}, \boldsymbol{u}_y^{(t)} \rangle = 0$, so the last inequality can be in turn cast as

$$\langle \boldsymbol{x}^{(t,\star)}, \boldsymbol{u}_x^{(t)} \rangle - \langle \boldsymbol{x}^{(t)}, \boldsymbol{u}_x^{(t)} \rangle + \langle \boldsymbol{y}^{(t,\star)}, \boldsymbol{u}_y^{(t)} \rangle - \langle \boldsymbol{y}^{(t)}, \boldsymbol{u}_y^{(t)} \rangle \geq 0,$$

for any $t \in [\![T]\!]$. As a result, summing over all $t \in [\![T]\!]$ we have shown that

$$\mathrm{DReg}_x^{(T)}(\boldsymbol{x}^{(1,\star)}, \ldots, \boldsymbol{x}^{(T,\star)}) + \mathrm{DReg}_y^{(T)}(\boldsymbol{y}^{(1,\star)}, \ldots, \boldsymbol{y}^{(T,\star)})$$

$$= \sum_{t=1}^{T} \langle \boldsymbol{x}^{(t,\star)}, \boldsymbol{u}_x^{(t)} \rangle - \langle \boldsymbol{x}^{(t)}, \boldsymbol{u}_x^{(t)} \rangle + \langle \boldsymbol{y}^{(t,\star)}, \boldsymbol{u}_y^{(t)} \rangle - \langle \boldsymbol{y}^{(t)}, \boldsymbol{u}_y^{(t)} \rangle \geq 0.$$

$\square$

**Property 3.1.** *Suppose that* $\mathcal{Z} \ni \boldsymbol{z}^{(t,\star)} = (\boldsymbol{x}^{(t,\star)}, \boldsymbol{y}^{(t,\star)})$ *is an* $\epsilon^{(t)}$-*approximate Nash equilibrium of the $t$-th game. Then, for* $s_x^{(T)} = (\boldsymbol{x}^{(t,\star)})_{1 \leq t \leq T}$ *and* $s_y^{(T)} = (\boldsymbol{y}^{(t,\star)})_{1 \leq t \leq T}$,

$$\mathrm{DReg}_x^{(T)}(s_x^{(T)}) + \mathrm{DReg}_y^{(T)}(s_y^{(T)}) \geq -2\sum_{t=1}^{T} \epsilon^{(t)}.$$

*Proof.* Given that $(\boldsymbol{x}^{(t,\star)}, \boldsymbol{y}^{(t,\star)}) \in \mathcal{Z}$ is an $\epsilon^{(t)}$-approximate Nash equilibrium of the $t$-th game, it follows that $\langle \boldsymbol{x}^{(t,\star)}, \mathbf{A}^{(t)}\boldsymbol{y}^{(t,\star)} \rangle \leq \langle \boldsymbol{x}^{(t)}, \mathbf{A}^{(t)}\boldsymbol{y}^{(t,\star)} \rangle + \epsilon_x^{(t)}$ and $\langle \boldsymbol{x}^{(t,\star)}, \mathbf{A}^{(t)}\boldsymbol{y}^{(t,\star)} \rangle \geq \langle \boldsymbol{x}^{(t,\star)}, \mathbf{A}^{(t)}\boldsymbol{y}^{(t)} \rangle - \epsilon_y^{(t)}$, for some $\epsilon_x^{(t)}, \epsilon_y^{(t)} \leq \epsilon^{(t)}$. Thus, we have that $\langle \boldsymbol{x}^{(t)}, \mathbf{A}^{(t)}\boldsymbol{y}^{(t,\star)} \rangle \geq \langle \boldsymbol{x}^{(t,\star)}, \mathbf{A}^{(t)}\boldsymbol{y}^{(t)} \rangle - \epsilon_x^{(t)} - \epsilon_y^{(t)}$, or equivalently, $\langle \boldsymbol{x}^{(t,\star)}, \boldsymbol{u}_x^{(t)} \rangle + \langle \boldsymbol{y}^{(t,\star)}, \boldsymbol{u}_y^{(t)} \rangle \geq -\epsilon_x^{(t)} - \epsilon_y^{(t)} \geq -2\epsilon^{(t)}$. As a result,

$$\langle \boldsymbol{x}^{(t,\star)}, \boldsymbol{u}_x^{(t)} \rangle - \langle \boldsymbol{x}^{(t)}, \boldsymbol{u}_x^{(t)} \rangle + \langle \boldsymbol{y}^{(t,\star)}, \boldsymbol{u}_y^{(t)} \rangle - \langle \boldsymbol{y}^{(t)}, \boldsymbol{u}_y^{(t)} \rangle \geq -2\epsilon^{(t)}, \tag{15}$$

for any $t \in [\![T]\!]$, and the statement follows by summing (15) over all $t \in [\![T]\!]$. $\qquad\square$

In fact, as we show below (in Property A.4), Property A.3 is a more general consequence of the minimax theorem. In particular, for a nonlinear online learning problem, we define dynamic regret with respect to a sequence of comparators $(\boldsymbol{x}^{(1,\star)}, \ldots, \boldsymbol{x}^{(T,\star)}) \in \mathcal{X}^T$ as follows:

$$\mathrm{DReg}_x^{(T)}(\boldsymbol{x}^{(1,\star)}, \ldots, \boldsymbol{x}^{(T,\star)}) := \sum_{t=1}^{T} \left( u_x^{(t)}(\boldsymbol{x}^{(t,\star)}) - u_x^{(t)}(\boldsymbol{x}^{(t)}) \right), \tag{16}$$

where $u_x^{(1)}, \ldots, u_x^{(T)} : \boldsymbol{x} \mapsto \mathbb{R}$ are the continuous utility functions observed by the learner, which could be in general nonconcave, and $(\boldsymbol{x}^{(t)})_{1 \leq t \leq T}$ is the sequence of strategies produced by the learner; (16) generalizes the notion of dynamic regret (1) in online linear optimization, that is, when $u_x^{(t)} : \boldsymbol{x} \mapsto \langle \boldsymbol{x}, \boldsymbol{u}_x^{(t)} \rangle$, where $\boldsymbol{u}_x^{(t)} \in \mathbb{R}^{d_x}$ for any time $t \in [\![T]\!]$.

**Property A.4.** *Suppose that $f^{(t)} : \mathcal{X} \times \mathcal{Y} \to \mathbb{R}$ is a continuous function such that for any $t \in [\![T]\!]$,*

$$\min_{\boldsymbol{x} \in \mathcal{X}} \max_{\boldsymbol{y} \in \mathcal{Y}} f^{(t)}(\boldsymbol{x}, \boldsymbol{y}) = \max_{\boldsymbol{y} \in \mathcal{Y}} \min_{\boldsymbol{x} \in \mathcal{X}} f^{(t)}(\boldsymbol{x}, \boldsymbol{y}).$$

*Let also $\boldsymbol{x}^{(t,\star)} \in \arg\min_{\boldsymbol{x} \in \mathcal{X}} \max_{\boldsymbol{y} \in \mathcal{Y}} f^{(t)}(\boldsymbol{x}, \boldsymbol{y})$ and $\boldsymbol{y}^{(t,\star)} \in \arg\max_{\boldsymbol{y} \in \mathcal{Y}} \min_{\boldsymbol{x} \in \mathcal{X}} f^{(t)}(\boldsymbol{x}, \boldsymbol{y})$, for any $t \in [\![T]\!]$. Then, for $s_x^{(T)} = (\boldsymbol{x}^{(t,\star)})_{1 \leq t \leq T}$ and $s_y^{(T)} = (\boldsymbol{y}^{(t,\star)})_{1 \leq t \leq T}$,*

$$\mathrm{DReg}_x^{(T)}(s_x^{(T)}) + \mathrm{DReg}_y^{(T)}(s_y^{(T)}) \geq 0.$$

*Proof.* By definition of dynamic regret (16), it suffices to show that $f^{(t)}(\boldsymbol{x}^{(t)}, \boldsymbol{y}^{(t,\star)}) \geq f^{(t)}(\boldsymbol{x}^{(t,\star)}, \boldsymbol{y}^{(t)})$, for any time $t \in [\![T]\!]$. Indeed,

$$f^{(t)}(\boldsymbol{x}^{(t)}, \boldsymbol{y}^{(t,\star)}) \geq \min_{\boldsymbol{x} \in \mathcal{X}} f^{(t)}(\boldsymbol{x}, \boldsymbol{y}^{(t,\star)}) \tag{17}$$

$$= \max_{\boldsymbol{y} \in \mathcal{Y}} \min_{\boldsymbol{x} \in \mathcal{X}} f^{(t)}(\boldsymbol{x}, \boldsymbol{y}) \tag{18}$$

$$= \min_{\boldsymbol{x} \in \mathcal{X}} \max_{\boldsymbol{y} \in \mathcal{Y}} f^{(t)}(\boldsymbol{x}, \boldsymbol{y}) \tag{19}$$

$$= \max_{\boldsymbol{y} \in \mathcal{Y}} f^{(t)}(\boldsymbol{x}^{(t,\star)}, \boldsymbol{y}) \tag{20}$$

$$\geq f^{(t)}(\boldsymbol{x}^{(t,\star)}, \boldsymbol{y}^{(t)}), \tag{21}$$

where (17) and (21) are obvious; (18) and (20) follow from the definition of $\boldsymbol{y}^{(t,\star)} \in \mathcal{Y}$ and $\boldsymbol{x}^{(t,\star)} \in \mathcal{X}$, respectively; and (19) holds by assumption. This concludes the proof. $\qquad\square$

### A.1.3 Variation of the Nash equilibria

In our next example, we point out the standard observation that an arbitrarily small change in the entries of the payoff matrix can lead to a substantial deviation in the Nash equilibrium.

*Example* A.5. Consider a $2 \times 2$ (two-player) zero-sum game, where $\mathcal{X} := \Delta^2, \mathcal{Y} := \Delta^2$, described by the payoff matrix

$$\mathbf{A} := \begin{bmatrix} 2\delta & 0 \\ 0 & \delta \end{bmatrix}, \tag{22}$$

for some $\delta > 0$. Then, it is easy to see that the unique Nash equilibrium of this game is such that $\boldsymbol{x}^\star, \boldsymbol{y}^\star := (\frac{1}{3}, \frac{2}{3}) \in \Delta^2$. Suppose now that the original payoff matrix (22) is perturbed to a new matrix

$$\mathbf{A}' := \begin{bmatrix} \delta & 0 \\ 0 & 2\delta \end{bmatrix}. \tag{23}$$

The new (unique) Nash equilibrium now reads $\boldsymbol{x}^\star, \boldsymbol{y}^\star := (\frac{2}{3}, \frac{1}{3}) \in \Delta^2$. We conclude that an arbitrarily small deviation in the entries of the payoff matrix can lead to a non-trivial change in the Nash equilibrium.

Next, we leverage the simple observation of the example above to establish Proposition 3.2, the statement of which is recalled below.

**Proposition 3.2.** *For any $T \geq 4$, there is a sequence of $T$ games such that $\mathcal{V}_{NE}^{(T)} \geq \frac{T}{2}$ while $\mathcal{V}_{\epsilon-NE}^{(T)} \leq \delta$, for any $\delta > 0$.*

*Proof.* We consider a sequence of $T \geq 4$ games such that $\mathcal{X}, \mathcal{Y} := \Delta^2$ and

$$\mathbf{A}^{(t)} = \begin{cases} \mathbf{A} & \text{if } t \mod 2 = 1, \\ \mathbf{A}' & \text{if } t \mod 2 = 0, \end{cases}$$

where $\mathbf{A}, \mathbf{A}'$ are the payoff matrices defined in (22) and (23), and are parameterized by $\delta > 0$ (Example A.5). Then, the exact Nash equilibria read

$$\boldsymbol{x}^{(t,\star)}, \boldsymbol{y}^{(t,\star)} = \begin{cases} (\frac{1}{3}, \frac{2}{3}) & \text{if } t \mod 2 = 1, \\ (\frac{2}{3}, \frac{1}{3}) & \text{if } t \mod 2 = 0. \end{cases}$$

As a result, it follows that $\mathcal{V}_{NE}^{(T)} := \sum_{t=1}^{T-1} \|\boldsymbol{z}^{(t+1,\star)} - \boldsymbol{z}^{(t,\star)}\|_2 = \frac{2}{3}(T-1) \geq \frac{T}{2}$, for $T \geq 4$. In contrast, it is clear that $\mathcal{V}_{\epsilon-NE}^{(T)} \leq C\delta T$, which follows by simply considering the sequence of strategies wherein both players are always selecting actions uniformly at random; we recall that $C > 0$ here is the value that parameterizes $\mathcal{V}_{\epsilon-NE}^{(T)}$. Thus, taking $\delta := \frac{\delta'}{CT}$, for some arbitrarily small $\delta' > 0$, concludes the proof. $\qquad\square$

In the above sequence of games, the variation measure $\mathcal{W}_{\mathbf{A}}^{(T)}$ is in fact also arbitrarily small as $\delta \to 0$; this reflects the fact that, although the exact Nash equilibria change abruptly, the sequence of payoff matrices exhibits small variation. Nevertheless, we next point out a modified sequence of games where $\mathcal{V}_{\epsilon-NE}$ remains small even though both $\mathcal{W}_{\mathbf{A}}^{(T)}$ and $\mathcal{V}_{NE}^{(T)}$ can be unbounded.

**Proposition A.6.** *For any $T \geq 4$, there is a sequence of games such that $\mathcal{V}_{NE}^{(T)}, \mathcal{W}_{\mathbf{A}}^{(T)} = \Omega(T)$ while $\mathcal{V}_{\epsilon-NE}^{(T)} \leq \delta$, for any $\delta > 0$.*

*Proof.* We consider a sequence of $T \geq 4$ games such that $\mathcal{X}, \mathcal{Y} := \Delta^2$ and

$$\mathbf{A}^{(t)} = \begin{cases} \mathbf{A} + \mathbf{1}_{2\times 2} & \text{if } t \mod 2 = 1, \\ \mathbf{A}' & \text{if } t \mod 2 = 0, \end{cases}$$

where $\mathbf{A}, \mathbf{A}'$ are the payoff matrices defined in (22) and (23), and $\mathbf{1}_{2\times 2}$ denotes the all-ones $2 \times 2$ matrix. Given that $\boldsymbol{x}^\top(\mathbf{A} + \mathbf{1}_{2\times 2})\boldsymbol{y} = \boldsymbol{x}^\top\mathbf{A}\boldsymbol{y} + 1$, for any $\boldsymbol{x}, \boldsymbol{y} \in \Delta^2$, it follows that the set of $\epsilon$-approximate Nash equilibria of each game coincides with the set of $\epsilon$-approximate Nash equilibria of the corresponding game in Proposition 3.2. Thus, by the same argument as in Proposition 3.2 earlier, we have that $\mathcal{V}_{NE}^{(T)} \geq \frac{T}{2}$ and $\mathcal{V}_{\epsilon-NE}^{(T)} \leq C\delta T$. It is also clear that $\mathcal{W}_{\mathbf{A}}^{(T)} = \Omega(T)$, concluding the proof. $\qquad\square$

In words, the phenomenon identified above occurs because $\mathcal{W}_{\mathbf{A}}^{(T)}$ is affected by perturbations in the payoff matrices that do not strategically alter the game—namely, the addition of the matrix $\mathbf{1}_{2\times 2}$.

### A.1.4   Proof of Theorem 3.3

Next, we proceed with the proof of one of our main results, namely Theorem 3.3. The key ingredient is Theorem A.7 below, which bounds the second-order path length of OGD in terms of the considered variation measures.

**Theorem A.7.** *Suppose that both players employ OGD with learning rate $\eta \leq \frac{1}{4L}$ in a time-varying bilinear saddle-point problem, where $L := \max_{1 \leq t \leq T} \|\mathbf{A}^{(t)}\|_2$. Then, for any time horizon $T \in \mathbb{N}$,*

$$\sum_{t=1}^{T} \left( \|\boldsymbol{z}^{(t)} - \hat{\boldsymbol{z}}^{(t)}\|_2^2 + \|\boldsymbol{z}^{(t)} - \hat{\boldsymbol{z}}^{(t+1)}\|_2^2 \right) \leq 2D_{\mathcal{Z}}^2 + 4\eta^2 L^2 \|\mathcal{Z}\|_2^2 + 4D_{\mathcal{Z}}\mathcal{V}_{\epsilon-NE}^{(T)} + 8\eta^2 \|\mathcal{Z}\|_2^2 \mathcal{V}_{\mathbf{A}}^{(T)}.$$

*Proof of Theorem A.7.* First, for any $t \geq 2$ we have that $\|\boldsymbol{u}_x^{(t)} - \boldsymbol{m}_x^{(t)}\|_2^2$ is equal to

$$\|\mathbf{A}^{(t)}\boldsymbol{y}^{(t)} - \mathbf{A}^{(t-1)}\boldsymbol{y}^{(t-1)}\|_2^2 \leq 2\|\mathbf{A}^{(t)}(\boldsymbol{y}^{(t)} - \boldsymbol{y}^{(t-1)})\|_2^2 + 2\|(\mathbf{A}^{(t)} - \mathbf{A}^{(t-1)})\boldsymbol{y}^{(t-1)}\|_2^2 \quad (24)$$

$$\leq 2\|\mathbf{A}^{(t)}\|_2^2\|\boldsymbol{y}^{(t)} - \boldsymbol{y}^{(t-1)}\|_2^2 + 2\|\mathbf{A}^{(t)} - \mathbf{A}^{(t-1)}\|_2^2\|\boldsymbol{y}^{(t-1)}\|_2^2 \quad (25)$$

$$\leq 2L^2\|\boldsymbol{y}^{(t)} - \boldsymbol{y}^{(t-1)}\|_2^2 + 2\|\mathcal{Y}\|_2^2\|\mathbf{A}^{(t)} - \mathbf{A}^{(t-1)}\|_2^2, \quad (26)$$

where (24) uses the triangle inequality for the norm $\|\cdot\|_2$ along with the inequality $2ab \leq a^2 + b^2$ for any $a, b \in \mathbb{R}$; (25) follows from the definition of the operator norm; and (26) uses the assumption that $\|\mathbf{A}^{(t)}\|_2 \leq L$ and $\|\boldsymbol{y}\|_2 \leq \|\mathcal{Y}\|_2$ for any $\boldsymbol{y} \in \mathcal{Y}$. A similar derivaiton shows that for $t \geq 2$,

$$\|\boldsymbol{u}_y^{(t)} - \boldsymbol{m}_y^{(t)}\|_2^2 \leq 2L^2\|\boldsymbol{x}^{(t)} - \boldsymbol{x}^{(t-1)}\|_2^2 + 2\|\mathcal{X}\|_2^2\|\mathbf{A}^{(t)} - \mathbf{A}^{(t-1)}\|_2^2. \quad (27)$$

Further, for $t = 1$ we have that $\|\boldsymbol{u}_x^{(1)} - \boldsymbol{m}_x^{(1)}\|_2 = \|\boldsymbol{u}_x^{(1)}\|_2 = \|-\mathbf{A}^{(1)}\boldsymbol{y}^{(1)}\|_2 \leq L\|\mathcal{Y}\|_2$, and $\|\boldsymbol{u}_y^{(1)} - \boldsymbol{m}_y^{(1)}\|_2 = \|\boldsymbol{u}_y^{(1)}\|_2 = \|(\mathbf{A}^{(1)})^\top\boldsymbol{x}^{(1)}\|_2 \leq L\|\mathcal{X}\|_2$. Next, we will use the following simple corollary, which follows similarly to Lemma A.1.

**Corollary A.8.** *For any sequence $s_z^{(T)} := (\boldsymbol{z}^{(t,\star)})_{1 \leq t \leq T}$, the dynamic regret $\mathrm{DReg}_z^{(T)}(s_z^{(T)}) := \mathrm{DReg}_x^{(T)}(s_x^{(T)}) + \mathrm{DReg}_y^{(T)}(s_y^{(T)})$ can be bounded by*

$$\frac{D_\mathcal{Z}^2}{2\eta} + \frac{D_\mathcal{Z}}{\eta}\sum_{t=1}^{T-1}\|\boldsymbol{z}^{(t+1,\star)} - \boldsymbol{z}^{(t,\star)}\|_2 + \eta\sum_{t=1}^T\|\boldsymbol{u}_z^{(t)} - \boldsymbol{m}_z^{(t)}\|_2^2$$

$$-\frac{1}{2\eta}\sum_{t=1}^T\left(\|\boldsymbol{z}^{(t)} - \hat{\boldsymbol{z}}^{(t)}\|_2^2 + \|\boldsymbol{z}^{(t)} - \hat{\boldsymbol{z}}^{(t+1)}\|_2^2\right),$$

*where $\boldsymbol{m}_z^{(t)} := (\boldsymbol{m}_x^{(t)}, \boldsymbol{m}_y^{(t)})$ and $\boldsymbol{u}_z^{(t)} := (\boldsymbol{u}_x^{(t)}, \boldsymbol{u}_y^{(t)})$ for any $t \in [\![T]\!]$.*

As a result, combining (27) and (26) with Corollary A.8 applied for the dynamic regret of both players with respect to the sequence of comparators $((\boldsymbol{x}^{(t,\star)}, \boldsymbol{y}^{(t,\star)}))_{1 \leq t \leq T}$ yields that $\mathrm{DReg}_x^{(T)}(\boldsymbol{x}^{(1,\star)}, \ldots, \boldsymbol{x}^{(T,\star)}) + \mathrm{DReg}_y^{(T)}(\boldsymbol{y}^{(1,\star)}, \ldots, \boldsymbol{y}^{(T,\star)})$ is upper bounded by

$$\frac{D_\mathcal{Z}^2}{2\eta} + \eta L^2\|\mathcal{Z}\|_2^2 + \frac{D_\mathcal{Z}}{\eta}\sum_{t=1}^{T-1}\|\boldsymbol{z}^{(t+1,\star)} - \boldsymbol{z}^{(t,\star)}\|_2 + 2\eta\|\mathcal{Z}\|_2^2\mathcal{V}_\mathbf{A}^{(T)}$$

$$-\frac{1}{4\eta}\sum_{t=1}^T\left(\|\boldsymbol{z}^{(t)} - \hat{\boldsymbol{z}}^{(t)}\|_2^2 + \|\boldsymbol{z}^{(t)} - \hat{\boldsymbol{z}}^{(t+1)}\|_2^2\right),$$

where we used the fact that

$$2\eta L^2\sum_{t=2}^T\|\boldsymbol{z}^{(t)} - \boldsymbol{z}^{(t-1)}\|_2^2 - \frac{1}{4\eta}\sum_{t=1}^T\left(\|\boldsymbol{z}^{(t)} - \hat{\boldsymbol{z}}^{(t)}\|_2^2 + \|\boldsymbol{z}^{(t)} - \hat{\boldsymbol{z}}^{(t+1)}\|_2^2\right)$$

$$\leq \left(2\eta L^2 - \frac{1}{8\eta}\right)\sum_{t=2}^T\|\boldsymbol{z}^{(t)} - \boldsymbol{z}^{(t-1)}\|_2^2 \leq 0,$$

for any leaning rate $\eta \leq \frac{1}{4L}$. Finally, using the fact that $\mathrm{DReg}_x^{(T)}(\boldsymbol{x}^{(1,\star)}, \ldots, \boldsymbol{x}^{(T,\star)}) + \mathrm{DReg}_y^{(T)}(\boldsymbol{y}^{(1,\star)}, \ldots, \boldsymbol{y}^{(T,\star)}) \geq -2\sum_{t=1}^T \epsilon^{(t)}$ for a suitable sequence of $\epsilon^{(t)}$-approximate Nash equilibria (Property 3.1)—one that attains the variation measure $\mathcal{V}_{\epsilon\text{-NE}}^{(T)}$—yields that

$$0 \leq \frac{D_\mathcal{Z}^2}{2\eta} + \eta L^2\|\mathcal{Z}\|_2^2 + \frac{D_\mathcal{Z}}{\eta}\mathcal{V}_{\epsilon\text{-NE}}^{(T)} + 2\eta\|\mathcal{Z}\|_2^2\mathcal{V}_\mathbf{A}^{(T)} - \frac{1}{4\eta}\sum_{t=1}^T\left(\|\boldsymbol{z}^{(t)} - \hat{\boldsymbol{z}}^{(t)}\|_2^2 + \|\boldsymbol{z}^{(t)} - \hat{\boldsymbol{z}}^{(t+1)}\|_2^2\right),$$

where in the above derivation it suffices if the parameter $C$ of $\mathcal{V}_{\epsilon\text{-NE}}^{(T)}$ is such that

$$2 \leq \frac{D_\mathcal{Z}}{\eta}C \iff C = C(\eta) \geq \frac{2\eta}{D_\mathcal{Z}}. \quad (28)$$

Thus, rearranging the last displayed inequality concludes the proof. $\qquad\square$

Next, we refine this theorem in time-varying games in which the deviation of the payoff matrices is bounded by the deviation of the players' strategies, in the following formal sense.

**Corollary A.9.** *Suppose that both players employ* OGD *with learning rate* $\eta \leq \min\left\{\frac{1}{4L}, \frac{1}{8W\|\mathcal{Z}\|}\right\}$ *in a time-varying bilinear saddle-point problem, where* $L := \max_{1 \leq t \leq T} \|\mathbf{A}^{(t)}\|_2$ *and* $\mathcal{V}_{\mathbf{A}}^{(T)} \leq W^2 \sum_{t=1}^{T-1} \|z^{(t+1)} - z^{(t)}\|_2^2$, *for some parameter* $W \in \mathbb{R}_{>0}$. *Then, for any time horizon* $T \in \mathbb{N}$,

$$\sum_{t=1}^{T} \left(\|z^{(t)} - \hat{z}^{(t)}\|_2^2 + \|\hat{z}^{(t)} - \hat{z}^{(t+1)}\|_2^2\right) \leq 4D_{\mathcal{Z}}^2 + 8\eta^2 L^2 \|\mathcal{Z}\|_2^2 + 8D_{\mathcal{Z}} \mathcal{V}_{NE}^{(T)}.$$

*Proof.* Following the proof of Theorem A.7, we have that for any $\eta \leq \frac{1}{4L}$,

$$0 \leq \frac{D_{\mathcal{Z}}^2}{2\eta} + \eta L^2 \|\mathcal{Z}\|_2^2 + \frac{D_{\mathcal{Z}}}{\eta} \mathcal{V}_{NE}^{(T)} + 2\eta \|\mathcal{Z}\|_2^2 \mathcal{V}_{\mathbf{A}}^{(T)} - \frac{1}{4\eta} \sum_{t=1}^{T} \left(\|z^{(t)} - \hat{z}^{(t)}\|_2^2 + \|z^{(t)} - \hat{z}^{(t+1)}\|_2^2\right).$$

Further, for $\eta \leq \frac{1}{8W\|\mathcal{Z}\|_2}$,

$$2\eta \|\mathcal{Z}\|_2^2 \mathcal{V}_{\mathbf{A}}^{(T)} - \frac{1}{8\eta} \sum_{t=1}^{T} \left(\|z^{(t)} - \hat{z}^{(t)}\|_2^2 + \|z^{(t)} - \hat{z}^{(t+1)}\|_2^2\right)$$
$$\leq \left(2\eta \|\mathcal{Z}\|_2^2 W^2 - \frac{1}{16\eta}\right) \sum_{t=1}^{T-1} \|z^{(t+1)} - z^{(t)}\|_2^2 \leq 0.$$

Thus, we have shown that

$$0 \leq \frac{D_{\mathcal{Z}}^2}{2\eta} + \eta L^2 \|\mathcal{Z}\|_2^2 + \frac{D_{\mathcal{Z}}}{\eta} \mathcal{V}_{NE}^{(T)} - \frac{1}{8\eta} \sum_{t=1}^{T} \left(\|z^{(t)} - \hat{z}^{(t)}\|_2^2 + \|z^{(t)} - \hat{z}^{(t+1)}\|_2^2\right),$$

and rearranging concludes the proof. $\qquad \square$

Thus, in such time-varying games it is the first-order variation term, $\mathcal{V}_{NE}^{(T)}$, that will drive our convergence bounds.

Now before proving Theorem 3.3, we state the connection between the equilibrium gap and the deviation of the players' strategies $\left(\|z^{(t)} - \hat{z}^{(t)}\|_2 + \|z^{(t)} - \hat{z}^{(t+1)}\|_2\right)$. In particular, the following claim can be extracted by [2, Claim A.14]. (We caution that we use a slightly different indexing for the secondary sequence $(\hat{x}_i^{(t)})$ in the definition of OMD (5) compared to [2].)

**Claim A.10.** *Suppose that the sequences* $(x^{(t)})_{1 \leq t \leq T}$ *and* $(\hat{x}^{(t)})_{1 \leq t \leq T+1}$ *are produced by* OMD *under a $G$-smooth regularizer 1-strongly convex with respect to a norm* $\|\cdot\|$. *Then, for any time* $t \in \llbracket T \rrbracket$ *and any* $x \in \mathcal{X}$,

$$\langle x^{(t)}, u_x^{(t)}\rangle \geq \langle x, u_x^{(t)}\rangle - \frac{GD_{\mathcal{X}}}{\eta} \|\hat{x}^{(t+1)} - \hat{x}^{(t)}\| - \|u_x^{(t)}\|_* \|x^{(t)} - \hat{x}^{(t+1)}\|.$$

We are now ready to prove Theorem 3.3, the precise version of which is stated below.

**Theorem A.11** (Detailed version of Theorem 3.3). *Suppose that both players employ* OGD *with learning rate* $\eta = \frac{1}{4L}$ *in a time-varying bilinear saddle-point problem, where* $L := \max_{1 \leq t \leq T} \|\mathbf{A}^{(t)}\|_2$. *Then,*

$$\sum_{t=1}^{T} \left(\text{EQGAP}^{(t)}(z^{(t)})\right)^2 \leq 2L^2 (4D_{\mathcal{Z}} + \|\mathcal{Z}\|_2)^2 \left(2D_{\mathcal{Z}}^2 + 4\eta^2 L^2 \|\mathcal{Z}\|_2^2 + 4D_{\mathcal{Z}} \mathcal{V}_{\epsilon-NE}^{(T)} + 8\eta^2 \|\mathcal{Z}\|_2^2 \mathcal{V}_{\mathbf{A}}^{(T)}\right),$$

*where* $(z^{(t)})_{1 \leq t \leq T}$ *is the sequence of joint strategy profiles produced by* OGD.

*Proof.* Let us fix $t \in \llbracket T \rrbracket$. For convenience, we denote by $\text{BR}_x^{(t)}(x^{(t)}) := \max_{x \in \mathcal{X}}\{\langle x, u_x^{(t)}\rangle\} - \langle x^{(t)}, u_x^{(t)}\rangle$, the best response gap of Player's $x$ strategy $x^{(t)} \in \mathcal{X}$, and similarly for $\text{BR}_y^{(t)}(y^{(t)})$.

By definition, it holds that $\text{EQGAP}^{(t)}(\boldsymbol{z}^{(t)}) := \max\{\text{BR}_x^{(t)}(\boldsymbol{x}^{(t)}), \text{BR}_y^{(t)}(\boldsymbol{y}^{(t)})\}$. By Claim A.10, we have that

$$\text{BR}_x^{(t)}(\boldsymbol{x}^{(t)}) \leq \frac{D_\mathcal{X}}{\eta}\|\hat{\boldsymbol{x}}^{(t+1)} - \hat{\boldsymbol{x}}^{(t)}\|_2 + \|\boldsymbol{u}_x^{(t)}\|_2\|\boldsymbol{x}^{(t)} - \hat{\boldsymbol{x}}^{(t+1)}\|_2 \tag{29}$$

$$\leq 4LD_\mathcal{X}\|\hat{\boldsymbol{x}}^{(t+1)} - \hat{\boldsymbol{x}}^{(t)}\|_2 + L\|\mathcal{Y}\|_2\|\boldsymbol{x}^{(t)} - \hat{\boldsymbol{x}}^{(t+1)}\|_2 \tag{30}$$

$$\leq L\left(4D_\mathcal{Z} + \|\mathcal{Z}\|_2\right)\left(\|\boldsymbol{x}^{(t)} - \hat{\boldsymbol{x}}^{(t)}\|_2 + \|\boldsymbol{x}^{(t)} - \hat{\boldsymbol{x}}^{(t+1)}\|_2\right), \tag{31}$$

where (29) follows from Claim A.10 for $G = 1$ (since the squared Euclidean regularizer $\phi_x : \boldsymbol{x} \mapsto \frac{1}{2}\|\boldsymbol{x}\|_2^2$) is trivially 1-smooth; (30) uses the fact that $\eta := \frac{1}{4L}$ and $\|\boldsymbol{u}_x^{(t)}\|_2 = \|-\mathbf{A}^{(t)}\boldsymbol{y}^{(t)}\|_2 \leq L\|\mathcal{Y}\|$; and (31) follows from the triangle inequality. A similar derivation shows that

$$\text{BR}_y^{(t)}(\boldsymbol{y}^{(t)}) \leq L(4D_\mathcal{Z} + \|\mathcal{Z}\|_2)\left(\|\boldsymbol{y}^{(t)} - \hat{\boldsymbol{y}}^{(t)}\|_2 + \|\boldsymbol{y}^{(t)} - \hat{\boldsymbol{y}}^{(t+1)}\|_2\right). \tag{32}$$

Thus,

$$
\begin{aligned}
\sum_{t=1}^{T}\left(\text{EQGAP}^{(t)}(\boldsymbol{z}^{(t)})\right)^2 &= \sum_{t=1}^{T}\left(\max\{\text{BR}_x^{(t)}(\boldsymbol{x}^{(t)}), \text{BR}_y^{(t)}(\boldsymbol{y}^{(t)})\}\right)^2 \\
&\leq \sum_{t=1}^{T}\left(\left(\text{BR}_x^{(t)}(\boldsymbol{x}^{(t)})\right)^2 + \left(\text{BR}_y^{(t)}(\boldsymbol{y}^{(t)})\right)^2\right) \\
&\leq 2L^2(4D_\mathcal{Z} + \|\mathcal{Z}\|_2)^2 \sum_{t=1}^{T}\left(\|\boldsymbol{z}^{(t)} - \hat{\boldsymbol{z}}^{(t)}\|_2^2 + \|\boldsymbol{z}^{(t)} - \hat{\boldsymbol{z}}^{(t+1)}\|_2^2\right), \tag{33}
\end{aligned}
$$

where the last bound uses (31) and (32). Combining (33) with Theorem A.7 concludes the proof. $\square$

### A.1.5 Variation-dependent regret bounds

Here we state for completeness an implication of Theorem 3.3 for deriving variation-dependent regret bounds in time-varying bilinear saddle-point problems; *cf.* [93].

**Corollary A.12.** *In the setup of Theorem A.7, it holds that*

$$
\begin{aligned}
\text{Reg}_x^{(T)} \leq \frac{D_\mathcal{X}^2}{\eta} + 8\eta L^2 D_\mathcal{Z}^2 + \eta L^2\|\mathcal{Y}\|_2^2 + 16\eta^3 L^4\|\mathcal{Z}\|_2^2 + 16\eta L^2 D_\mathcal{Z}\mathcal{V}_{NE}^{(T)} \\
+ (2\eta\|\mathcal{Y}\|_2^2 + 32\eta^3 L^2\|\mathcal{Z}\|_2^2)\mathcal{V}_\mathbf{A}^{(T)},
\end{aligned}
$$

*and*

$$
\begin{aligned}
\text{Reg}_y^{(T)} \leq \frac{D_\mathcal{Y}^2}{\eta} + 8\eta L^2 D_\mathcal{Z}^2 + \eta L^2\|\mathcal{X}\|_2^2 + 16\eta^3 L^4\|\mathcal{Z}\|_2^2 + 16\eta L^2 D_\mathcal{Z}\mathcal{V}_{NE}^{(T)} \\
+ (2\eta\|\mathcal{X}\|_2^2 + 32\eta^3 L^2\|\mathcal{Z}\|_2^2)\mathcal{V}_\mathbf{A}^{(T)}.
\end{aligned}
$$

*Proof.* First, applying Lemma A.1 under $\boldsymbol{x}^{(1,\star)} = \cdots = \boldsymbol{x}^{(T,\star)}$, we have

$$\text{Reg}_x^{(T)} \leq \frac{D_\mathcal{X}^2}{\eta} + \eta L^2\|\mathcal{Y}\|_2^2 + 2\eta L^2 \sum_{t=2}^{T}\|\boldsymbol{y}^{(t)} - \boldsymbol{y}^{(t-1)}\|_2^2 + 2\eta\|\mathcal{Y}\|_2^2 \sum_{t=2}^{T}\|\mathbf{A}^{(t)} - \mathbf{A}^{(t-1)}\|_2^2, \tag{34}$$

and similarly,

$$\text{Reg}_y^{(T)} \leq \frac{D_\mathcal{Y}^2}{\eta} + \eta L^2\|\mathcal{X}\|_2^2 + 2\eta L^2 \sum_{t=2}^{T}\|\boldsymbol{x}^{(t)} - \boldsymbol{x}^{(t-1)}\|_2^2 + 2\eta\|\mathcal{X}\|_2^2 \sum_{t=2}^{T}\|\mathbf{A}^{(t)} - \mathbf{A}^{(t-1)}\|_2^2. \tag{35}$$

Now, by Theorem A.7 we have

$$\sum_{t=1}^{T}\left(\|\boldsymbol{z}^{(t)} - \hat{\boldsymbol{z}}^{(t)}\|_2^2 + \|\boldsymbol{z}^{(t)} - \hat{\boldsymbol{z}}^{(t+1)}\|_2^2\right) \leq 2D_\mathcal{Z}^2 + 4\eta^2 L^2\|\mathcal{Z}\|_2^2 + 4D_\mathcal{Z}\mathcal{V}_{NE}^{(T)} + 8\eta^2\|\mathcal{Z}\|_2^2\mathcal{V}_\mathbf{A}^{(T)}. \tag{36}$$

Further,

$$\sum_{t=1}^{T} \left( \|\boldsymbol{z}^{(t)} - \hat{\boldsymbol{z}}^{(t)}\|_2^2 + \|\boldsymbol{z}^{(t)} - \hat{\boldsymbol{z}}^{(t+1)}\|_2^2 \right) \geq \sum_{t=1}^{T} \left( \|\boldsymbol{x}^{(t)} - \hat{\boldsymbol{x}}^{(t)}\|_2^2 + \|\boldsymbol{x}^{(t)} - \hat{\boldsymbol{x}}^{(t+1)}\|_2^2 \right)$$

$$\geq \frac{1}{2} \sum_{t=2}^{T} \|\boldsymbol{x}^{(t)} - \boldsymbol{x}^{(t-1)}\|_2^2.$$

Combining this bound with (36) and (35) gives the claimed regret bound on $\operatorname{Reg}_y^{(T)}$, and a similar derivation also gives the claimed bound on $\operatorname{Reg}_x^{(T)}$. □

Hence, selecting optimally the learning rate gives an $O\left(\sqrt{1 + \mathcal{V}_{\mathrm{NE}}^{(T)} + \mathcal{V}_{\mathbf{A}}^{(T)}}\right)$ bound on the *individual* regret of each player; while that optimal value depends on the variation measures, which are not known to the learners, there are techniques that would allow bypassing this [93]. Corollary A.12 can also be readily parameterized in terms of the improved variation measure $\mathcal{V}_{\epsilon-\mathrm{NE}}^{(T)}$.

### A.1.6   Meta-learning

We next provide the implication of Theorem 3.3 in the meta-learning setting. We first make a remark regarding the effect of the prediction of `OGD` to Theorem 3.3, and how that relates to an assumption present in earlier work on this problem [48].

*Remark* A.13 (Improved predictions). Throughout Section 3.1, we have considered the standard prediction $\boldsymbol{m}_x^{(t)} := \boldsymbol{u}_x^{(t-1)} = -\mathbf{A}^{(t-1)}\boldsymbol{y}^{(t-1)}$ for $t \geq 2$, and similarly for Player $y$. It is easy to see that using the predictions

$$\boldsymbol{m}_x^{(t)} := -\mathbf{A}^{(t)}\boldsymbol{y}^{(t-1)} \quad \text{and} \quad \boldsymbol{m}_y^{(t)} := (\mathbf{A}^{(t)})^\top \boldsymbol{x}^{(t-1)} \tag{37}$$

for $t \geq 1$ (where $\boldsymbol{z}^{(0)} := \hat{\boldsymbol{z}}^{(1)}$) entirely removes the dependency on $\mathcal{V}_{\mathbf{A}}^{(T)}$ on all our convergence bounds. While such a prediction cannot be implemented in the standard online learning model, there are settings in which we might, for example, know the sequence of matrices in advance; the meta-learning setting offers such examples, and indeed, Harris et al. [48] use the improved prediction of (37).

**Proposition A.14** (Meta-learning). *Suppose that both players employ* `OGD` *with learning rate* $\eta = \frac{1}{4L}$, *where* $L := \max_{1 \leq h \leq H} \|\mathbf{A}^{(h)}\|_2$, *and the prediction of* (37) *in a meta-learning bilinear saddle-point problem with* $H \in \mathbb{N}$ *games, each repeated for* $m \in \mathbb{N}$ *consecutive iterations. Then, for an average game,*

$$\left\lceil \frac{P}{H\epsilon^2} + \frac{P'\mathcal{V}_{NE}^{(H)}}{H\epsilon^2} \right\rceil + 1 \tag{38}$$

*iterations suffice to reach an* $\epsilon$*-approximate Nash equilibrium, where* $P := 4L^2(4D_{\mathcal{Z}} + \|\mathcal{Z}\|_2)^2 D_{\mathcal{Z}}^2$, $P' := 8L^2(4D_{\mathcal{Z}} + \|\mathcal{Z}\|_2)^2 D_{\mathcal{Z}}$, *and*

$$\mathcal{V}_{NE}^{(H)} := \inf_{\boldsymbol{z}^{(h,\star)} \in \mathcal{Z}^{(h,\star)}, \forall h \in \llbracket H \rrbracket} \sum_{h=1}^{H-1} \|\boldsymbol{z}^{(h+1,\star)} - \boldsymbol{z}^{(h,\star)}\|_2.$$

The proof is a direct application of Theorem A.11, where we remark that the term depending on $\mathcal{V}_{\mathbf{A}}^{(T)}$ and the term $4\eta^2 L^2 \|\mathcal{Z}\|_2^2$ from Theorem A.11 are eliminated because of the improved prediction of Remark A.13. More precisely, if we let $m^{(h)} \in \mathbb{N}$ be the number of iterations required so that the equilibrium gap is at most $\epsilon$ at the $h$-th game, then Theorem A.11 implies that

$$P + P'\mathcal{V}_{\mathrm{NE}}^{(H)} \geq \sum_{t=1}^{T} \left( \mathrm{EQGAP}^{(t)}(\boldsymbol{z}^{(t)}) \right)^2 > \epsilon^2 \sum_{h=1}^{H} (m^{(h)} - 1).$$

Solving with respect to $\frac{1}{H} \sum_{h=1}^{H} m^{(h)}$ thus yields Proposition A.14.

The first term in the iteration complexity bound (38) vanishes in the meta-learning regime—as the number of games increases $H \gg 1$—while the second term is proportional to $\frac{\mathcal{V}_{\mathrm{NE}}^{(H)}}{H}$, a natural similarity measure; (38) always recovers the $m^{-1/2}$ rate, but offers significant gains if the games as similar, in the sense that $\frac{\mathcal{V}_{\mathrm{NE}}^{(H)}}{H} \ll 1$. It is worth noting that, unlike the similarity measure derived by Harris et al. [48], $\frac{\mathcal{V}_{\mathrm{NE}}^{(H)}}{H}$ depends on the order of the games. As a result, a natural problem that arises in settings where the sequence of games is known in advance is to optimize the order of the games so as to minimize $\mathcal{V}_{\mathrm{NE}}^{(H)}$; the key challenge of course is to minimize $\mathcal{V}_{\mathrm{NE}}^{(H)}$ without actually solving each game since that without defeat the purpose. We further remark that Proposition A.14 can be readily extended even if each game in the meta-learning sequence is not repeated for the same number of iterations.

Another natural question in this context is to compare the similarity metric of Proposition A.14 compared to that obtained by Harris et al. [48]—namely $\min_{\boldsymbol{z} \in \mathcal{Z}} \sum_{h=1}^{H} \|\boldsymbol{z}^{(h,\star)} - \boldsymbol{z}\|_2^2$, where we can assume here that $\boldsymbol{z}^{(h,\star)}$ is the unique Nash equilibrium of the $h$-th game. Below, we observe that $\mathcal{V}_{\mathrm{NE}}^{(H)}$ can be arbitrarily smaller.

**Proposition A.15.** *There exists a sequence of $H$ games such that $\mathcal{V}_{NE}^{(H)} = O(1)$ while $\min_{\boldsymbol{z} \in \mathcal{Z}} \sum_{h=1}^{H} \|\boldsymbol{z}^{(h,\star)} - \boldsymbol{z}\|_2^2 = \Omega(H)$.*

*Proof.* We let $\mathcal{X}, \mathcal{Y} = \Delta^3$. We consider a sequence of $H$ games so that $\boldsymbol{x}^{(h,\star)} = \boldsymbol{y}^{(h,\star)}$ for all $h \in \llbracket H \rrbracket$, and the sequence of points $(\boldsymbol{x}^{(h,\star)})_{1 \le h \le H}$ forms a regular $H$-gon within the relative interior of $\Delta^3$ with a constant radius. We note that a unique fully-mixed equilibrium $((\alpha, \beta, \gamma), (\alpha, \beta, \gamma)) \in \Delta^3 \times \Delta^3$ can be obtained by considering the diagonal payoff matrix

$$\begin{bmatrix} 1/\alpha & 0 & 0 \\ 0 & 1/\beta & 0 \\ 0 & 0 & 1/\gamma \end{bmatrix}.$$

It is easy to see that the above sequence of games establishes the claim. $\qquad \square$

### A.1.7 Polymatrix zero-sum games

As we claimed earlier in Section 3.1.1, Theorem 3.3 can be extended to more general time-varying VIs as long as the MVI property holds. For concreteness, here we focus on a multi-player generalization of two-player zero-sum games, namely *polymatrix zero-sum* games [15]. More precisely, here it is assumed that there is an underlying $n$-node graph so that each player is uniquely associated with a node in the graph. The utility of each player $i \in \llbracket n \rrbracket$ can be expressed as $\sum_{i' \in \mathcal{N}_i} \boldsymbol{x}_i^\top \mathbf{A}_{i,i'} \boldsymbol{x}_{i'}$, where $\mathcal{N}_i$ denotes the set of neighbors of Player $i$ and $\mathbf{A}_{i,i'} \in \mathbb{R}^{d_i \times d_{i'}}$. It is also assumed that $\mathbf{A}_{i,i'}^\top = -\mathbf{A}_{i',i}$, for any edge $\{i, i'\}$. If $\boldsymbol{z} := (\boldsymbol{x}_1, \ldots, \boldsymbol{x}_n) \in \mathcal{Z}$, the corresponding operator is defined as

$$F(\boldsymbol{z}) = -\left( \sum_{i \in \mathcal{N}_1} \mathbf{A}_{1,i} \boldsymbol{x}_i, \ldots, \sum_{i \in \mathcal{N}_n} \mathbf{A}_{n,i} \boldsymbol{x}_i \right).$$

It is easy to see that $F$ is *monotone*, in that $\langle \boldsymbol{z} - \boldsymbol{z}', F(\boldsymbol{z}) - F(\boldsymbol{z}') \rangle \ge 0$, for any $\boldsymbol{z}, \boldsymbol{z}' \in \mathcal{Z}$. Now, for any $\epsilon$-approximate Nash equilibrium $\boldsymbol{z}^\star$ it holds, by definition, that $\langle \boldsymbol{z} - \boldsymbol{z}^\star, F(\boldsymbol{z}^\star) \rangle \ge -n\epsilon$, for any $\boldsymbol{z} \in \mathcal{Z}$. Thus, by monotonicity, $\langle \boldsymbol{z} - \boldsymbol{z}^\star, F(\boldsymbol{z}) \rangle \ge \langle \boldsymbol{z} - \boldsymbol{z}^\star, F(\boldsymbol{z}^\star) \rangle \ge -n\epsilon$, for any $\boldsymbol{z} \in \mathcal{Z}$. In other words, approximate Nash equilibria approximately satisfy the MVI property. This suffices to directly extend Property 3.1 to time-varying polymatrix zero-sum games. The rest of the argument of Theorem 3.3 is analogous, leading to the following result. Below, we define $\mathcal{V}_{\epsilon-\mathrm{NE}}^{(T)}$ as in (2), for a suitable parameter $C > 0$, and $\mathcal{V}_{\mathbf{A}}^{(T)} := \sum_{t=1}^{T-1} \sum_{i=1}^{n} \sum_{i' \in \mathcal{N}_{i'}} \|\mathbf{A}_{i,i'}^{(t+1)} - \mathbf{A}_{i,i'}^{(t)}\|_2^2$.

**Theorem A.16.** *Suppose that each player employs* OGD *with a sufficiently small learning rate in a sequence of time-varying polymatrix zero-sum games. Then, $\sum_{t=1}^{T} \left( \mathrm{EQGAP}^{(t)}(\boldsymbol{z}^{(t)}) \right)^2 = O\left( 1 + \mathcal{V}_{\epsilon-NE}^{(T)} + \mathcal{V}_{\mathbf{A}}^{(T)} \right)$, where $(\boldsymbol{z}^{(t)})_{1 \le t \le T}$ is the sequence of joint strategies produced by* OGD.

### A.1.8 General variational inequalities

Although our main focus in this paper is on the convergence of learning algorithms in time-varying games, our techniques could also be of interest for solving (static) general variational inequality (VI) problems.

In particular, let $F : \mathcal{Z} \to \mathcal{Z}$ be a single-valued operator. Solving general VIs is well-known to be computationally intractable, and so instead focus has been on identifying broad subclasses that elude those intractability barriers (see the references below). Our framework in Section 3.1 motivates introducing the following measure of complexity for a VI problem:

$$\mathfrak{C}(F) \coloneqq \inf_{\boldsymbol{z}^{(1,\star)},\ldots,\boldsymbol{z}^{(T,\star)} \in \mathcal{Z}} \sum_{t=1}^{T-1} \|\boldsymbol{z}^{(t+1,\star)} - \boldsymbol{z}^{(t,\star)}\|_2, \tag{39}$$

subject to

$$\mathrm{DReg}_z^{(T)}(\boldsymbol{z}^{(1,\star)}, \ldots, \boldsymbol{z}^{(T,\star)}) \geq 0 \iff \sum_{t=1}^{T} \langle \boldsymbol{z}^{(t)} - \boldsymbol{z}^{(t,\star)}, F(\boldsymbol{z}^{(t)}) \rangle \geq 0. \tag{40}$$

In words, (39) expresses the infimum first-order variation that a sequence of comparators must have in order to guarantee nonnegative dynamic regret (40); it is evident that (40) always admits a feasible sequence, namely $s_z^{(T)} \coloneqq (\boldsymbol{z}^{(t)})_{1 \leq t \leq T}$. We note that, in accordance to our results in Section 3.1, one can also consider an approximate version of the complexity measure (39), which could behave much more favorably (recall Proposition 3.2).

Now in a (static) bilinear saddle-point problem, it holds that $\mathfrak{C}(F) = 0$ given that there exists a static comparator that guarantees nonnegativity of the dynamic regret. More broadly, our techniques imply $O(\mathrm{poly}(1/\epsilon))$ iteration-complexity bounds for any VI problem such that $\mathfrak{C}(F) \leq CT^{1-\omega}$, for a time-independent parameter $C > 0$ and $\omega \in (0,1]$:

**Proposition A.17.** *Consider a variational inequality problem described with the operator $F : \mathcal{Z} \to \mathcal{Z}$ such that $F$ is $L$-Lipschitz continuous, in the sense that $\|F(\boldsymbol{z}) - F(\boldsymbol{z}')\|_2 \leq L\|\boldsymbol{z} - \boldsymbol{z}'\|_2$, and $\mathfrak{C}(F) \leq CT^{1-\omega}$ for $C > 0$ and $\omega \in (0,1]$. Then, $\mathtt{OGD}$ with learning rate $\eta = \frac{1}{4L}$ reaches an $\epsilon$-strong solution $\boldsymbol{z}^\star \in \mathcal{Z}$ in $O(\epsilon^{-2/\omega})$ iterations; that is, $\langle \boldsymbol{z} - \boldsymbol{z}^\star, F(\boldsymbol{z}^\star) \rangle \geq -\epsilon$ for any $\boldsymbol{z} \in \mathcal{Z}$.*

This result should be viewed as part of an ongoing effort to characterize the class of variational inequalities (VIs) that are amenable to efficient algorithms; see [33, 13, 4, 7, 21, 23, 56, 57, 66, 76, 87, 24], and the many references therein.

It is worth comparing (39) with another natural complexity measure, namely $\inf_{\boldsymbol{z}^\star \in \mathcal{Z}} \sum_{t=1}^{T} \langle \boldsymbol{z}^{(t)} - \boldsymbol{z}^\star, F(\boldsymbol{z}^{(t)}) \rangle$; the latter measures how negative (external) regret can be, and has already proven useful in certain settings that go bilinear saddle-point problems [88], although unlike (39) it does not appear to be of much use in characterizing time-varying bilinear saddle-point problems. In this context, $O(\mathrm{poly}(1/\epsilon))$ iteration-complexity bounds can also be established whenever

- $\inf_{\boldsymbol{z}^\star \in \mathcal{Z}} \sum_{t=1}^{T} \langle \boldsymbol{z}^{(t)} - \boldsymbol{z}^\star, F(\boldsymbol{z}^{(t)}) \rangle \geq -CT^{1-\omega}$ for a time-invariant $C > 0$, or
- $\inf_{\boldsymbol{z}^\star \in \mathcal{Z}} \sum_{t=1}^{T} \langle \boldsymbol{z}^{(t)} - \boldsymbol{z}^\star, F(\boldsymbol{z}^{(t)}) \rangle \geq -C \sum_{t=1}^{T-1} \|\boldsymbol{z}^{(t+1)} - \boldsymbol{z}^{(t)}\|_2^2$, for a sufficiently small $C > 0$.

Following [88], identifying VIs that satisfy those relaxed conditions but not the MVI property is an interesting direction. In particular, it is important to understand if those relaxations can shed led more light into the convergence properties of $\mathtt{OGD}$ in Shapley's two-player zero-sum *stochastic games*.

### A.2 Proofs from Section 3.2

In this subsection, we provide the proofs from Section 3.2, leading to our main result in Theorem 3.5.

Let us first introduce some additional notation. We let $f^{(t)} : \mathcal{X} \times \mathcal{Y} \to \mathbb{R}$ be a continuously differentiable function for any $t \in \llbracket T \rrbracket$.[6] We recall that in Section 3.2 it is assumed that the objective

---

[6]In case the interior $(\mathcal{X} \times \mathcal{Y})^\circ$ is empty, here and throughout this paper we tacitly posit differentiability on a closed and convex neighborhood $\tilde{\mathcal{X}} \times \tilde{\mathcal{Y}}$ such that $\mathcal{X} \subseteq \tilde{\mathcal{X}}^\circ$ and $\mathcal{Y} \subseteq \tilde{\mathcal{Y}}^\circ$. This assumption suffices for our arguments; see, for example, the treatment of Daskalakis et al. [28].

function changes only after $m \in \mathbb{N}$ (consecutive) repetitions, which is akin to the meta-learning setting. Analogously to our setup for bilinear saddle-point problems (Section 3.1), it is assumed that Player $x$ is endeavoring to minimizing the objective function, while Player $y$ is trying to maximize it. We will denote by $\operatorname{Reg}_{\mathcal{L},x}^{(T)}(\boldsymbol{x}^\star) := \sum_{t=1}^T \langle \boldsymbol{x}^{(t)} - \boldsymbol{x}^\star, -\boldsymbol{u}_x^{(t)} \rangle$ and $\operatorname{Reg}_{\mathcal{L},y}^{(T)}(\boldsymbol{y}^\star) := \sum_{t=1}^T \langle \boldsymbol{y}^\star - \boldsymbol{y}^{(t)}, \boldsymbol{u}_y^{(t)} \rangle$, where $\boldsymbol{u}_x^{(t)} := -\nabla_{\boldsymbol{x}} f(\boldsymbol{x}^{(t)}, \boldsymbol{y}^{(t)})$ and $\boldsymbol{u}_y^{(t)} := \nabla_{\boldsymbol{y}} f(\boldsymbol{x}^{(t)}, \boldsymbol{y}^{(t)})$ for any $t \in \llbracket T \rrbracket$; similar notation is used for $\operatorname{DReg}_{\mathcal{L},x}^{(T)}, \operatorname{DReg}_{\mathcal{L},y}^{(T)}$.

Furthermore, we let $s_z^{(T)} = ((\boldsymbol{x}^{(t,\star)}, \boldsymbol{y}^{(t,\star)}))_{1 \le t \le T}$, so that $\boldsymbol{x}^{(t,\star)} = \boldsymbol{x}^{(h,\star)}$ and $\boldsymbol{y}^{(t,\star)} = \boldsymbol{y}^{(h,\star)}$ for any $t \in \llbracket T \rrbracket$ such that $\lfloor (t-1)/m \rfloor = h \in \llbracket H \rrbracket$. The first important step in our analysis is that, following the proof of Lemma A.1,

$$\operatorname{DReg}_{\mathcal{L},x}^{(T)}(s_x^{(T)}) \le \frac{1}{2\eta} \sum_{h=1}^H \|\hat{\boldsymbol{x}}^{(h,1)} - \boldsymbol{x}^{(h,\star)}\|_2^2 - \|\hat{\boldsymbol{x}}^{(h,m+1)} - \boldsymbol{x}^{(h,\star)}\|_2^2 + \eta \sum_{t=1}^T \|\boldsymbol{u}_x^{(t)} - \boldsymbol{m}_x^{(t)}\|_2^2$$

$$- \frac{1}{2\eta} \sum_{t=1}^T \left( \|\boldsymbol{x}^{(t)} - \hat{\boldsymbol{x}}^{(t)}\|_2^2 + \|\boldsymbol{x}^{(t)} - \hat{\boldsymbol{x}}^{(t+1)}\|_2^2 \right), \qquad (41)$$

where $\hat{\boldsymbol{x}}^{(h,k)} := \hat{\boldsymbol{x}}^{((h-1) \times m) + k}$ for any $(h,k) \in \llbracket H \rrbracket \times \llbracket m \rrbracket$, $\hat{\boldsymbol{x}}^{(h,m+1)} := \hat{\boldsymbol{x}}^{(h+1,1)}$ for $h \in \llbracket H-1 \rrbracket$, and $\hat{\boldsymbol{x}}^{(H,m+1)} := \hat{\boldsymbol{x}}^{(T+1)}$. Similarly,

$$\operatorname{DReg}_{\mathcal{L},y}^{(T)}(s_y^{(T)}) \le \frac{1}{2\eta} \sum_{h=1}^H \|\hat{\boldsymbol{y}}^{(h,1)} - \boldsymbol{y}^{(h,\star)}\|_2^2 - \|\hat{\boldsymbol{y}}^{(h,m+1)} - \boldsymbol{y}^{(h,\star)}\|_2^2 + \eta \sum_{t=1}^T \|\boldsymbol{u}_y^{(t)} - \boldsymbol{m}_y^{(t)}\|_2^2$$

$$- \frac{1}{2\eta} \sum_{t=1}^T \left( \|\boldsymbol{y}^{(t)} - \hat{\boldsymbol{y}}^{(t)}\|_2^2 + \|\boldsymbol{y}^{(t)} - \hat{\boldsymbol{y}}^{(t+1)}\|_2^2 \right). \qquad (42)$$

Next, we will use the following key observation, which lower bounds the sum of the players' (external) regrets under strong convexity-concavity.

**Lemma A.18.** *Suppose that $f : \mathcal{X} \times \mathcal{Y} \to \mathbb{R}$ is a $\mu$-strongly convex-concave function with respect to $\|\cdot\|_2$. Then, for any Nash equilibrium $\boldsymbol{z}^\star = (\boldsymbol{x}^\star, \boldsymbol{y}^\star) \in \mathcal{Z}$,*

$$\operatorname{Reg}_{\mathcal{L},x}^{(m)}(\boldsymbol{x}^\star) + \operatorname{Reg}_{\mathcal{L},y}^{(m)}(\boldsymbol{y}^\star) \ge \frac{\mu}{2} \sum_{t=1}^m \|\boldsymbol{z}^{(t)} - \boldsymbol{z}^\star\|_2^2.$$

*Proof.* First, by $\mu$-strong convexity of $f(\boldsymbol{x}, \cdot)$, we have that for any time $t \in \llbracket m \rrbracket$,

$$\langle \boldsymbol{x}^{(t)} - \boldsymbol{x}^\star, \nabla_{\boldsymbol{x}} f(\boldsymbol{x}^{(t)}, \boldsymbol{y}^{(t)}) \rangle \ge f(\boldsymbol{x}^{(t)}, \boldsymbol{y}^{(t)}) - f(\boldsymbol{x}^\star, \boldsymbol{y}^{(t)}) + \frac{\mu}{2} \|\boldsymbol{x}^{(t)} - \boldsymbol{x}^\star\|_2^2. \qquad (43)$$

Similarly, by $\mu$-strong concavity of $f(\cdot, \boldsymbol{y})$, we have that for any time $t \in \llbracket m \rrbracket$,

$$\langle \boldsymbol{y}^\star - \boldsymbol{y}^{(t)}, \nabla_{\boldsymbol{y}} f(\boldsymbol{x}^{(t)}, \boldsymbol{y}^{(t)}) \rangle \ge f(\boldsymbol{x}^{(t)}, \boldsymbol{y}^\star) - f(\boldsymbol{x}^{(t)}, \boldsymbol{y}^{(t)}) + \frac{\mu}{2} \|\boldsymbol{y}^{(t)} - \boldsymbol{y}^\star\|_2^2. \qquad (44)$$

Further, for any Nash equilibrium $(\boldsymbol{x}^\star, \boldsymbol{y}^\star) \in \mathcal{Z}$ it holds that $f(\boldsymbol{x}^{(t)}, \boldsymbol{y}^\star) \ge f(\boldsymbol{x}^{(t)}, \boldsymbol{y}^{(t)}) \ge f(\boldsymbol{x}^\star, \boldsymbol{y}^{(t)})$. Combining this fact with (43) and (44) and summing over all $t \in \llbracket m \rrbracket$ gives the statement. $\square$

In turn, this readily implies the following lower bound for the dynamic regret.

**Lemma A.19.** *Suppose that $f^{(h)} : \mathcal{X} \times \mathcal{Y} \to \mathbb{R}$ is a $\mu$-strongly convex-concave function with respect to $\|\cdot\|_2$, for any $h \in \llbracket H \rrbracket$. Consider a sequence $s_z^{(T)} = ((\boldsymbol{x}^{(t,\star)}, \boldsymbol{y}^{(t,\star)}))_{1 \le t \le T}$, so that $\boldsymbol{x}^{(t,\star)} = \boldsymbol{x}^{(h,\star)}$ and $\boldsymbol{y}^{(t,\star)} = \boldsymbol{y}^{(h,\star)}$ for any $t \in \llbracket T \rrbracket$ such that $\lfloor (t-1)/m \rfloor = h \in \llbracket H \rrbracket$. If $(\boldsymbol{x}^{(h,\star)}, \boldsymbol{y}^{(h,\star)}) \in \mathcal{Z}$ is a Nash equilibrium of $f^{(h)}$,*

$$\operatorname{DReg}_{\mathcal{L},x}^{(T)}(s_x^{(T)}) + \operatorname{DReg}_{\mathcal{L},y}^{(T)}(s_y^{(T)}) \ge \frac{\mu}{2} \sum_{h=1}^H \sum_{k=1}^m \|\boldsymbol{z}^{(h,k)} - \boldsymbol{z}^{(h,\star)}\|_2^2,$$

*where $\boldsymbol{z}^{(h,k)} := \boldsymbol{z}^{((h-1) \times m) + k}$ for any $(h,k) \in \llbracket H \rrbracket \times \llbracket m \rrbracket$.*

We next combine this with the following monotonicity property of OGD: If $z^\star$ is a Nash equilibrium, $\|\hat{z}^{(t)} - z^\star\|_2$ is a decreasing function in $t$ [48, Proposition C.10]. This leads to the following refinement of Lemma A.19.

**Lemma A.20.** *Under the assumptions of Lemma A.19, if additionally $\eta \leq \frac{1}{2\mu}$,*

$$
\mathrm{DReg}_{\mathcal{L},x}^{(T)}(s_x^{(T)}) + \mathrm{DReg}_{\mathcal{L},y}^{(T)}(s_y^{(T)}) + \frac{1}{4\eta}\sum_{t=1}^{T}\|z^{(t)} - \hat{z}^{(t+1)}\|_2^2 \geq \frac{\mu m}{4}\sum_{h=1}^{H}\|\hat{z}^{(h,m+1)} - z^{(h,\star)}\|_2^2.
$$

*Proof.* By Lemma A.19,

$$
\mathrm{DReg}_{\mathcal{L},x}^{(T)}(s_x^{(T)}) + \mathrm{DReg}_{\mathcal{L},y}^{(T)}(s_y^{(T)}) + \frac{1}{4\eta}\sum_{t=1}^{T}\|z^{(t)} - \hat{z}^{(t+1)}\|_2^2
$$

$$
\geq \frac{\mu}{2}\sum_{h=1}^{H}\sum_{k=1}^{m}\|z^{(h,k)} - z^{(h,\star)}\|_2^2 + \frac{1}{4\eta}\sum_{t=1}^{T}\|z^{(t)} - \hat{z}^{(t+1)}\|_2^2
$$

$$
\geq \frac{\mu}{4}\sum_{h=1}^{H}\sum_{k=1}^{m}\|\hat{z}^{(h,k+1)} - z^{(h,\star)}\|_2^2 \tag{45}
$$

$$
\geq \frac{\mu m}{4}\sum_{h=1}^{H}\|\hat{z}^{(h,m+1)} - z^{(h,\star)}\|_2^2, \tag{46}
$$

where (45) uses that $\frac{1}{4\eta} \geq \frac{\mu}{2}$ along with Young's inequality and triangle inequality, and (46) follows from [48, Proposition C.10]. $\square$

Armed with this important lemma, we are ready to establish our main result (Theorem 3.5), the detailed version of which is given below. We first recall that the function $f : \mathcal{X} \times \mathcal{Y} \to \mathbb{R}$ is said to be $L$-smooth if $\|F(z) - F(z')\|_2 \leq L\|z - z'\|_2$, where $F(z) := (\nabla_x f(x,y), -\nabla_y f(x,y))$.

**Theorem A.21** (Detailed version of Theorem 3.5). *Let $f^{(h)} : \mathcal{X} \times \mathcal{Y}$ be a $\mu$-strongly convex-concave and $L$-smooth function, for $h \in [\![H]\!]$. Suppose that both players employ OGD with learning rate $\eta = \min\left\{\frac{1}{8L}, \frac{1}{2\mu}\right\}$ for $T$ repetitions, where $T = m \times H$ and $m \geq \frac{2}{\eta\mu}$. Then,*

$$
\sum_{t=1}^{T}\left(\|z^{(t)} - \hat{z}^{(t)}\|_2^2 + \|z^{(t)} - \hat{z}^{(t+1)}\|_2^2\right) \leq 4D_{\mathcal{Z}}^2 + 8\eta^2\|F(z^{(1)})\|_2^2 + 8\mathcal{S}_{NE}^{(H)} + 16\eta^2\mathcal{V}_{\nabla f}^{(H)}.
$$

*Thus, $\sum_{t=1}^{T}\left(\mathrm{EqGap}^{(t)}(z^{(t)})\right)^2 = O(1 + \mathcal{S}_{NE}^{(H)} + \mathcal{V}_{\nabla f}^{(H)})$.*

*Proof.* Combining Lemma A.20 with (41) and (42),

$$
0 \leq \frac{1}{2\eta}\sum_{h=1}^{H}\left(\|\hat{z}^{(h,1)} - z^{(h,\star)}\|_2^2 - \frac{2 + \eta\mu m}{2}\|\hat{z}^{(h,m+1)} - z^{(h,\star)}\|_2^2\right)
$$

$$
+ \eta\sum_{t=1}^{T}\|u_z^{(t)} - m_z^{(t)}\|_2^2 - \frac{1}{4\eta}\sum_{t=1}^{T}\left(\|z^{(t)} - \hat{z}^{(t)}\|_2^2 + \|z^{(t)} - \hat{z}^{(t+1)}\|_2^2\right),
$$

for a sequence of Nash equilibria $(z^{(h,\star)})_{1 \leq h \leq H}$, where we used the notation $u_z^{(t)} := (u_x^{(t)}, u_y^{(t)})$ and $m_z^{(t)} := (m_x^{(t)}, m_y^{(t)})$. Now we bound the first term of the right-hand side above as

$$
\frac{1}{2\eta}\sum_{h=1}^{H}\left(\|\hat{z}^{(h,1)} - z^{(h,\star)}\|_2^2 - 2\|\hat{z}^{(h,m+1)} - z^{(h,\star)}\|_2^2\right) \leq
$$

$$
\frac{1}{2\eta}\|\hat{z}^{(1,1)} - z^{(1,\star)}\|_2^2 + \frac{1}{2\eta}\sum_{h=1}^{H-1}\left(\|\hat{z}^{(h+1,1)} - z^{(h+1,\star)}\|_2^2 - 2\|\hat{z}^{(h+1,1)} - z^{(h,\star)}\|_2^2\right),
$$

where we used the fact that $m \geq \frac{2}{\eta\mu}$ and $\hat{z}^{(h,m+1)} = \hat{z}^{(h+1,1)}$, for $h \in [\![H-1]\!]$. Hence, continuing from above,

$$\frac{1}{2\eta} \sum_{h=1}^{H} \left( \|\hat{z}^{(h,1)} - z^{(h,\star)}\|_2^2 - 2\|\hat{z}^{(h,m+1)} - z^{(h,\star)}\|_2^2 \right) \leq \frac{1}{2\eta} \|\hat{z}^{(1,1)} - z^{(1,\star)}\|_2^2$$
$$+ \frac{1}{\eta} \sum_{h=1}^{H-1} \|z^{(h+1,\star)} - z^{(h,\star)}\|_2^2,$$

since $\|\hat{z}^{(h+1,1)} - z^{(h+1,\star)}\|_2^2 \leq 2\|\hat{z}^{(h+1,1)} - z^{(h,\star)}\|_2^2 + 2\|z^{(h,\star)} - z^{(h+1,\star)}\|_2^2$, by the triangle inequality and Young's inequality. Moreover, for $t \geq 2$,

$$\|u_z^{(t)} - u_z^{(t-1)}\|_2^2 = \|F^{(t)}(z^{(t)}) - F^{(t-1)}(z^{(t-1)})\|_2^2 \leq 2L^2 \|z^{(t)} - z^{(t-1)}\|_2^2$$
$$+ 2\|F^{(t)}(z^{(t-1)}) - F^{(t-1)}(z^{(t-1)})\|_2^2,$$

by $L$-smoothness. As a result,

$$\sum_{t=1}^{T-1} \|u_z^{(t+1)} - u_z^{(t)}\|_2^2 \leq 2L^2 \sum_{t=1}^{T-1} \|z^{(t)} - z^{(t-1)}\|_2^2 + 2\mathcal{V}_{\nabla f}^{(H)},$$

and the claimed bound on the second-order path length follows. Finally, the second claim of the theorem follows from Claim A.10 using convexity-concavity, analogously to Theorem 3.3. □

We conclude this subsection by pointing out an improved variation-dependent regret bound, which follows directly from Theorem A.21 (*cf.* Corollary A.12).

**Corollary A.22.** *In the setup of Theorem A.21, the maximum of the two players' regrets,* $\max\{\mathrm{Reg}_{\mathcal{L},x}^{(T)}, \mathrm{Reg}_{\mathcal{L},y}^{(T)}\}$, *can be upper bounded by*

$$\frac{D_{\mathcal{Z}}^2}{\eta} + 16\eta L^2 D_{\mathcal{Z}}^2 + (32\eta^3 L^2 + \eta)\|F(z^{(1)})\|_2^2 + 32\eta L^2 \mathcal{S}_{NE}^{(H)} + (64\eta^3 L^2 + 2\eta)\mathcal{V}_{\nabla f}^{(H)}.$$

Thus, setting the learning rate optimally implies that $\mathrm{Reg}_{\mathcal{L},x}^{(T)}, \mathrm{Reg}_{\mathcal{L},y}^{(T)} = O\left(\sqrt{1 + \mathcal{S}_{NE}^{(H)} + \mathcal{V}_{\nabla f}^{(H)}}\right)$.

## A.3 Proofs from Section 3.3

In this subsection, we provide the proofs from Section 3.3.

### A.3.1 Potential games

We first characterize the behavior of (online) gradient descent (GD) in time-varying potential games; we recall that GD is equivalent to OGD under the prediction $m_x^{(t)} = 0$ for all $t$. Below we give the formal definition of a potential game.

**Definition A.23** (Potential game). An $n$-player game admits a *potential* if there exists a function $\Phi : \bigtimes_{i=1}^{n} \mathcal{X}_i \to \mathbb{R}$ such that for any player $i \in [\![n]\!]$, any joint strategy profile $x_{-i} \in \bigtimes_{i' \neq i} \mathcal{X}_{i'}$, and any pair of strategies $x_i, x_i' \in \mathcal{X}_i$,

$$\Phi(x_i, x_{-i}) - \Phi(x_i', x_{-i}) = u_i(x_i, x_{-i}) - u_i(x_i', x_{-i}).$$

The key ingredient in the proof of Theorem 3.7 is the following bound on the second-order path length of the dynamics.

**Proposition A.24.** *Suppose that each player employs* GD *with a sufficiently small learning rate* $\eta > 0$ *and initialization* $(x_1^{(1)}, \ldots, x_n^{(1)}) \in \bigtimes_{i=1}^{n} \mathcal{X}_i$ *in a sequence of time-varying potential games. Then,*

$$\frac{1}{2\eta} \sum_{t=1}^{T} \sum_{i=1}^{n} \|x_i^{(t+1)} - x_i^{(t)}\|_2^2 \leq \sum_{t=1}^{T} \left( \Phi^{(t)}(x_1^{(t+1)}, \ldots, x_n^{(t+1)}) - \Phi^{(t)}(x_1^{(t)}, \ldots, x_n^{(t)}) \right). \quad (47)$$

This bound can be derived from [2, Theorem 4.3]. Now, we note that if $\Phi^{(1)} = \Phi^{(2)} = \cdots = \Phi^{(T)}$, the right-hand side of (47) telescops, thereby implying that the second-order path-length is bounded. More generally, the right-hand side of (47) can be upper bounded by

$$2\Phi_{\max} + \sum_{t=1}^{T-1}\left(\Phi^{(t)}(\boldsymbol{x}_1^{(t+1)},\ldots,\boldsymbol{x}_n^{(t+1)}) - \Phi^{(t+1)}(\boldsymbol{x}_1^{(t+1)},\ldots,\boldsymbol{x}_n^{(t+1)})\right) \leq 2\Phi_{\max} + \mathcal{V}_{\Phi}^{(T)}, \quad (48)$$

where $\Phi_{\max}$ is an upper bound on $|\Phi^{(t)}(\cdot)|$ for any $t \in [\![T]\!]$, and $\mathcal{V}_{\Phi}^{(T)}$ is the variation measure of the potential functions we introduced in Section 3.3. Namely, $\mathcal{V}_{\Phi}^{(T)} := \sum_{t=1}^{T-1} d(\Phi^{(t)}, \Phi^{(t+1)})$, where $d : (\Phi, \Phi') \mapsto \max_{\boldsymbol{z} \in \times_{i=1}^n \mathcal{X}_i} (\Phi(\boldsymbol{z}) - \Phi'(\boldsymbol{z}))$. Furthermore, similarly to Claim A.10, we know that the Nash equilibrium gap in the $t$-th potential game can be bounded in terms of $\sum_{i=1}^n \|\boldsymbol{x}_i^{(t+1)} - \boldsymbol{x}_i^{(t)}\|_2$. As a result, combining this property with Proposition A.24 and (48) establishes Theorem 3.7, the statement of which is recalled below.

**Theorem 3.7.** *Suppose that each player employs (online) GD with a sufficiently small learning rate in a sequence of time-varying potential games. Then,* $\sum_{t=1}^T \left(\text{EQGAP}^{(t)}(\boldsymbol{z}^{(t)})\right)^2 = O(\Phi_{max} + \mathcal{V}_{\Phi}^{(T)})$, *where* $\Phi_{max}$ *is such that* $|\Phi^{(t)}(\cdot)| \leq \Phi_{max}$.

### A.3.2 General-sum games

We next turn out attention to general-sum multi-player games in normal form using the well-known bilinear formulation of correlated equilibria (Section 3.3). More precisely, this zero-sum game is played between a *mediator*, who selects strategies from the set of correlated distributions $\Xi := \Delta(\times_{i=1}^n \mathcal{A}_i)$, and the $n$ players. Per the usual description of correlated equilibria, a mediator recommends to each player $i \in [\![n]\!]$ an action in $\mathcal{A}_i$, and then that player selects an action that could be different from the one recommended by the mediator. As a result, each pure strategy of Player $i$ corresponds to a different mapping from $\mathcal{A}_i$ to $\mathcal{A}_i$. Player $i$ wants to maximize the deviation benefit, while the mediator wants to minimize it. We let $\bar{\mathcal{X}}_i$ be the set of (mixed) strategies of Player $i$. The identity mapping $\boldsymbol{d}_i : \mathcal{A}_i \ni a_i \mapsto a_i$ will be referred to as the *direct* (or obedient) strategy, as it prescribes following the recommendation of the mediator. As a result, this game can be naturally cast as the bilinear saddle-point problem in the form of (4). By the existence of correlated equilibria [50], it follows that there exists a mediator strategy $\boldsymbol{\mu}^\star \in \Xi$ such that $\bar{\boldsymbol{x}}_i^\top \mathbf{A}_i^\top \boldsymbol{\mu}^\star \leq 0$, for any player $i \in [\![n]\!]$ and $\bar{\boldsymbol{x}}_i \in \bar{\mathcal{X}}_i$; in words, there is no benefit for any player to deviate from the recommendation of the mediator.

Now, to establish Property 3.8, let us first define the regret of any player $i \in [\![n]\!]$ as

$$\text{Reg}_i^{(T)}(\bar{\boldsymbol{x}}_i^\star) := \sum_{t=1}^T \langle \bar{\boldsymbol{x}}_i^\star - \bar{\boldsymbol{x}}_i^{(t)}, (\mathbf{A}_i^{(t)})^\top \boldsymbol{\mu}^{(t)} \rangle,$$

where $\bar{\boldsymbol{x}}_i^\star \in \bar{\mathcal{X}}_i$, so that $\sum_{i=1}^n \text{Reg}_i^{(T)}$ is easily seen to be equal to the regret of Player max in (4). Further, the dynamic regret of the mediator—Player min in (4)—can be expressed as

$$\text{DReg}_\mu^{(T)}(\boldsymbol{\mu}^{(1,\star)},\ldots,\boldsymbol{\mu}^{(T,\star)}) := \sum_{t=1}^T \langle \boldsymbol{\mu}^{(t)} - \boldsymbol{\mu}^{(t,\star)}, \sum_{i=1}^n \mathbf{A}_i^{(t)} \bar{\boldsymbol{x}}_i^{(t)} \rangle.$$

The key idea in the proof of Property 3.8 is that the *time-invariant* direct strategy $(\boldsymbol{d}_1,\ldots,\boldsymbol{d}_n)$ suffices to guarantee Property 3.1 as long as the mediator is selecting a sequence of CE.

**Property 3.8.** *Suppose that* $\Xi \ni \boldsymbol{\mu}^{(t,\star)}$ *is a correlated equilibrium of the game at any time* $t \in [\![T]\!]$. *Then,* $\text{DReg}_\mu^{(T)}(\boldsymbol{\mu}^{(1,\star)},\ldots,\boldsymbol{\mu}^{(T,\star)}) + \sum_{i=1}^n \text{Reg}_i^{(T)} \geq 0$.

*Proof.* We have that

$$\text{DReg}_\mu^{(T)} + \sum_{i=1}^n \text{Reg}_i^{(T)}(\bar{\boldsymbol{x}}_i^\star) = \sum_{i=1}^n \sum_{t=1}^T \langle \bar{\boldsymbol{x}}_i^\star, (\mathbf{A}_i^{(t)})^\top \boldsymbol{\mu}^{(t)} \rangle - \sum_{t=1}^T \langle \boldsymbol{\mu}^{(t,\star)}, \sum_{i=1}^n \mathbf{A}_i^{(t)} \bar{\boldsymbol{x}}_i^{(t)} \rangle.$$

Now for any correlated equilibrium $\boldsymbol{\mu}^{(t,\star)}$ of the $t$-th game, we have that $\langle \boldsymbol{\mu}^{(t,\star)}, \mathbf{A}_i^{(t)} \bar{\boldsymbol{x}}_i^{(t)} \rangle \leq 0$ for any player $i \in [\![n]\!]$, $\bar{\boldsymbol{x}}_i \in \bar{\mathcal{X}}_i$, and time $t \in [\![T]\!]$, which in turn implies that $-\sum_{t=1}^{T} \langle \boldsymbol{\mu}^{(t,\star)}, \sum_{i=1}^{n} \mathbf{A}_i^{(t)} \bar{\boldsymbol{x}}_i^{(t)} \rangle \geq 0$. Moreover, $\sum_{i=1}^{n} \max_{\bar{\boldsymbol{x}}_i^{\star} \in \bar{\mathcal{X}}_i} \sum_{t=1}^{T} \langle \bar{\boldsymbol{x}}_i^{\star}, (\mathbf{A}_i^{(t)})^{\top} \boldsymbol{\mu}^{(t)} \rangle \geq \sum_{i=1}^{n} \sum_{t=1}^{T} \langle \boldsymbol{d}_i, (\mathbf{A}_i^{(t)})^{\top} \boldsymbol{\mu}^{(t)} \rangle = 0$, by definition of the direct strategy $\boldsymbol{d}_i \in \bar{\mathcal{X}}_i$. This concludes the proof. $\qquad\square$

Property 3.8 in conjunction with the argument of Theorem 3.3 readily imply Theorem 3.9. We should note here that the variation measure $\mathcal{V}_{\mathbf{A}}^{(T)} := \sum_{i=1}^{n} \sum_{t=1}^{T-1} \|\mathbf{A}_i^{(t+1)} - \mathbf{A}_i^{(t)}\|_2^2$ is also immediately bounded in terms of the second-order variation of the sum of the players' utility tensors. It is also worth noting that the description of the bilinear formulation (4) grows exponentially with the number of players; this is unavoidable in explicitly represented (normal-form) games, but it is worth investigating whether more compact representations exist in succinct classes of games, such as multi-player polymatrix.

Next, we provide the main implication of Theorem 3.9 in the meta-learning setting, which is similar to the meta-learning guarantee of Proposition A.14 we established earlier in two-player zero-sum games. Below, we denote by $\Xi^{(h,\star)}$ the set of correlated equilibria of the $h$-th game in the meta-learning sequence.

**Corollary A.25** (Meta-learning in general games). *Suppose that each player employs* `OGD` *in* (4) *with a suitable learning rate $\eta > 0$ and the prediction of* (37) *in a meta-learning general-sum problem with $H \in \mathbb{N}$ games, each repeated for $m \in \mathbb{N}$ consecutive iterations. Then, for an average game,*

$$O\left( \frac{1}{\epsilon^2 H} + \frac{\mathcal{V}_{CE}^{(H)}}{\epsilon^2 H} \right) \tag{49}$$

*iterations suffice so that the mediator reaches an $\epsilon$-approximate correlated equilibrium, where*

$$\mathcal{V}_{CE}^{(H)} := \inf_{\boldsymbol{\mu}^{(h,\star)} \in \Xi^{(h,\star)}} \|\boldsymbol{\mu}^{(h+1,\star)} - \boldsymbol{\mu}^{(h,\star)}\|_2.$$

In particular, in the meta-learning regime, $H \gg 1$, the iteration-complexity bound (49) is dominated by the (algorithm-independent) similarity metric of the correlated equilibria $\frac{\mathcal{V}_{CE}^{(H)}}{H}$. Corollary A.25 establishes significant gains when $\frac{\mathcal{V}_{CE}^{(H)}}{H} \ll 1$.

Finally, we conclude this subsection by providing a variation-dependent regret bound in general-sum multi-player games. To do so, we combine Corollary A.12 with Theorem 3.9, leading to the following guarantee.

**Corollary A.26** (Regret in general-sum games). *In the setup of Theorem 3.9,*

$$\mathrm{Reg}_{\mu}^{(T)}, \mathrm{Reg}_i^{(T)} = O\left( \frac{1}{\eta} + \eta \left( 1 + \mathcal{V}_{CE}^{(T)} + \mathcal{V}_{\mathbf{A}}^{(T)} \right) \right),$$

*for any player $i \in [\![n]\!]$.*

In particular, if one selects optimally the learning rate, Corollary A.26 implies that the individual regret of each player is bounded by $O\left( \sqrt{1 + \mathcal{V}_{CE}^{(T)} + \mathcal{V}_{\mathbf{A}}^{(T)}} \right)$. We note again that there are techniques that would allow (nearly) recovering such regret guarantees without having to know the variation measures in advance [93].

## A.4 Proofs from Section 3.4

Finally, in this subsection we present the proofs omitted from Section 3.4. We begin with Proposition 3.10, the statement of which is recalled below. We first recall that a regularizer $\phi_x$, 1-strongly convex with respect to a norm $\|\cdot\|$, is said to be $G$-smooth if $\|\nabla\phi_x(\boldsymbol{x}) - \nabla\phi_x(\boldsymbol{x}')\|_* \leq G\|\boldsymbol{x} - \boldsymbol{x}'\|$, for all $\boldsymbol{x}, \boldsymbol{x}'$.

**Proposition 3.10.** *Suppose that both players in a (static) two-player zero-sum game employ* `OMD` *with a smooth regularizer. Then,* $\mathrm{DReg}_x^{(T)}, \mathrm{DReg}_y^{(T)} = O(\sqrt{T})$.

*Proof.* First, using Claim A.10, it follows that the dynamic regret $\mathrm{DReg}_x^{(T)}$ of Player $x$ up to time $T$ can be bounded as

$$\sum_{t=1}^{T} \left( \max_{\boldsymbol{x}^{(t,\star)} \in \mathcal{X}} \left\{ \langle \boldsymbol{x}^{(t,\star)}, \boldsymbol{u}_x^{(t)} \rangle \right\} - \langle \boldsymbol{x}^{(t)}, \boldsymbol{u}_x^{(t)} \rangle \right)$$
$$\leq \sum_{t=1}^{T} \left( \left( \frac{GD_{\mathcal{X}}}{\eta} + \|\boldsymbol{u}_x^{(t)}\|_* \right) \|\boldsymbol{x}^{(t)} - \hat{\boldsymbol{x}}^{(t+1)}\| + \frac{GD_{\mathcal{X}}}{\eta} \|\boldsymbol{x}^{(t)} - \hat{\boldsymbol{x}}^{(t)}\| \right), \quad (50)$$

where $G > 0$ is the smoothness parameter of the regularizer, and $\eta > 0$ is the learning rate. We further know that $\sum_{t=1}^{T} \left( \|\boldsymbol{x}^{(t)} - \hat{\boldsymbol{x}}^{(t)}\|^2 + \|\boldsymbol{x}^{(t)} - \hat{\boldsymbol{x}}^{(t+1)}\|^2 \right) = O(1)$ for any instance of `OMD` in a two-player zero-sum game [2], which in turn implies that $\sum_{t=1}^{T} \left( \|\boldsymbol{x}^{(t)} - \hat{\boldsymbol{x}}^{(t)}\| + \|\boldsymbol{x}^{(t)} - \hat{\boldsymbol{x}}^{(t+1)}\| \right) = O(\sqrt{T})$ by Cauchy-Schwarz. Thus, combining with (50) we have shown that $\mathrm{DReg}_x^{(T)} = O(\sqrt{T})$. Similar reasoning yields that $\mathrm{DReg}_y^{(T)} = O(\sqrt{T})$, concluding the proof. $\qquad \square$

Let us next make a certain refinement of Observation 3.11. Staying on (static) bilinear saddle-point problems, we consider the unconstrained setting where $\mathcal{X} = \mathbb{R}^{d_x}$ and $\mathcal{Y} = \mathbb{R}^{d_y}$, and we focus on the following update rule for $\tau \in \mathbb{N}$.

$$\begin{aligned} \boldsymbol{x}^{(\tau+1)} &= \boldsymbol{x}^{(\tau)} - 2\eta \mathbf{A} \boldsymbol{y}^{(\tau)} + \eta \mathbf{A} \boldsymbol{y}^{(\tau-1)}, \\ \boldsymbol{y}^{(\tau+1)} &= \boldsymbol{y}^{(\tau)} + 2\eta \mathbf{A}^\top \boldsymbol{x}^{(\tau)} - \eta \mathbf{A}^\top \boldsymbol{x}^{(\tau-1)}, \end{aligned} \quad (51)$$

where $\boldsymbol{x}^{(0)}, \boldsymbol{x}^{(-1)} \in \mathcal{X}$ and $\boldsymbol{y}^{(0)}, \boldsymbol{y}^{(-1)} \in \mathcal{Y}$. This is an instance of *optimistic follow the regularized leader (`OFTRL`)* [78], although our discussion here also covers the corresponding `OMD` variant. Then, summing (51) for $\tau = 1, 2, \ldots, t$ we have

$$\begin{aligned} \sum_{\tau=1}^{t} \boldsymbol{x}^{(\tau+1)} &= \sum_{\tau=1}^{t} \boldsymbol{x}^{(\tau)} - 2\eta \mathbf{A} \left( \sum_{\tau=1}^{t} \boldsymbol{y}^{(\tau)} \right) + \eta \mathbf{A} \left( \sum_{\tau=1}^{t} \boldsymbol{y}^{(\tau-1)} \right), \\ \sum_{\tau=1}^{t} \boldsymbol{y}^{(\tau+1)} &= \sum_{\tau=1}^{t} \boldsymbol{y}^{(\tau)} + 2\eta \mathbf{A}^\top \left( \sum_{\tau=1}^{t} \boldsymbol{x}^{(\tau)} \right) - \eta \mathbf{A}^\top \left( \sum_{\tau=1}^{t} \boldsymbol{x}^{(\tau-1)} \right), \end{aligned}$$

in turn implying that

$$\begin{aligned} \bar{\boldsymbol{x}}^{(t+1)} &= \frac{1}{t+1} \bar{\boldsymbol{x}}^{(1)} + \eta \mathbf{A} \frac{1}{t+1} \boldsymbol{y}^{(0)} + \frac{t}{t+1} \bar{\boldsymbol{x}}^{(t)} - 2\eta \frac{t}{t+1} \mathbf{A} \bar{\boldsymbol{y}}^{(t)} + \eta \frac{t-1}{t+1} \mathbf{A} \bar{\boldsymbol{y}}^{(t-1)}, \\ \bar{\boldsymbol{y}}^{(t+1)} &= \frac{1}{t+1} \bar{\boldsymbol{y}}^{(1)} - \eta \mathbf{A}^\top \frac{1}{t+1} \boldsymbol{x}^{(0)} + \frac{t}{t+1} \bar{\boldsymbol{y}}^{(t)} + 2\eta \frac{t}{t+1} \mathbf{A}^\top \bar{\boldsymbol{x}}^{(t)} + \eta \frac{t-1}{t+1} \mathbf{A}^\top \bar{\boldsymbol{x}}^{(t-1)}. \end{aligned} \quad (52)$$

Above, we define $\bar{\boldsymbol{x}}^{(t)} \coloneqq \frac{1}{t} \sum_{\tau=1}^{t} \boldsymbol{x}^{(\tau)}$ and $\bar{\boldsymbol{y}}^{(t)} \coloneqq \frac{1}{t} \sum_{\tau=1}^{t} \boldsymbol{y}^{(\tau)}$. That is, the update rule (52) describes the evolution of the time average of (51). As a result, this implies that (52) converges *in a last-iterate sense* with a $T^{-1}$ rate, simply because `OFTRL` incurs $O(1)$ regret [78]. This is conceptually interesting as it suggests a simple way to analyze the convergence of the last iterate solely through a regret-based analysis. In fact, (52) is a variant of the so-called *Halpern's iteration* [32]—a classical technique in optimization—that incorporates optimism; a similar update rule was recently shown to exhibit a new form of acceleration by Yoon and Ryu [90], and has engendered considerable interest ever since. Returning to Observation 3.11, one can use (52) to guarantee $O(\log T)$ dynamic regret in the standard feedback model, although it is not clear how to extend this argument in the constrained setting (*cf.* [14]).

**General-sum games** We next extend our scope beyond bilinear saddle-point problems. We first observe that obtaining sublinear dynamic regret is precluded in general-sum games. In particular, we note that the computational-hardness result below (Proposition 3.12) holds beyond the online learning setting. It should be stressed that, at least in normal-form games, without imposing computational or memory restrictions there are trivial online algorithms that guarantee even $O(1)$ dynamic regret by first exploring the payoff tensors and then computing a Nash equilibrium [26]. We suspect that under the memory limitations imposed by Daskalakis et al. [26] there could be unconditional information-theoretic lower bounds, but that is left for future work.

**Proposition 3.12.** *Any polynomial-time algorithm incurs* $\sum_{i=1}^{n} \mathrm{DReg}_i^{(T)} \geq CT$ *for any polynomial* $T \in \mathbb{N}$, *even if* $n = 2$ *and* $C > 0$ *is an absolute constant, unless ETH for* PPAD *is false [73].*

*Proof.* We will make use of the fact that computing a Nash equilibrium in two-player (normal-form) games to a sufficiently small accuracy $\epsilon = O(1)$ requires superpolynomial time, unless the exponential-time hypothesis for PPAD fails [73]. Indeed, suppose that there exist polynomial-time algorithms that always guarantee that $\sum_{i=1}^{n} \mathrm{DReg}_i^{(T)} \leq \epsilon T$ for some polynomial $T \in \mathbb{N}$, where $n := 2$. Then, this implies that there exists a time $t \in [\![T]\!]$ such that

$$\max_{\boldsymbol{x}_1^{(t,\star)} \in \mathcal{X}_1} \langle \boldsymbol{x}_1^{(t,\star)}, \boldsymbol{u}_1^{(t)} \rangle - \langle \boldsymbol{x}_1^{(t)}, \boldsymbol{u}_1^{(t)} \rangle + \max_{\boldsymbol{x}_2^{(t,\star)} \in \mathcal{X}_2} \langle \boldsymbol{x}_2^{(t,\star)}, \boldsymbol{u}_2^{(t)} \rangle - \langle \boldsymbol{x}_2^{(t)}, \boldsymbol{u}_2^{(t)} \rangle \leq \epsilon,$$

which in turn implies that $(\boldsymbol{x}_1^{(t)}, \boldsymbol{x}_2^{(t)})$ is an $\epsilon$-approximate Nash equilibrium. Further, such a time $t \in [\![T]\!]$ can be identified in polynomial time since $T \leq \mathrm{poly}(|\mathcal{A}_1|, |\mathcal{A}_2|)$. This concludes the proof. $\qquad\square$

We also note the following computational hardness result, which rests on the more standard complexity assumption that PPAD is not contained in P. As in Proposition 3.12, below we tacitly assume that the underlying algorithm is deterministic.

**Proposition A.27.** *Unless* PPAD $\subseteq$ P, *no polynomial-time algorithm guarantees* $\sum_{i=1}^{n} \mathrm{DReg}_i^{(T)} \leq \mathrm{poly}(|\mathcal{A}_1|, |\mathcal{A}_2|)T^{1-\omega}$ *for all* $T \in \mathbb{N}$, *where* $\omega \in (0, 1]$ *is an absolute constant, even if* $n = 2$.

The proof follows directly from the PPAD-hardness result of Chen et al. [19]. Of course, if we do not operate in the online learning model, it is computationally trivial to come up with algorithms that guarantee that one of the two players will have $0$ dynamic regret by simply best responding to the strategy of the other player.

Finally, we provide the proof of Theorem 3.13, the detailed version of which is provided below.

**Theorem A.28** (Detailed version of Theorem 3.13). *Consider an* $n$-*player game such that* $\|\nabla_{\boldsymbol{x}_i} u_i(\boldsymbol{z}) - \nabla_{\boldsymbol{x}_i} u_i(\boldsymbol{z}')\|_2 \leq L\|\boldsymbol{z} - \boldsymbol{z}'\|_2$, *where* $\boldsymbol{z}, \boldsymbol{z}' \in \bigtimes_{i=1}^{n} \mathcal{X}_i$, *for any player* $i \in [\![n]\!]$. *Then, if all players employ* OGD *with learning rate* $\eta > 0$ *it holds that*

1. $\sum_{i=1}^{n} K\text{-}\mathrm{DReg}_i^{(T)} = O(K\sqrt{n}LD_{\mathcal{Z}}^2)$ *if* $\eta = \Theta\left(\frac{1}{L\sqrt{n}}\right)$;

2. $K\text{-}\mathrm{DReg}_i^{(T)} = O(K^{3/4}T^{1/4}n^{1/4}L^{1/2}D_{\mathcal{X}_i}^{3/2})$, *for any* $i \in [\![n]\!]$, *if* $\eta = \Theta\left(\frac{K^{1/4}D_{\mathcal{X}_i}^{1/2}}{n^{1/4}L^{1/2}T^{1/4}}\right)$.

*Proof.* First, applying Lemma A.1 subject to the constraint that $\sum_{t=1}^{T-1} \mathbb{1}\{\boldsymbol{x}_i^{(t+1,\star)} \neq \boldsymbol{x}_i^{(t,\star)}\} \leq K - 1$ gives that for any Player $i \in [\![n]\!]$,

$$K\text{-}\mathrm{DReg}_i^{(T)} \leq \frac{D_{\mathcal{X}_i}^2}{2\eta}(2K-1) + \eta\|\boldsymbol{u}_i^{(1)}\|_2^2 + \eta\sum_{t=1}^{T-1}\|\boldsymbol{u}_i^{(t+1)} - \boldsymbol{u}_i^{(t)}\|_2^2$$
$$- \frac{1}{4\eta}\sum_{t=1}^{T-1}\|\boldsymbol{x}_i^{(t+1)} - \boldsymbol{x}_i^{(t)}\|_2^2. \qquad (53)$$

Further, by $L$-smoothness we have that

$$\|\boldsymbol{u}_i^{(t+1)} - \boldsymbol{u}_i^{(t)}\|_2^2 = \|\nabla_{\boldsymbol{x}_i} u_i(\boldsymbol{z}^{(t+1)}) - \nabla_{\boldsymbol{x}_i} u_i(\boldsymbol{z}^{(t)})\|_2^2 \leq L^2\sum_{i=1}^{n}\|\boldsymbol{x}_i^{(t+1)} - \boldsymbol{x}_i^{(t)}\|_2^2,$$

for any $t \in [\![T-1]\!]$, where $(\boldsymbol{x}_1^{(t)}, \dots, \boldsymbol{x}_n^{(t)}) = \boldsymbol{z}^{(t)} \in \bigtimes_{i=1}^{n} \mathcal{X}_i$ is the joint strategy profile at time $t$. Thus, summing (53) over all $i \in [\![n]\!]$ and taking $\eta \leq \frac{1}{2L\sqrt{n}}$ implies that $\sum_{i=1}^{n} K\text{-}\mathrm{DReg}_i^{(T)} \leq \frac{2K-1}{2\eta}\sum_{i=1}^{n} D_{\mathcal{X}_i}^2 + \eta\sum_{i=1}^{n}\|\boldsymbol{u}_i^{(1)}\|_2^2$, yielding the first part of the statement since $D_{\mathcal{Z}}^2 = \sum_{i=1}^{n} D_{\mathcal{X}_i}^2$, where we recall the notation $\mathcal{Z} := \bigtimes_{i=1}^{n} \mathcal{X}_i$. The second part follows directly from (53) using the stability property of OGD: $\|\boldsymbol{x}_i^{(t+1)} - \boldsymbol{x}_i^{(t)}\|_2 = O(\eta)$, for any time $t \in [\![T-1]\!]$. $\qquad\square$

*Remark* A.29. It is not hard to show that Item 2 above can be improved to $O_{K,T}(K)$ using *clairvoyant mirror descent (CMD)* [68], but under a stronger feedback model. In particular, using the (squared) Euclidean regularizer it is direct to show (see [39, 65]) that the dynamic regret of CMD is bounded solely by the first-order variation of the sequence of comparators, which suffices for this claim. It is open whether similar results can be obtained in the standard feedback model.

# B  Experimental examples

Finally, although the focus of this paper is theoretical, in this section we provide some illustrative experimental examples. In particular, Appendix B.1 contains experiments on time-varying potential games, while Appendix B.2 focuses on time-varying (two-player) zero-sum games. For simplicity, we will be assuming that each game is represented in normal form.

## B.1  Time-varying potential games

Here we consider time-varying 2-player identical-interest games. We point out that such games are potential games (recall Definition A.23), and as such, they are indeed amenable to our theory in Section 3.3.

In our first experiment, we first sampled two matrices $\mathbf{A}, \mathbf{P} \in \mathbb{R}^{d_x \times d_y}$, where $d_x = d_y = 1000$. Then, we defined each payoff matrix as $\mathbf{A}^{(t)} := \mathbf{A}^{(t-1)} + \mathbf{P}t^{-\alpha}$ for $t \geq 1$, where $\mathbf{A}^{(0)} := \mathbf{A}$. Here, $\alpha > 0$ is a parameter that controls the variation of the payoff matrices. In this time-varying setup, we let each player employ (online) GD with learning rate $\eta := 0.1$. The results obtained under different random initializations of matrices $\mathbf{A}$ and $\mathbf{P}$ are illustrated in Figure 1.

Next, we operate in the same time-varying setup but each player is now employing multiplicative weights update (MWU), instead of gradient descent, with $\eta := 0.1$. As shown in Figure 2, while the cumulative equilibrium gap is much larger compared to using GD (Figure 1), the dynamics still appear to be approaching equilibria, although our theory does not cover MWU. We suspect that theoretical results such as Theorem 3.7 should hold for MWU as well, but that has been left for future work.

In our third experiment for identical-interest games, we again first sampled two matrices $\mathbf{A}, \mathbf{P} \in \mathbb{R}^{d_x \times d_y}$, where $d_x = d_y = 1000$. Then, we defined $\mathbf{A}^{(t)} := \mathbf{A}^{(t-1)} + \epsilon\mathbf{P}$ for $t \geq 1$, where $\mathbf{A}^{(0)} := \mathbf{A}$. Here, $\epsilon > 0$ is the parameter intended to capture the variation of the payoff matrices. The results obtained under different random initializations of $\mathbf{A}$ and $\mathbf{P}$ are illustrated in Figure 3. As an aside, it is worth pointing out that this particular setting can be thought of as a game in which the variation in the payoff matrices is controlled by another learning agent. In particular, our theoretical results could be helpful for characterizing the convergence properties of *two-timescale* learning algorithms, in which the deviation of the game is controlled by a player constrained to be updating its strategies with a much smaller learning rate.

## B.2  Time-varying zero-sum games

We next conduct experiments on time-varying bilinear saddle-point problems when players are employing OGD. Such problems were studied extensively earlier in Section 3.1 from a theoretical standpoint.

First, we sampled two matrices $\mathbf{A}, \mathbf{P} \in \mathbb{R}^{d_x \times d_y}$, where $d_x = d_y = 10$; here we consider lower-dimensional payoff matrices compared to the experiments in Appendix B.1 for convenience in the graphical illustrations. Then, we defined each payoff matrix as $\mathbf{A}^{(t)} := \mathbf{A}^{(t-1)} + \mathbf{P}t^{-\alpha}$ for $t \geq 1$, where $\mathbf{A}^{(1)} := \mathbf{A}$. The results obtained under different random initializations are illustrated in Figure 4.

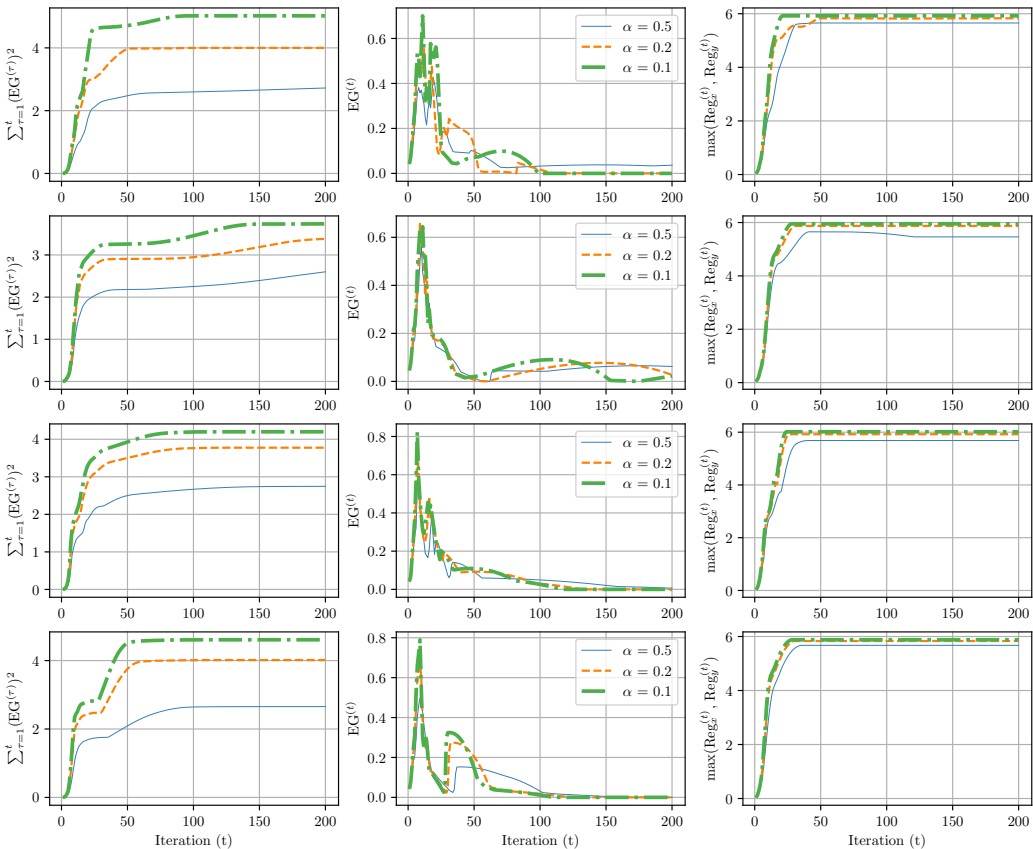

Figure 1: The equilibrium gap and the players' regrets in 2-player time-varying identical-interest games when both players are employing (online) GD with learning rate $\eta := 0.1$ for $T := 200$ iterations. Each row corresponds to a different random initialization of the matrices $\mathbf{A}, \mathbf{P} \in \mathbb{R}^{d_x \times d_y}$, which in turn induces a different time-varying game. Further, each figure contains trajectories corresponding to three different values of $\alpha \in \{0.1, 0.2, 0.5\}$, but under the same initialization of $\mathbf{A}$ and $\mathbf{P}$. As expected, smaller values of $\alpha$ generally increase the equilibrium gap since the variation of the games is more significant. Nevertheless, for all games we observe that the players are gradually approaching equilibria.

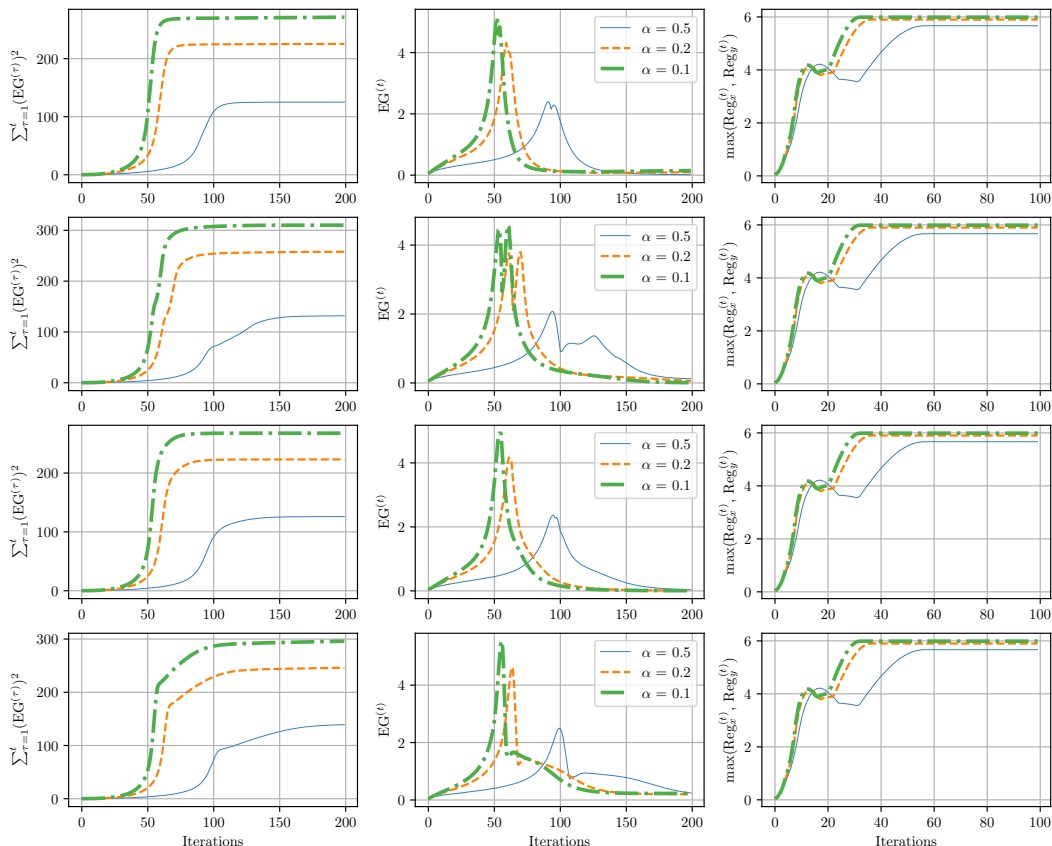

Figure 2: The equilibrium gap and the players' regrets in 2-player time-varying identical-interest games when both players are employing (online) GD with learning rate $\eta := 0.1$ for $T := 200$ iterations. Each row corresponds to a different random initialization of the matrices $\mathbf{A}, \mathbf{P} \in \mathbb{R}^{d_x \times d_y}$, which in turn induces a different time-varying game. Further, each figure contains trajectories corresponding to three different values of $\alpha \in \{0.1, 0.2, 0.5\}$, but under the same initialization of $\mathbf{A}$ and $\mathbf{P}$. The MWU dynamics still appear to be approaching equilibria, although the cumulative gap is much larger compared to GD (Figure 1).

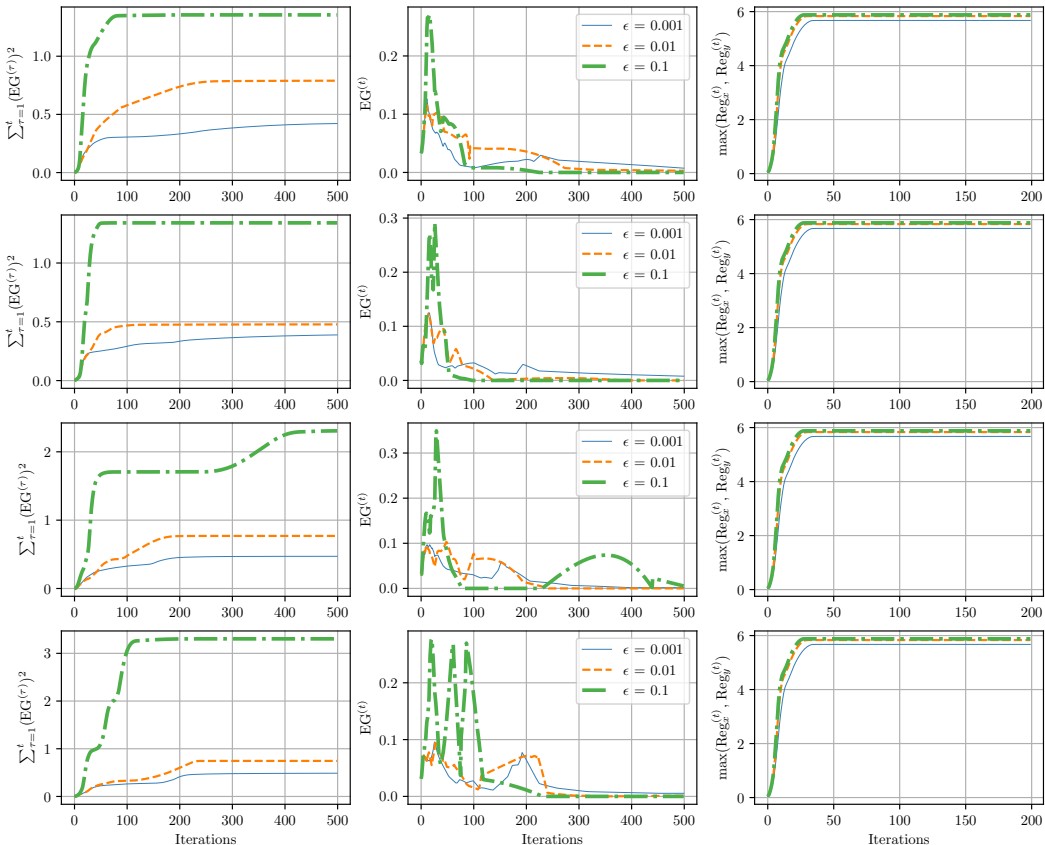

Figure 3: The equilibrium gap and the players' regrets in 2-player time-varying identical-interest games when both players are employing (online) GD with learning rate $\eta := 0.1$ for $T := 500$ iterations. Each row corresponds to a different random initialization of the matrices $\mathbf{A}, \mathbf{P} \in \mathbb{R}^{d_x \times d_y}$, which in turn induces a different time-varying game. Further, each figure contains trajectories from three different values of $\epsilon \in \{0.1, 0.01, 0.001\}$, but under the same initialization of $\mathbf{A}$ and $\mathbf{P}$. As expected, larger values of $\epsilon$ generally increase the equilibrium gap since the variation of the games is more significant. Yet, even for the larger value $\epsilon = 0.1$, the dynamics are still appear to be approaching Nash equilibria.

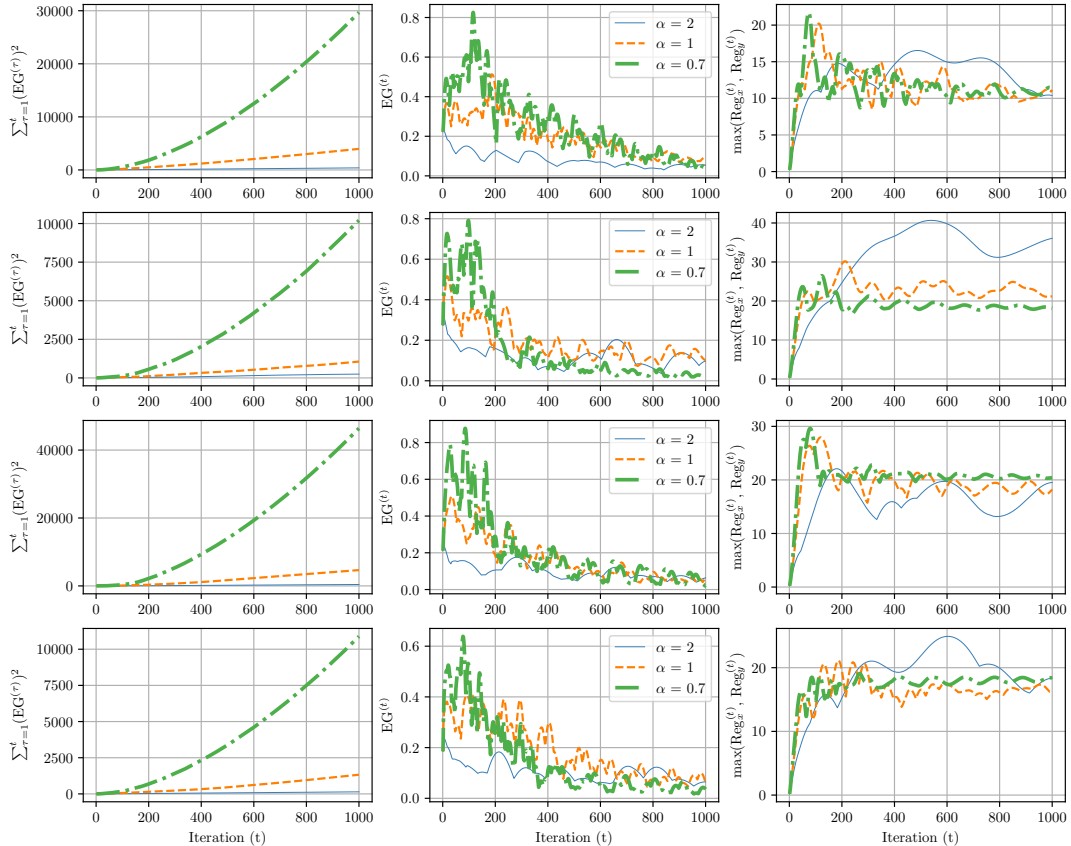

Figure 4: The equilibrium gap and the players' regrets in 2-player time-varying zero-sum games when both players are employing `OGD` with learning rate $\eta := 0.01$ and $T := 1000$ iterations. Each row corresponds to a different random initialization of the matrices $\mathbf{A}, \mathbf{P} \in \mathbb{R}^{d_x \times d_y}$, which in turn induces a different time-varying game. Further, each figure contains trajectories from three different values of $\alpha \in \{0.7, 1, 2\}$, but under the same initialization of $\mathbf{A}$ and $\mathbf{P}$. The `OGD` dynamics appear to be approaching equilibria, albeit with a much slower rate compared to the ones observed earlier for potential games (Figure 1).

