# OpenReview forum: "On the Convergence of No-Regret Learning Dynamics in Time-Varying Games"
_NeurIPS.cc/2023/Conference — NeurIPS 2023 poster_

### Official Review · Reviewer_3wR1 · 2023-06-12

**Soundness:** 3 good
**Presentation:** 3 good
**Contribution:** 2 fair
**Rating:** 4
**Confidence:** 4

**Summary:**

This paper considers the problem of online learning in time-varying games under different setups. Specifically, the authors consider the case where all the players apply optimistic gradient descent (OGD) algorithm with a certain choice of learning rate. The main results that in the two-player bilinear game setup, the sum of squared duality gap is bounded by $O(1+V_{\epsilon-NE}+V_A)$, recovering the best-known result in the stationary game setup. This result is based on the bounded second-order path length of the learning dynamic and the important observation that sum of dynamic regret with respect to (approximated) Nash is (almost) non-negative. Next, they extend this result to the strongly convex-concave game with multiple steps and obtain similar duality gap guarantees. The author also consider the potential game and general-sum game setup with guarantees on the duality gap and CEgap bound respectively. Experiments are also done to support their theoretical results.

**Strengths:**

- The problem considered in this paper is important and the authors show that the classic OGD algorithm achieves desirable average duality-gap and other equilibrium-related gap bound with provable guarantees.
- The authors also do experiments on time-varying potential games and time-varying zero-sum games to verify their obtained NEgap bound.

**Weaknesses:**

The main concern is the novelty of this paper. Specifically, this paper shares similarity to the results in [60] in many perspectives, although the authors explain in many places in the paper that how their results are different from [60].
- One of the main lemma Lemma A.1 is the same as the DRVU property shown in [60].
- Property A.3 is very similar to Eq.(32) shown in [60].
- Example shown in Proposition 3.2 is almost the same as the example shown in Appendix C, case 2 in [60].
- From the technical perspective, I feel like the analysis is very similar to the one shown in [60]. While [60] does not consider the approximated NE path-length, it is not hard to extend their analysis to the approximated NE path-length by replacing P_T with \epsilon-NE path-length +T\epsilon. Also, the boundedness of the second-order dynamic is also shown in Lemma 16/18 in [60].
- In addition, as mentioned by the authors, there are parameter-tuning issues in achieving better individual regret guarantees in bilinear and  strongly convex-concave games, which is also handled by the meta-base structure proposed in [60].

The authors also derive results for general-sum games and potential games. Given Property 3.8 that the sum of regret with respect to CE is positive, it seems that the average CEgap bound is also not very hard to obtain given the bounded second-order dynamic of OGD.

**Questions:**

- As mentioned in the above section, can the authors explain more in detail on the main technical difficulties and challenges compared to the analysis shown in [60].
- In Property 3.1, the authors argue that this extends the exact NE path-length in [60] to the approximated NE path-length. In [60], they also provide a bound with respect to W_T, which is the variance of the game matrix. Can the authors explain more on the comparison between the epsilon-NE path-length and W_T? Is there a case where both the exact NE path-length and the W_T are large but the epsilon-NE path-length is very small?


**Limitations:**

See Weakness section.

---

> ### Author Rebuttal · Authors · 2023-08-08
>
> We are grateful to the reviewer for their constructive feedback. Below, we stress the key differences between our results and the ones in [60].
>
> Starting from Section 3.1, we indeed build on a number of ingredients from [60], as we carefully acknowledge throughout the paper; this includes the dynamic RVU bound (Lemma A.1) and using second-order path lengths, as the reviewer points out. We stress, however, that using RVU-type bounds and second-order path lengths is very standard in this line of work, so we do not believe that those similarities weaken our contribution. Indeed, we provide a number new insights that are of interest, leading to our main result in Theorem 3.3 that is different from the results in [60] in many aspects. First, we use a variation measure depending on an approximate sequence of NE; note that while the example in Proposition 3.2 is similar to that in Appendix C of [60], as the reviewer points out, it serves a different purpose in our context. Furthermore, we connect nonnegativity of dynamic regret with the MVI property from variational inequalities. This allows extensions to settings such as polymatrix zero-sum games and convex-concave games; given the tremendous interest such settings have received, we believe that those extensions are important. Our results are also based on a simpler algorithm: simply run optimistic gradient descent (OGD)—or variants of optimistic mirror descent—with a time-invariant (constant) learning rate. We believe that this has an independent interest given how well-studied those algorithms are in the static setting, and it is also worth noting that several prior papers have motivated using a constant learning rate—which has not been done in this context. In contrast, the algorithm of [60] has further layers of complications, which are of course there to handle issues such as parameter tuning; while such issues are crucial in the setup of [60] in order to minimzie regret, they are not present in our setting precisely because our focus is different. Namely, our focus (in Section 3.1) is to characterize the equilibrium gap of OGD.
>
> We also want to highlight that our results and our technical approach beyond Section 3.1, namely in Sections 3.2-3.4, are in general different from those in [60]; this includes time-varying potential games, general-sum games, and strongly convex-concave settings. Each of those settings introduces their own challenges that are in general different from the ones encountered for bilinear saddle-point problems.
>
> So, despite a number of similarities with [60] that we carefully point out throughout the paper, most of those similarities are technical tools used consistently in this line of work (such as RVU-type properties and second-order path lengths), and we do not believe that they weaken our contribution. We believe that we provide a number of new insights beyond what was known in prior work, and our results are overall complementary with [60].
>
> We finally answer the reviewer's following question.
>
> *“Is there a case where both the exact NE path-length and the $W_T$ are large but the epsilon-NE path-length is very small?”*
>
> Yes. Consider the sequence of matrices $A^{(1)}, \dots, A^{(T)}$ provided in Proposition 3.2. Now let us instead take the sequence $A^{(1)} + c^{(1)} I, \dots, A^{(T)} + c^{(T)} I$, where $c^{(1)}, \dots, c^{(T)} \in \mathbb{R}$ and $I$ is the all-ones matrix. It is clear that the variation measures that depend on the Nash equilibrium are exactly the same no matter how $c^{(1)}, \dots, c^{(T)}$ are chosen. On the other hand, by suitably selecting the sequence of $c^{(1)}, \dots, c^{(T)}$ we can make $W_T$ to be arbitrarily large. This is a quite trivial example, but demonstrates the volatility of $W_T$ as a variation measure since it is not robust to transformations that retain (approximate) Nash equilibria.

---

> > ### Comment · Reviewer_3wR1 · 2023-08-11
> > **Thanks for the authors' response**
> >
> > Thanks for the authors' response on my questions and the comparison between the submission and [60]. However, I am still not convinced the technical contribution of this submission compared with [60].
> >
> > - While the authors argue that "using RVU-type bounds and second-order path lengths is very standard in this line of work", the main theorem (Theorem 3.1) of this submission is by directly combining the dynamic RVU-type bound proposed in [60] and the property of (almost) non-negative sum-of-regret property, which has already been shown in [60] (see Equation (32) of [60]) and section 3.1 of [Anagnostides et al., 2022]. As mentioned, the only difference I think is that [60] consider the exact NE divergence but the authors consider the approximate NE divergence, which is not hard to be obtained by extending the analysis in [60] by replacing P_T with \epsilon-NE path-length +T\epsilon following in Lemma 16/18 in [60]. From the technical perspective, the extension to poly-matrix zero-sum games, convex-concave games and the general sum game (using CE) is not complicated based on the (almost) non-negative sum-of-regret property as shown in proposition 3.2 in [Anagnostides et al., 2022].
> >
> > - While the author argued on the choice of \eta, I believe in [60], if the measure is only for equilibrium gap (duality gap in [60]), constant learning rate is also enough to prove the results as I think using a meta-base algorithm design in [60] is mainly about adapting to different metrics. Also as mentioned in Section A.1.5, to achieve better regret bound, an adaptive tuning method proposed in [60] is necessary. Then this is exactly what is proposed in [60].
> >
> > Based on the above, I am not convinced that the contribution is significant.
> >
> > [Anagnostides et al., 2022] On Last-Iterate Convergence Beyond Zero-Sum Games, ICML 2022

---

> > > ### Author Response · Authors · 2023-08-15
> > >
> > > We thank the reviewer for the constructive feedback.
> > >
> > > We first want to point out that the paper of Anagnostides et al. referenced by the reviewer deals solely with static games; unlike [60], it does not rely on the nonnegativity of dynamic regret, which is an important ingredient in the dynamic setting. All similarities of our approach with [60]-including all points made by the reviewer above-are already carefully explained throughout the paper, especially in the subsection describing our contributions.
> > >
> > > To address the reviewer's point, many of the extensions we provide, including using an improved variation measure based on approximate Nash equilibria, time-varying variational inequalities based on the MVI property, and correlated equilibria, are not complicated, but do require combining multiple suitable ingredients, which we believe to be a valuable contribution. And in any event those are new results concerning well-studied problems not derived in prior work, so we consider them to be an important addition to the existing literature. For example, the bilinear formulation of correlated equilibria we employ is definitely not standard, which is another reason why that important setting was not addressed in prior work. Our results also cover time-varying potential games and time-varying strongly convex-concave games, which depart considerably from [60].

---

### Official Review · Reviewer_jL2o · 2023-07-03

**Soundness:** 4 excellent
**Presentation:** 3 good
**Contribution:** 3 good
**Rating:** 7
**Confidence:** 3

**Summary:**

In this work the authors consider no-regret learning in multiagent games where the underlying game varies across different rounds. They study several classes of games and various learning algorithms that the agents can use. Naturally, the results they obtained are parametrized by variation measures of the underlying game that the agents participate in. To be more precise:
* For time varying zero-sum games, they focus on the setting where both of the agents are using optimistic gradient descent (OGD),  which is a variant of gradient descent that puts a bias on more recent rounds of the game. Interestingly, they show that almost all iterates of OGD are approximate Nash equilibria provided that some variation measures related to changes in the set of approximate equilibria of the games and the underlying payoff matrices are $o(T)$.
* Then, they consider sequences of games where the games in each round are strongly convex-concave. For this class of games, they are able to show a similar result as above, but under weaker variation conditions for the underlying sequence of games.
* Finally, they consider time-varying general-sum games. Naturally, since Nash equilibria are not tractable in this setting, they consider convergence to correlated equilibria. They prove similar results as above, but now the variation measure they use is related to the set of correlated equilibria of the game.
Their results have implications to other settings as well such as meta-learning and dynamic regret guarantees in static games.

**Strengths:**

* The paper studies a very natural problem and provides strong results under various settings of interest, which also have implications in other settings, as I mentioned in the summary.
* For the most part, the paper is easy to follow and the authors have done a good job placing their work in the literature.
* The authors are not trying to oversell the proof technique, which is heavily inspired by prior works, but uses some natural and clever modifications. For example, instead of letting the variation measure to depend on variations of the set of exact Nash equilibria which would make the problem very difficult to handle since this set is very sensitive to any changes of the payoff matrices, the authors consider variations of the set of approximate Nash equilibria, which behaves much nicer. Since the results are strong and general, I don't think that the authors should be penalized for the fact that the proof techniques are not very novel.


**Weaknesses:**

* Some parts of the paper might be a bit hard to follow for non-experts, especially in Section 1.1. For example, the MVI property and the RVU bound were not defined. I think the authors could make the transition to this section a bit smoother, although I understand that the space limitations are making it trickier.
* Even though the variation measures the authors use are intuitive and it makes sense that the regret should scale with these quantities, there are no lower bounds to show the extent to which these results are optimal.


Some minor comments:
* In Proposition 3.10, it might be useful to state which dynamic benchmark you consider for their dynamic regret bound.
* With this bibliography style it is a bit hard to keep track of the references, although I understand that it saves some valuable space.

**Questions:**

* What are the technical challenges to generalize the results to the bandit or some other partial feedback model?
* Would there be any benefit if you considered a variational measure wrt approximate correlated equilibria in general sum games instead of exact correlated equilibria?
* To what extent do you think that the results are tight?
* Do the results for two-player zero-sum games generalize to multiagent zero-sum polymatrix games? I don't see any inherent obstacles to do that using your approach, but I might be missing something.
* Another class of general sum games that is tractable in the single-shot setting are games in which the underlying matrices are rank-1. Is there any hope to obtain similar results for this class of games? I would imagine that the techniques would need to be substantially different from your approach.
* In Theorem 3.3 (and similar results) if the parameter $L$ is not known does the usual guess-and-double trick work to get the bound?

---

> ### Author Rebuttal · Authors · 2023-08-08
>
> We are grateful to the reviewer for their feedback.
>
> *“Some parts of the paper might be a bit hard to follow for non-experts, especially in Section 1.1.”*
>
> We will make sure to introduce further background in the revised version of the introduction.
>
> *“What are the technical challenges to generalize the results to the bandit or some other partial feedback model?”*
>
> An important challenge beyond the full-feedback setting is that it is not known whether RVU-type properties hold, which is crucial for our analysis; see, for example, the paper “More Adaptive Algorithms for Adversarial Bandits” by Luo and Wei. So it seems that an entirely different approach is needed beyond the full-feedback setting.
>
> *“Would there be any benefit if you considered a variational measure wrt approximate correlated equilibria in general sum games instead of exact correlated equilibria?”*
>
> Indeed, Theorem 3.9 can be more generally expressed with respect to approximate correlated equilibria (as Theorem 3.3). In light of Proposition 3.2, this can lead to a substantial improvement in the convergence bounds; we will point this out in the revised version.
>
> *“To what extent do you think that the results are tight?”*
>
> We believe that the dependence on the variation measure $\mathcal{V}^{(T)}_{\epsilon-NE}$ is unavoidable for any online learning algorithm, but it is less clear whether the dependence on $\mathcal{V}_A^{(T)}$ is necessary; closing the upper and lower bounds here requires further work.
>
> *“Do the results for two-player zero-sum games generalize to multiagent zero-sum polymatrix games? I don't see any inherent obstacles to do that using your approach, but I might be missing something.”*
>
> Indeed, as we point out in Remark A.5, Property 3.1 can be generalized to any time-varying variational inequality problem that satisfies the MVI property, which includes zero-sum polymatrix games. As such, our analysis readily carries over; we will point this out in the revised version.
>
> *“Another class of general sum games that is tractable in the single-shot setting are games in which the underlying matrices are rank-1. Is there any hope to obtain similar results for this class of games? I would imagine that the techniques would need to be substantially different from your approach.”*
>
> This is an interesting question. We believe that the MVI property no longer holds when the sum of the matrices is only known to be rank-1. So we agree that it seems to require very different techniques.
>
> *“In Theorem 3.3 (and similar results) if the parameter $L$  is not known does the usual guess-and-double trick work to get the bound?”*
>
> Depending on the normalization assumptions that we make, $L$ can be upper bounded by a parameter that depends on the number of actions of each player, and it is a mild assumption that this is known to the players. Alternatively, one could also use the doubling trick, as the reviewer suggested.

---

> > ### Comment · Reviewer_jL2o · 2023-08-10
> > **Authors' Rebuttal**
> >
> > I would like to thank the authors for their detailed response. I don't have any further questions.

---

### Official Review · Reviewer_PH29 · 2023-07-03

**Soundness:** 3 good
**Presentation:** 2 fair
**Contribution:** 3 good
**Rating:** 6
**Confidence:** 3

**Summary:**

This paper studies learning dynamics in games that change over time. This is a similar setting to [60], but while [60] focuses on regret guarantees, this paper focuses on iterate convergence to NE.

The main result states that for bilinear zero-sum games, running optimistic OGD guarantees that,
$$
\sum_{t=1}^T EqGap_t = O(V_{NE-\epsilon}^T + V_A^T),
$$
where $EqGap_t$ is the NE gap (i.e., the difference between the player utility and best response), $V_{NE-\epsilon}^T$ is the variation of $\epsilon$-approximate NEs of the games $+\epsilon T$ (in fact they allow different $\epsilon$ for each $T$), and $V_A^T$ measures the variation of the game matrices. $V_{NE-\epsilon}^T$ can be much smaller compared to the variation of the exact NEs.

The paper continues by providing variation-dependent bound for strongly convex games. Next, they provide a bound on the sum of NE gaps for general sum-potential games (which depends on some notion of variation of the potential function), as well as bound on the sum of CE gaps in general games. Finally, they present results for *dynamic* regret in static games.

**Strengths:**

The paper provides a set of interesting results. These include,
- Various results on the sum of NE gaps. Specifically, I find the notion of $V_{NE-\epsilon}^T$ very elegant, and indeed, it seems like a much more reasonable notion for characterizing the complexity of time-varying-games instance.
- Bounds on the dynamic regret in static games - even though this is a basic question, according to the authors these are the first results that show $\sqrt{T}$ dynamic regret in games (they also show $\log T$ dynamic regret under a stronger feedback model)


**Weaknesses:**

- The main text lacks proofs/proof sketches. So it is impossible to understand the main ideas and techniques, even at a high level, without diving into the full technical proofs in the appendix (+ it makes it hard to evaluate the technical contribution of the paper).

- In several places, it is quite hard to follow the text. Specifically, section 1.1 is rather technical, given that it is part of the intro. In addition, the section on general games that starts in line 287 was not sufficiently clear to me. For example, the authors mention that *"there exist matrices
$A_1, . . . , A_n$, with each matrix $A_i$ depending solely on the utility of Player i..."* but what are exactly these matrices represent? What does the value of the optimization problem in (3) represents? Why does *"incorporating the 0 vector will be useful"*? and why does there exist $\mu^\star$ that satisfy the condition in line 300?

- Lack of comparison to previous work: what is the result of [30] and how is it compars to your result in "meta-learning"? How does Corollary 3.4(2) compares to the result in [60] (except for the difference between $V_{NE-\epsilon}^T$ and $V_{NE}^T$)

**Questions:**

What is the difference between the sum NE gaps (Corollary 3.4) and dynamic regret (line 342)?

See additional questions in the Weaknesses part.

---

> ### Author Rebuttal · Authors · 2023-08-08
>
> We are grateful to the reviewer for their feedback.
>
> *“The main text lacks proofs/proof sketches. So it is impossible to understand the main ideas and techniques, even at a high level, without diving into the full technical proofs in the appendix”*
>
> We will make sure to provide high-level proof sketches in the main body, so that the main body is self-contained in the revised version.
>
> *“the section on general games that starts in line 287 was not sufficiently clear to me.”*
>
> We will provide a more self-contained presentation regarding the derivation of the bilinear formulation of correlated equilibria in the revised version; a textbook treatment can be found, for example, in Chapter 12 in the book “Game Theory Basics” by Von Stengel. Below, we address the reviewer’s questions regarding the bilinear formulation.
>
> This bilinear formulation represents a game between an additional player (namely, the mediator) and the set of players. Each player tries to optimally deviate from the mediator strategy, while the mediator is trying to find a strategy so that no player has an incentive to deviate, which is by definition a correlated equilibrium. So, each matrix $A_i$ encodes the deviation benefit of player $i$ (assuming that all other players are following the mediator's recommendation), and the value of the optimization problem (3) is the sum of the players’ deviation benefits. Notice that since a correlated equilibrium exists, there exists a mediator strategy $\mu^\star$ such that $(\mu^\star)^\top A_i x_i \leq 0$, for any player $i$. The same of course holds by allowing players to select strategies in conv$(X_i, 0)$ (since we just multiply by a nonnegative scalar). So, the bilinear formulation remains legitimate after taking the convex hull with $0$, and it proves that a $\mu^\star$ satisfying the condition of Line 300 indeed exists.
>
> *“Lack of comparison to previous work: what is the result of [30] and how is it compars to your result in "meta-learning"?”*
>
>
> Compared to the results in [30], we note that our guarantees depend on different similarity metrics, which are in general incomparable; yet, we remark that there are settings in which our similarity metric can be arbitrarily smaller than the one in [30], even in zero-sum games. Furthermore, for general-sum games, we obtain algorithm-independent similarity metrics, which was left open in that work.
>
> The algorithms we employ are also different. In particular, unlike [30], our approach is essentially agnostic to the meta-learning in that we do not need to know the boundaries of each game. Instead, our meta-learning guarantees are byproducts of our results for time-varying games, which is a more general problem than meta-learning.
>
>
> *“How does Corollary 3.4(2) compares to the result in [60]”*
>
> First, the authors of [60] focus on regret minimization in time-varying games, while Corollary 3.4 provides guarantees of iterate-convergence to Nash equilibria. Those two problems are in general unrelated; for example, even in static games an algorithm can have vanishing regret but at the same time all iterates can have a large Nash equilibrium gap. Moreover, all of our results concern the behavior of optimistic mirror descent (OMD), while the algorithm in [60] is more complicated. Given the tremendous amount of interest OMD has received in recent years, understanding its behavior is a question of independent interest. Finally, leveraging the connection we make with the MVI property (Remark A.5), our Corollary 3.4 directly applies to time-varying variational inequality problems as well, such as time-varying zero-sum polymatrix games, not just time-varying bilinear saddle-point problems.
>
>
> *“What is the difference between the sum NE gaps (Corollary 3.4) and dynamic regret (line 342)?”*
>
> Those are indeed the same; as such, notice that Line 342 is in fact a direct consequence of Corollary 3.4.

---

> > ### Comment · Reviewer_PH29 · 2023-08-14
> >
> > Thank you for your response. I have no further questions.

---

### Official Review · Reviewer_PhTp · 2023-07-10

**Soundness:** 4 excellent
**Presentation:** 4 excellent
**Contribution:** 4 excellent
**Rating:** 7
**Confidence:** 3

**Summary:**

The paper studies optimistic gradient descent for time-varying games.  Authors prove convergence bounds for zero sum games involving the first order variation of approximate nash equilibrium, which can be significantly tigher than variation bounds involving exact nash equilibria and second order bounds in payoff matrices. The paper also includes refined convergence bounds involving second-order variation for strongly-convex-concave games.  The results also have implications for meta-learning, where games are repeated many times.

The authors also extend results to time-varying general-sum multiplayer games with correlated equilibria, extending exisiting meta-learning similarity measures to general sum games and proving new single-player regret bounds. Techniques are applied to static games, improving our understanding of dynamic regret.

**Strengths:**

Nonstationarity is an important and challenging area of study for learning in games, and is under-explored.

Bounds involving approximate nash equilibrium variation can be significantly tighter than existing bounds involving variation in exact nash equilibria.

The results for general sum games solving two independent open problems.

Ideas provide improved understanding of dynamic regret for static games, including both positive and negative result.

The paper is well written, including a variety of results while still providing technical exposition on insights behind the proofs and contextualizing the result.

**Weaknesses:**

Observation 3.11 requires two point feedback, so it's less clear this is a significant improvement.

The paper could probably be a bit more self contained.  The paper borrows ideas from [60], like a dynamic RVU bound but it is hard to follow without additional context.

**Questions:**

Typo 88: accelerates-> accelerated

**Limitations:**

Yes.

---

> ### Author Rebuttal · Authors · 2023-08-08
>
> We are grateful to the reviewer for their feedback.
>
> We will make sure to provide further background in order to make the paper more self-contained in the revised version. We also thank the reviewer for spotting a typo; we will fix it in the revised version.

---

> > ### Comment · Reviewer_PhTp · 2023-08-15
> >
> > Thank you for your response. I have no other comments at this time.

---

### Decision · Program_Chairs · 2023-09-21

**Decision:**

Accept (poster)

**Comment:**

A paper with overall nice results on learning eps-Nash, and in my view, a good contribution to the growing no-regret learning in games literature that has been receiving increasing attention. In the accepted camera-ready version, please incorporate one of the reviewers' comments in the introduction by clearly describing the diff with respect to [60] and [Anagnostides et al., 2022]: what set of techniques are combined to lead to what new results that were previously unknown.